# Towards Learning High-Precision Least Squares Algorithms with Sequence Models

**Jerry Liu**[*][†]
Institute of Computational & Mathematical Engineering
Stanford University
Stanford, CA, USA

**Jessica Grogan**[†]**, Atri Rudra**
Department of Computer Science & Engineering
University at Buffalo
Buffalo, NY, USA

**Owen Dugan, Ashish Rao, Simran Arora, Chris Ré**
Department of Computer Science
Stanford University
Stanford, CA, USA

## Abstract

This paper investigates whether sequence models can learn to perform numerical algorithms, e.g. gradient descent, on the fundamental problem of least squares. Our goal is to inherit two properties of standard algorithms from numerical analysis: (1) *machine precision*, i.e. we want to obtain solutions that are accurate to near floating point error, and (2) *numerical generality*, i.e. we want them to apply broadly across problem instances. We find that prior approaches using Transformers fail to meet these criteria, and identify limitations present in existing architectures and training procedures. First, we show that softmax Transformers struggle to perform high-precision multiplications, which prevents them from precisely learning numerical algorithms. Second, we identify an alternate class of architectures, comprised entirely of polynomials, that can efficiently represent high-precision gradient descent iterates. Finally, we investigate precision bottlenecks during training and address them via a high-precision training recipe that reduces stochastic gradient noise. Our recipe enables us to train two polynomial architectures, gated convolutions and linear attention, to perform gradient descent iterates on least squares problems. For the first time, we demonstrate the ability to train to near *machine precision*. Applied iteratively, our models obtain $100,000\times$ lower MSE than standard Transformers trained end-to-end and they incur a $10,000\times$ smaller generalization gap on out-of-distribution problems. We make progress towards end-to-end learning of numerical algorithms for least squares.

## 1 Introduction

Least squares is the workhorse of modern numerics: it is well understood theoretically (Boyd & Vandenberghe, 2004; Trefethen & Bau, 2022) and has important downstream applications in science and engineering, including solving regression problems and differential equations (Orszag, 1972; Trefethen, 2000). Thus, least squares has gained interest as a natural testbed for investigating how well ML models can learn to implement algorithms (Garg et al., 2022; Von Oswald et al., 2023).

A surge of recent work suggests that Transformers (Vaswani et al., 2017) can learn to solve least squares using *optimization algorithms* like gradient descent and Newton's method (Akyürek et al., 2022; Fu et al., 2023; Ahn et al., 2024; Bai et al., 2024; Zhang et al., 2023b). These arguments rest on two observations: (1) simplified Transformer architectures (e.g. non-causal linear attention) can exactly implement such algorithms; (2) standard (softmax attention) Transformers learn solutions with similar properties (e.g. convergence rates) as iterative algorithms. Crucially, these works focus on *statistical* least squares: they evaluate Transformer solutions in underdetermined/noisy settings and compare to Bayes-optimal estimators. However, scientific applications like climate or fluids

---

[*]Corresponding author: `jwl50@stanford.edu`.
[†]Equal contribution.

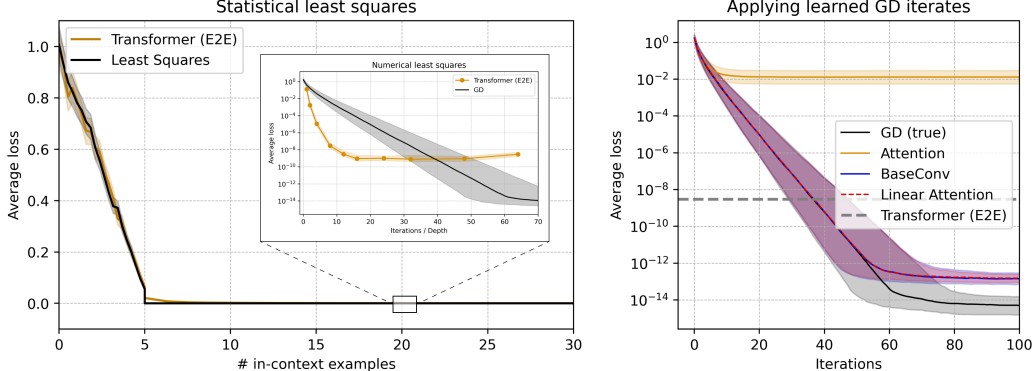

Figure 1: Prior work focuses on *statistical* least squares: Transformers approximate Bayes-optimal estimators (left, adapted from Garg et al. (2022)). In this work, we focus on *numerical* least squares: Transformers struggle to obtain precise solutions (inset). Using a high-precision training recipe, we train two polynomial architectures, BASECONV and linear attention, to perform high-precision gradient descent iterates on least squares (right): applied iteratively, they reach $\approx 10^{-13}$ MSE.

modeling require *numerically precise* solutions to least squares, e.g. to accurately model turbulence or to maintain stable temporal rollouts (Frisch, 1995; Wilcox, 2006). Prior works do not engage with the issue of high precision, so it is still unclear how well Transformers can solve least squares from this perspective.

In this work, we thus study whether existing approaches can solve *numerical* least squares. Specifically, numerical analysis requires that algorithms exhibit (1) *machine precision*, i.e. they should obtain solutions that are accurate to near floating point error, and (2) *numerical generality*, i.e. they are computational procedures that should apply broadly across problem instances. (See Section 2.1 for details.) Since traditional least squares algorithms (e.g. gradient descent and conjugate gradients) provably meet these criteria (Trefethen & Bau, 2022), it is crucial to evaluate machine learning methods against these same standards to determine their ability to learn numerical algorithms.

We focus on learning the gradient descent (GD) algorithm for least squares. Our study has three parts:

- **We benchmark standard Transformers for precision/generality and identify an expressivity gap on least squares.** When we replicate the standard end-to-end training setup for least squares with Transformers, we find that solutions do not exhibit machine precision and numerical generality (Figure 1a, 2). We identify high-precision multiplications as a fundamental challenge for softmax Transformers. Empirically, on a synthetic element-wise multiplication task, we find precision scales poorly with larger Transformers: an 8-layer model trains to an MSE that is still *10 million* times worse than machine epsilon (Figure 3). Theoretically, we argue that a single layer of softmax attention is unable to exactly express element-wise multiplications. Since implementing GD involves high-precision multiplications, this observation suggests standard Transformers are unable to even precisely *express* GD, much less precisely *learn* the algorithm.

- **We identify an alternate architecture class which does not suffer from expressivity problems.** Motivated by the expressivity limitations of softmax attention, we investigate alternate sequence mixer architectures. Prior work notes that non-causal linear attention is able to exactly implement algorithms like GD and Newton's method (Von Oswald et al., 2023; Giannou et al., 2024) because it consists entirely of *polynomials*. We provide a unified framework to understand existing expressivity results from the lens of arithmetic circuits. In our work, we focus on BASECONV, a gated convolutional architecture, as a case study, since it is provably equivalent to the entire class of polynomial architectures (Arora et al., 2023; 2024). We demonstrate that gated convolutions can express a high-precision GD algorithm ($\approx 10^{-13}$ MSE when implemented in practice, Figure 4).

- **We identify an optimization precision bottleneck and propose a high-precision training recipe.** Although polynomial architectures can precisely express the GD algorithm, we find that standard training procedures struggle to find a solution with sufficiently high precision ($10^{-5}$ MSE, Figure 9). Therefore, towards disentangling precision bottlenecks during training, we first focus on the intermediate task of explicitly learning GD iterates. We identify stochastic gradient noise from

minibatching as the main optimization bottleneck, and we find that a simple metric, cosine similarity of minibatch gradients (Liu et al., 2023b), is diagnostic of precision saturation. Towards reducing stochasticity, we propose (1) a learning rate (LR) scheduler that adaptively adjusts LR based on the cosine similarity metric, and (2) to apply EMA over optimizer updates to maintain strong gradient signal. Our high-precision training recipe allows us to train ML architectures to near *machine precision* for the first time. We successfully train two 3-layer models, with BASECONV and linear attention, that learn to perform a single high-precision iteration of GD (Figure 1b). Excitingly, we can also learn multiple GD iterates at once, scaling up to 4 iterations with $10^{-10}$ MSE.

Overall, our work makes the following contributions: (1) we specify the desiderata of learning numerical algorithms, *machine precision* and *numerical generality*, and we demonstrate that standard Transformers fall short because of expressivity limitations of softmax attention; (2) we provide a unified framework using arithmetic circuits to investigate the expressivity of the class of polynomial architectures; (3) we address additional precision bottlenecks that emerge *during training*, even when using expressive polynomial architectures. Although we do not achieve end-to-end learning of GD, we make significant headway: we propose a high-precision training recipe, which, for the first time, allows us to learn iterates of the GD algorithm to near *machine precision*.

## 2 LEARNING NUMERICAL ALGORITHMS FOR LEAST SQUARES

In this section, we distinguish between statistical vs. numerical least squares and discuss the two properties we want our models to inherit from numerical algorithms: *machine precision* and *numerical generality*. We then briefly discuss prior work and, in doing so, tease apart two increasingly end-to-end notions of performing algorithms with ML: *expressing an algorithm in-weights* and *learning algorithm iterates*.

### 2.1 PROBLEM FORMULATION AND RELATED WORK

In this work, our goal is to train a model that solves least squares problems: find $x \in \mathbb{R}^D$ given $A \in \mathbb{R}^{N \times D}$ and $b \in \mathbb{R}^N$ such that $Ax = b$. Here, we briefly discuss two different perspectives on least squares: statistical (in the form of *in-context learning*) and numerical.

**Statistical least squares.** Originally motivated by applications in language modeling, prior works on solving least squares with Transformers typically take a *statistical* perspective. Transformers are trained using an *in-context learning* setup (Garg et al., 2022; Akyürek et al., 2022): problem instances $Ax = b$ are sampled from a pre-specified distribution $\mathcal{D}_{train}$, and the model is trained to minimize mean squared error (MSE) over $\mathcal{D}_{train}$. Trained models are then evaluated on unseen problem instances, both in and out-of-distribution, and their performance is compared to Bayes-optimal estimators (Garg et al., 2022; Akyürek et al., 2022). We define the in-context least squares training setup in Appendix B and leave a more detailed discussion of related in-context learning work to Appendix A.

**Numerical least squares.** In this work, we instead take a *numerical* perspective on least squares. A prototypical numerical algorithm for least squares is GD. For a problem instance $Ax = b$, we initialize $x_0$, an estimate of $x$, and iteratively improve our estimate via

$$x_{i+1} = x_i - \eta \nabla \mathcal{L}(x_i), \tag{1}$$

where $\mathcal{L}(\hat{x}) := \frac{1}{2}||A\hat{x} - b||_2^2$ is the squared residual error. GD exhibits two properties of numerical algorithms that we want our models to inherit:

- *Machine precision.* Numerical algorithms provably obtain high-precision solutions. For GD, obtaining higher precision simply requires performing more iterations until convergence to machine precision (i.e. the smallest achievable error with floating-point arithmetic) (see Chapter 11 of Trefethen & Bau (2022)). In this work, we use `float32` throughout, where machine precision is $2^{-23} \approx 1.19 \times 10^{-7}$, so we hope for MSEs around $2^{-46} \approx 1.42 \times 10^{-14}$.
- *Numerical generality.* Although the convergence rate of GD depends on the spectrum of $A$ (see Chapter 9 of Boyd & Vandenberghe (2004)), the computational procedure comprising GD is general and can be applied broadly to problem instances. This is unlike statistical generalization and notions of in vs. out-of-distribution. In this work, we are interested to study how closely ML models can emulate the numerical generality of algorithms despite training on a data distribution.

## 2.2 OUTLINE OF THIS WORK

A recent line of work probes the estimators learned by Transformers on in-context least squares, and suggests that Transformers learn to solve least squares by mimicking iterative algorithms like gradient descent and Newton's method (Von Oswald et al., 2023; Ahn et al., 2024; Fu et al., 2023; Giannou et al., 2024). These works typically analyze simplified models theoretically and extrapolate to standard training regimes, backed by empirical observations:

- **Theoretical results for simplified models**, e.g. *non-causal linear attention* can implement GD using a specific choice of model weights.
- **Empirical experiments training standard Transformers**, e.g. decoder-only softmax attention Transformers trained *end-to-end* on in-context least squares display convergence rates reminiscent of iterative algorithms.

Although prior works suggest that trained Transformers learn to solve least squares with algorithms, it is still unclear whether statements about learning algorithms in simplified settings transfer to standard Transformers trained end-to-end. We note two significant gaps between previously-analyzed settings and standard in-context least squares:

- **Architectural differences.** Standard Transformers use softmax instead of linear attention, causal instead of non-causal sequence mixers, and include MLPs and LayerNorms (Ba et al., 2016).
- **Optimization.** Even if a model can express a precise and general algorithm, it is unclear whether the model can learn the algorithm from data.

In this work, we tease apart bottlenecks caused by architecture expressivity limitations (Sections 3.3, 4) and optimization difficulties (Section 5) by investigating two increasingly sophisticated notions of performing GD for least squares with ML: *expressing GD in-weights* and *learning GD iterates*.

## 3 TRANSFORMERS DO NOT LEARN NUMERICAL ALGORITHMS IN-CONTEXT

In this section, we evaluate standard Transformers, trained end-to-end, on the criteria of *machine precision* and *numerical generality*. Surprisingly, we demonstrate that existing approaches fail to exhibit these properties: the precision of Transformer solutions (in MSE) saturates a *million* times worse than machine precision (Section 3.1), and their performance further degrades as problem instances deviate from the model's training distribution (Section 3.2). These results suggest that Transformers are not learning proper algorithms as numerical analysis defines them.

Towards identifying expressivity bottlenecks, we identify three linear algebra primitives that comprise standard algorithms including GD and Newton's method (Section 3.3). We find empirically that Transformers struggle to implement high-precision multiplication, and theoretically we argue that softmax attention faces an expressivity gap when trying to exactly express multiplications.

### 3.1 TRANSFORMERS STRUGGLE TO REACH MACHINE PRECISION

Recent work (Von Oswald et al., 2023; Ahn et al., 2024; Fu et al., 2023; Giannou et al., 2024) studying in-context least squares suggests that Transformers learn to mimic iterative algorithms like GD and Newton's method. Note that if Transformers are able to implement iterative algorithms, the depth of the model should correspond to the number of iterations performed. We thus focus on the simplest case of fully determined least squares problems with fixed size design matrices and investigate whether precision improves as we scale to larger and deeper models.

In Figure 1b, following prior work (Ahn et al., 2024), we fix the size of $A \in \mathbb{R}^{20 \times 5}$ and train Transformers end-to-end on least squares, scaling up to $L = 64$ layers. We compare their precision to the convergence rate of the full-batch gradient descent algorithm on least squares. For more details about the training setup, refer to Appendix B.3.1.

At first, Transformer precision scaling exceeds the convergence rate of gradient descent: this finding mirrors similar results reported by Fu et al. (2023), who suggest Transformers may instead be learning higher-order algorithms like Newton's method. However, we further observe that the precision gains for Transformers *rapidly diminish*, such that we observe very little difference in precision between $L = 32$ and $L = 64$ layers. The deepest Transformer models we are able to train achieve an MSE around $10^{-8}$. In contrast, gradient descent converges linearly to machine precision, almost $1,000,000\times$ better precision. The diminishing returns of the Transformer precision scaling imply that Transformers are not learning standard numerical algorithms like GD.

## 3.2 Transformers do not exhibit the generality of gradient descent

We further investigate whether Transformers learn solutions to least squares that exhibit numerical generality. Recall that models are trained on a predefined distribution of least squares problems, $\mathcal{D}_{train}$. If Transformers learn to solve least squares using a standard numerical algorithm like GD, then we expect the performance of the model should be robust to out-of-distribution inputs.

For GD specifically, the convergence criterion $(0 < \eta < 2/\sigma_{max}^2)$ depends on $\sigma_{max}$, the maximum singular value of $\boldsymbol{A}$, and the optimal rate of convergence depends on the condition number of $\boldsymbol{A}$, $\kappa = \sigma_{max}/\sigma_{min}$ (Boyd & Vandenberghe, 2004). Thus we specify our training distribution $\mathcal{D}_{train}$ over least squares problems ($\boldsymbol{A} \in \mathbb{R}^{20\times5}$, $\boldsymbol{b} = \boldsymbol{Ax} \in \mathbb{R}^{20}$) as follows. First, as in prior work (Garg et al., 2022), we sample the entries of $\boldsymbol{A}$ and $\boldsymbol{x}$ i.i.d from a standard Gaussian $N(0,1)$. We then shift and rescale the singular values of $\boldsymbol{A}$ so that $\sigma_{max} = \kappa = 5$. After training a 12-layer Transformer model on the in-context objective, we evaluate our model on out-of-distribution regression targets $\boldsymbol{b}$.

We define $\mathcal{D}^b_{OOD}(\sigma)$ by sampling each entry of $\boldsymbol{x}$ i.i.d. from $N(0,\sigma)$ and computing $\boldsymbol{b} = \boldsymbol{Ax}$. Al-

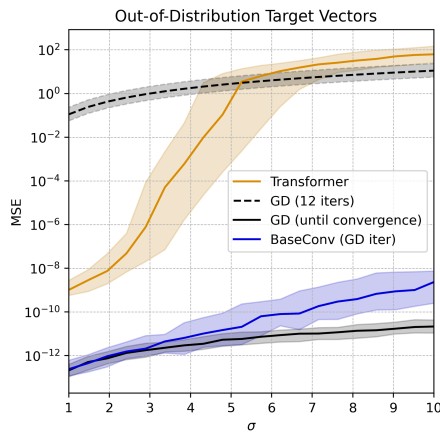

Figure 2: Transformers generalize poorly to out-of-distribution regression targets. In contrast, using our training recipe, we train a BaseConv model to perform high-precision GD iterates. Applied iteratively, our BaseConv model incurs $10,000\times$ less generalization error on out-of-distribution target vectors than the Transformer.

though the distribution of $\boldsymbol{b}$'s and $\boldsymbol{x}$'s changes with $\sigma$, because the spectra of the $\boldsymbol{A}$'s is consistent, we know that GD with fixed choice of $\eta$ will provably converge to high precision.

We find that compared to GD, the Transformer solutions are brittle to unseen regression target distributions. Simply scaling the inputs by a factor of $10\times$, the MSE of the trained Transformer degrades by a factor of $10^8$. In contrast, GD is robust: a fixed number of GD iterations consistently converges to the same order of magnitude of precision (Figure 2). The brittleness of the learned Transformer solution compared to GD again suggests that Transformers are not performing standard numerical algorithms.

## 3.3 Identifying an expressivity gap with standard Transformers

Toward understanding the limitations of the Transformer architecture, we start with GD and Newton's method, two algorithms used to solve least squares, and look into primitives that comprise them.

**Linear algebra primitives.** We observe that GD and Newton's method can be expressed as compositions of three simple linear algebra operations: sequence-wise read/write (READ), affine transformations (LINEAR), and element-wise multiplications (MULTIPLY). For input $\boldsymbol{u} \in \mathbb{R}^{N\times D}$:

$$\text{READ}(i,j,a,b)(\mathbf{u}) = \begin{cases} \mathbf{u}[k, a{:}b] & k \neq j \\ \mathbf{u}[i, a{:}b] & k = j \end{cases},$$

$$\text{LINEAR}(\boldsymbol{H})(\mathbf{u}) = \mathbf{u}\boldsymbol{H}, \quad \text{where } \boldsymbol{H} : \mathbb{R}^D \to \mathbb{R}^{d_{out}} \text{ is linear,}$$

$$\text{MULTIPLY}(a,b,d_{out})(\mathbf{u}) = \mathbf{u}[:, a{:}a{+}d_{out}] \odot \mathbf{u}[:, b{:}b{+}d_{out}]$$

In Appendix D.2, we define these primitives formally and describe how GD and Newton's method iterates can each be expressed as a composition of these primitives. Intuitively, READ is required to transfer information across the sequence dimension, LINEAR to transfer information across the hidden dimension, and MULTIPLY to compute high-degree interaction terms (like dot products or element-wise squaring).

**Empirical analysis: standard Transformers struggle with multiplication.** We train Transformers on synthetic formulations of these tasks to investigate how precision scales with model size. Details about our training setups are in Appendix B.3.2.

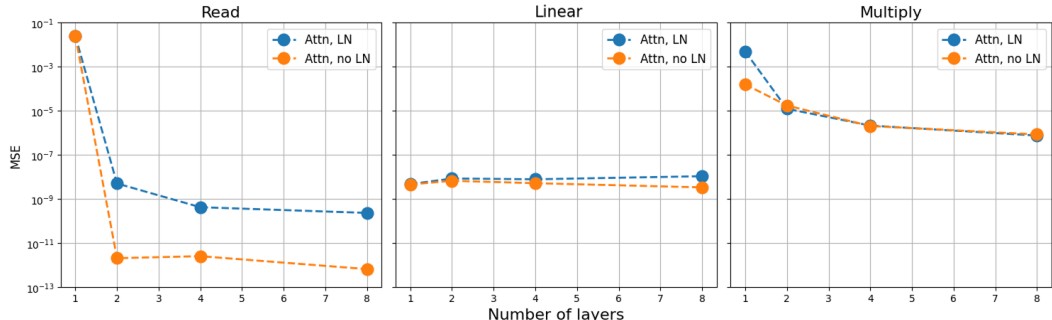

Figure 3: Precision vs. Transformer depth, with and without LayerNorms (LN), on synthetic tasks. While shallow Transformers are able to learn the READ and LINEAR tasks to high precision ($< 10^{-8}$ with 2-layer models), precision on the MULTIPLY task scales poorly with depth (only $10^{-6}$ with 8-layer models).

In Figure 3, we show that even 2-layer Transformers are able to achieve $10^{-8}$ MSE on the READ and LINEAR tasks. However, we find that Transformers struggle with the MULTIPLY task: precision scales poorly with model depth, such that an 8-layer Transformer is only able to achieve $10^{-6}$ MSE. In Appendix C.1, we further show that precision on the MULTIPLY task also scales poorly with increased attention dimension, number of attention heads, and MLP upscaling factor.

**Theoretical analysis: softmax attention struggles to exactly express multiplication.** In Appendix D.2.4, we provide a proof that a single layer of softmax attention cannot exactly express the simple element-wise squaring function $\text{SQUARE}(\boldsymbol{u})[i, j] = \boldsymbol{u}[i, j]^2$ (intuitively, because softmax cannot implement polynomials). Crucially, we note that element-wise squaring is a special case of element-wise multiply, so softmax attention cannot exactly implement MULTIPLY either:

**Theorem 3.1** (Informal statement of Theorem D.31 and Corollary D.32)**.** *One-layer single-headed (causal) softmax attention cannot exactly represent* SQUARE *and* MULTIPLY *for all possible inputs.*

Since precisely implementing numerical algorithms like GD hinges on performing high-precision multiplications, this result suggests that the standard Transformer architecture struggles to precisely implement these algorithms because of a fundamental expressivity gap.

We mention briefly that these findings do not conflict with prior results (Yun et al., 2020b) proving universal approximation theorems for Transformers, because they typically require parameter count to scale exponentially with dimension: see Appendix A.

## 4 ALTERNATE ARCHITECTURES CLOSE THE EXPRESSIVITY GAP

Motivated by the finding that softmax attention struggles to precisely express multiplications, we next investigate alternate sequence mixer architectures. We are inspired by prior results (Von Oswald et al., 2023; Giannou et al., 2024) that show non-causal linear attention is able to exactly implement algorithms like GD and Newton's method. Thus, we focus on the class of *polynomial architectures*, i.e. sequence mixers comprised entirely of polynomial operations, in order to explicitly bake in multiplications. In this section, we present a unified framework that integrates previous findings through the perspective of arithmetic circuits. Specifically, we focus on BASECONV, a gated convolutional model that combines element-wise multiplications (gating) with long convolutions. We work with BASECONV for two reasons:

- Recent work (Arora et al., 2023; 2024) has shown that BASECONV is equivalent to general arithmetic circuits, including all polynomial architectures. Thus, existing results with other polynomial architectures, e.g. linear attention, transfer directly to BASECONV.
- Empirically, gated convolutional models have been shown to perform comparably to attention-based architectures on tasks like language, audio, and DNA modeling (Arora et al., 2024; Nguyen et al., 2024; Zhang et al., 2023a).

We emphasize that although we find gated convolutions are convenient to work with theoretically and empirically, we believe that other sequence mixer architectures may also be able to alleviate the expressivity issues we highlight in Section 3.3. In particular, we show promising empirical results for non-causal linear attention in Section 5.2.

## 4.1 GATED CONVOLUTIONS ARE EQUIVALENT TO ARITHMETIC CIRCUITS

**BASECONV definition.** In this work, we focus on a variant of the BASECONV operator from Arora et al. (2023). Given an input $\boldsymbol{u} \in \mathbb{R}^{N \times D}$, BASECONV($\boldsymbol{u}$) is defined as:

$$(\underbrace{(\boldsymbol{u}\boldsymbol{W}_{gate} + \boldsymbol{b}_{gate})}_{\textbf{Linear Projection}} \odot \underbrace{(\boldsymbol{h} * (\boldsymbol{u}\boldsymbol{W}_{in} + \boldsymbol{b}_{in}) + \boldsymbol{b}_{conv})}_{\textbf{Convolution}})\boldsymbol{W}_{out} + \boldsymbol{b}_{out} \tag{2}$$

where the layer is parameterized by learnable filters $\boldsymbol{h} \in \mathbb{R}^{N \times D}$, linear projections $\boldsymbol{W}_{in}, \boldsymbol{W}_{gate}, \boldsymbol{W}_{out} \in \mathbb{R}^{D \times D}$, and bias matrices $\boldsymbol{b}_{conv}, \boldsymbol{b}_{in}, \boldsymbol{b}_{gate}, \boldsymbol{b}_{out} \in \mathbb{R}^{N \times D}$. Here, $\odot$ represents the Hadamard product, and convolution of two matrices is computed as convolution of the corresponding columns.

**BASECONVS can exactly express linear algebra primitives.** In Appendix D.2.1, we provide explicit constructions of single-layer BASECONV models that exactly implement the READ, LINEAR, and MULTIPLY primitives from Section 3.3.

We note that this result is stronger than prior BASECONV expressivity results (e.g. Theorem H.21 from Arora et al. (2023)), which imply a poly-log-factor increase in parameters (specifically layers) translating from arbitrary arithmetic circuits. Here, we show by construction that these specific primitives, and any circuits that are compositions of them, incur only a constant factor loss.

**BASECONVS can perfectly recover linear algebra primitives from data.** In Appendix D.5, for SQUARE and LINEAR, we further show the following under mild assumptions, which our input distribution satisfies (see details in Assumptions D.45, D.46, D.48, D.49):

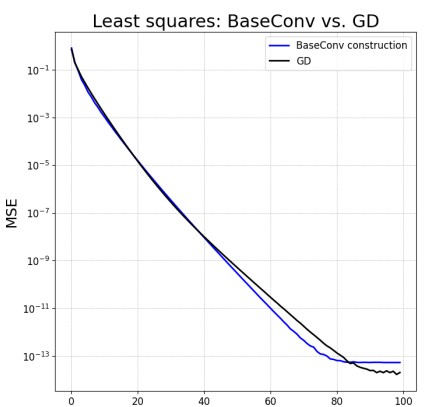

Figure 4: BASECONV can express high-precision gradient descent: our implementation of the weight construction reaches $10^{-13}$ MSE in practice.

**Theorem 4.1** (Informal statement of Theorems D.62, D.60). BASECONV *perfectly recovers* SQUARE *and* LINEAR *when it achieves zero population gradient w.r.t. MSE loss.*

We note that although results of the form "exact solution implies zero population gradient" exist in the literature (Ahn et al., 2024; Mahankali et al., 2023), to the best of our knowledge, we are the first to show the *converse* ("zero population gradient implies recovery of exact solution") for sequence model architectures. In Appendix C.1, we show that BASECONV models can learn the READ, LINEAR, and MULTIPLY primitives to high precision in practice (Figure 6).

**BASECONVS are universal approximators.** Finally, we show in Appendix D.4 that BASECONV can efficiently approximate *smooth functions* by implementing polynomials:

**Theorem 4.2** (Informal statement of Theorem D.39). *Given a* $k$*-times differentiable function* $\bar{f}$ : $[-1, 1] \to \mathbb{R}$*, define* $f : [-1, 1]^{N \to D} \to \mathbb{R}^{N \times D}$*, which applies* $\bar{f}$ *element-wise to all inputs. Then* $\forall \epsilon > 0$*, there exists a* BASECONV *model approximates* $f$ *to within error* $\epsilon$*, with* $O\left(\sqrt[k]{\frac{L}{\epsilon}}\right) + k$ *depth and* $O(ND)$ *parameters, where* $||f^{(k)}||_\infty \leq L$*.*

We additionally prove a universal approximation theorem for general smooth multivariate functions in Appendix D.4 (Theorem D.44).

## 4.2 BASECONV CAN PRECISELY EXPRESS GRADIENT DESCENT FOR LEAST SQUARES

We now focus on the gradient descent algorithm for least squares. Explicitly, given a least squares problem instance $\boldsymbol{A}\boldsymbol{x} = \boldsymbol{b}$ and an initial iterate $\boldsymbol{x}_0$, a single iteration of gradient descent computes

$$\boldsymbol{x}_1 := \boldsymbol{x}_0 - \eta \nabla \mathcal{L}(\boldsymbol{x}_0), \text{ where } \nabla_{\boldsymbol{x}} \mathcal{L} = \boldsymbol{A}^T(\boldsymbol{A}\boldsymbol{x} - \boldsymbol{b}). \tag{3}$$

We provide two explicit $O(1)$-layer weight constructions to express a GD iterate using BASECONV in Appendix D.3.1. One requires a $O(D)$ state size using a *non-causal* model (i.e. each entry can access

any other entry of the sequence) and one requires a $O(D^2)$ state size using a *causal* model (i.e. entries cannot access later entries of the sequence). In Appendix D.3.2, we prove that both constructions are asymptotically optimal with respect to state size.

In Figure 4, we implement our non-causal weight construction into a deep BASECONV model as a proof of concept. We confirm that gated convolutions can empirically implement high-precision gradient descent – notably, roundoff errors due to machine precision do not significantly accumulate in practice, despite scaling up to a depth-100 BASECONV model.

## 5 TOWARDS TRAINING MODELS TO MACHINE PRECISION

Although BASECONVs are expressive enough to solve least squares precisely, we find that simply swapping out softmax attention with BASECONV and training end-to-end is insufficient for high precision: our BASECONV models perform as poorly as standard Transformers (Figure 9). This suggests that additional precision bottlenecks are present *during high-precision training*. In this section, we thus investigate what it takes to train polynomial architectures to machine precision.

Recent works (Rodionov & Prokhorenkova, 2023; 2024) on algorithm learning find that intermediate supervision is crucial for learning long computation trajectories. We hypothesize that end-to-end least squares faces a similar challenge. Thus, to study high-precision optimization, we first investigate a simplified setting: learning to perform *explicit GD updates* for least squares.

Using this task as a benchmark, we identify a fundamental bottleneck in high-precision regimes, *gradient variance from minibatching*, and we identify a metric based on cosine similarity of successive gradients that is diagnostic of precision saturation during training. We then propose a high-precision training recipe, which for the first time allows us to train ML models to near *machine precision*. Using our training recipe, we learn to perform explicit GD updates to $10^{-13}$ average MSE (Figure 1b), and we can also learn up to 4 iterates of GD at once with an MSE of $10^{-10}$ (Table 7).

**Simplifying the training setup.**   We first define a sequence of *k-th iterate* tasks, where the goal is to explicitly produce the $k$-th iterate of GD given a least squares problem instance $(\boldsymbol{A}, \boldsymbol{b})$, an initial iterate $\boldsymbol{x}_0$, and a step size $\eta$:

$$\{(\boldsymbol{a}_1, b_1), \ldots, (\boldsymbol{a}_N, b_N), \boldsymbol{x}_0\} \to \boldsymbol{x}_k, \text{ where } \boldsymbol{x}_{i+1} = \boldsymbol{x}_i - \eta \nabla \mathcal{L}(\boldsymbol{x}_i), \, i \in [k-1]. \tag{4}$$

We then define the *explicit gradient* task, where the goal is to produce the GD update vector:

$$\{(\boldsymbol{a}_1, b_1), \ldots, (\boldsymbol{a}_N, b_N), \boldsymbol{x}_0\} \to \nabla \mathcal{L}(\boldsymbol{x}_0). \tag{5}$$

Note that (up to a residual connection), the explicit gradient task is equivalent to 1-step GD, and standard in-context least squares is equivalent to taking $k \to \infty$. Thus, the explicit gradient task is a natural simplification of standard in-context least squares, and the $k$-th iterate task allows us to smoothly interpolate between the two extremes of difficulty. Refer to Appendix B for more details.

### 5.1 TOWARDS A HIGH-PRECISION TRAINING RECIPE

Our theoretical results in Section 4 imply that a 3-layer BASECONV is expressive enough to solve the explicit gradient task, so we use training a 3-layer BASECONV on this task as our benchmark for studying the challenges of high-precision learning.

**Precision saturates with standard training procedures.**   Motivated by prior work (Garg et al., 2022; Von Oswald et al., 2023; Ahn et al., 2024), we start by investigating two basic optimization procedures: Adam with constant learning rate (LR) and with exponentially decaying LR.

In Appendix C.2 (Figure 10), we sweep initial LR and LR steprate across 2-3 orders of magnitude for constant and decaying LR schedules. We find:

- *Precision saturation occurs with both constant and decaying LR schedules.* After a number of training iterations, the average loss saturates and is unable to improve. We note that this occurs even while gradients magnitudes and LR are non-zero.
- *Slower-decaying LR schedules perform better but require exponentially more training iterations.* In Figure 8, we further analyze this phenomenon in the simpler case of 1-layer Transformers/BASECONVs on the MULTIPLY synthetic. We observe a power-law relation between precision

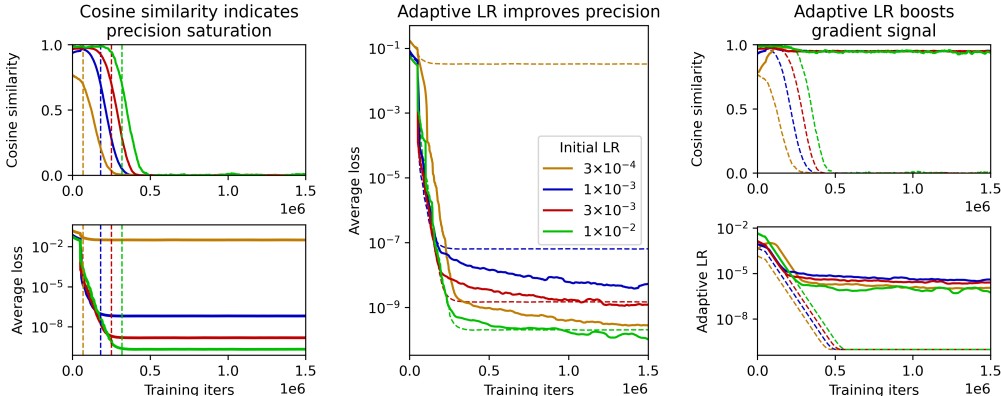

Figure 5: Gradient metric is predictive of precision saturation (left). We propose a simple adaptive LR scheduler that alleviates precision saturation (middle). Adaptive LR effectively boosts gradient signal during training (right).

and number of training iterations as we sweep steprate; although it may be possible to train to high precision in theory, this approach seems infeasible in practice.

- *With aggressively-decaying LR schedules, higher initial LR is better.* For a fixed scheduler step rate, increasing initial LR leads to significant improvements in final MSE, e.g. in Figure 10, an improvement of $1000\times$ simply by increasing initial LR from $10^{-3}$ to $10^{-2}$. Choosing a LR that is too large causes training instability, so in practice we set it to the *largest value that trains stably*.

Our analysis suggests that Adam with an exponentially decreasing LR scheduler gets us only part of the way to a machine precision training recipe. We next address the issue of precision saturation.

**Stochastic gradients bottleneck precision.** We identify *minibatch gradient variance* as the main source of precision saturation. Although our goal is to minimize the expected loss over problem instances from $\mathcal{D}_{train}$, in practice we minimize over finite minibatch samples instead. Minibatch training is the standard in ML, but interestingly we find that the variance in minibatch gradients can dominate the population gradient signal in high-precision regimes, causing the loss to stagnate.

To demonstrate this, we define a simple metric to assess the strength of the gradient signal during training. At a given training step, we take the current model weights, sample $n$ different minibatches of least squares problems, and compute the minibatch model gradients $\{g_1, \ldots, g_n\}$. We then compute the *average cosine similarity* between all pairs, as in Liu et al. (2023b):

$$\sigma_g := \frac{2}{n(n-1)} \sum_{i \neq j} \frac{g_i^T g_j}{||g_i||_2 ||g_j||_2} \tag{6}$$

We observe that this cosine similarity metric is predictive of precision saturation across MSE scales and optimizer hyperparameters (Figure 5).

**An adaptive LR scheduler boosts gradient signal beyond precision saturation.** We thus propose an *adaptive* LR scheduler based on the gradient variance. Our scheduler is motivated by two intuitions:

- Whenever the cosine similarity metric is *high*, gradient signal is strong. In order to refine the highest-precision bits of the model weights, we need to slowly *decrease* the LR.
- Whenever the cosine similarity metric is *low*, the model weights are stuck in a local region of the loss landscape. To allow the model to escape this region, we need to *increase* the LR.

The basic scheduler we use in this work simply decreases the LR exponentially while the metric is above a threshold $\sigma_{th}$ and increases the LR instead if the metric is below $\sigma_{th}$. In Figure 5, we show that this simple approach alleviates the loss saturation phenomenon: we see a boost in population loss across our LR settings, and we observe the models consistently improve as we continue training. We note that proper choice of LR hyperparameters is still crucial for *efficient* convergence to machine precision – we leave speeding up the convergence rate via better adaptive schedulers to future work.

**Exponential Moving Average (EMA) over optimizer updates.** Finally, motivated by our observation that gradient variance bottlenecks precision and inspired by recent works (Lee et al., 2024; Pagliardini et al., 2024), we apply an additional EMA over Adam's update vectors to help smooth out minibatch noise. Empirically, we find this boosts the final MSE by as much as $100,000\times$ on the explicit gradient task: see Appendix C.2 (Figure 11).

Our training recipe for efficient high-precision convergence thus involves two techniques: (1) an adaptive LR scheduler that exponentially increases or decays LR according to the cosine similarity metric; and (2) applying EMA over optimizer updates.

### 5.2 LEARNING HIGH-PRECISION GRADIENT DESCENT WITH POLYNOMIAL ARCHITECTURES

Using our training recipe, we successfully train two 3-layer models with polynomial architectures, BASECONV and non-causal linear attention, on the explicit gradient task. For the first time, we are able to train to near *machine precision*: we achieve an average loss of $10^{-13}$ MSE.

In Figure 1b, we slot our trained models into the standard GD algorithm, using their predictions in place of the true least squares gradients $\nabla\mathcal{L}$. Specifically, for a least squares problem $Ax = b$ and initial iterate $x_0$, we repeatedly compute $x_{i+1} := x_i - \eta\Delta_i$, where $\Delta_i := T_\theta(A, b, x_i)$ is the prediction of the model. We iteratively apply the model until convergence to a fixed point $x_\infty$.

We find that both our models achieve high precision. In this setting, we reach an average MSE of $10^{-12}$ (Figure 1, right): this is $100,000\times$ better MSE than the biggest Transformers we are able to train end-to-end. Moreover, our BASECONV model exhibits better numerical generality than the Transformer, incurring a $10,000\times$ smaller generalization gap on problems outside its training distribution (Figure 2). Interestingly, we find that our linear attention model exhibits markedly worse generality: its out-of-distribution performance nearly matches the Transformer's, and the model iterates eventually diverge: see Figure 13.

**Learning $k$-iterates of GD for larger $k$.** We find that our training recipe also allows us to learn up to $k = 4$ iterates of GD at once with $10^{-10}$ MSE: see Table 7 and Figure 14 for results. We are not able to stably train deeper models without reintroducing non-polynomial normalization techniques like LayerNorms, which causes precision bottlenecks. For small $k$, we observe that LayerNorms worsen precision by over $1,000\times$. See Appendix C.3 for details.

**Experiments with in-context ODE solving.** Finally, towards high-precision ML for more realistic tasks, we provide preliminary results on in-context ODE solving. We find that our proposed techniques outperform standard Transformers by up to $1,000,000\times$ in MSE (up to $\approx 10^{-10}$ with iterative BASECONVs vs. $\approx 10^{-4}$ with 12-layer Transformers). See Appendix C.4 for details.

## 6 DISCUSSION AND LIMITATIONS

In this work, we investigate learning to solve least squares from a numerical perspective. We find that Transformers fail to learn solutions that exhibit the properties of machine precision and numerical generality. Disentangling effects from the model architecture and optimizer, we find that standard design choices perform surprisingly poorly from the lens of numerics. We identify expressivity limitations with softmax attention, and find surprisingly that even MLPs and LayerNorms significantly affect precision (up to $1,000,000\times$ worse MSE on the explicit gradients task). On the optimization front, we find stochastic gradient noise from minibatch training becomes a precision bottleneck in high-precision regimes. We propose an adaptive LR scheduler that alleviates this issue on a simplified task, but we suspect that this issue remains a fundamental challenge on harder problems. Crucially, although we make progress toward learning to solve numerical least squares end-to-end, our techniques struggle to maintain stable and precise training with deep networks.

We note that the *numerical* criteria we consider in this work represent a fundamentally different type of learning and generalization from *statistical* notions that are prevalent in ML. We believe these numerical perspectives may be relevant to the wider scientific ML community. For example, existing approaches to solving PDEs have shown promise but are known to be brittle outside their training distributions (Wang & Lai, 2023; Rathore et al., 2024). This inhibits their usefulness in high-impact applications like climate or fluids modeling, where high precision and robustness are crucial. We believe learning to implement precise numerical algorithms directly from data is an exciting prospect that has the potential to unlock new capabilities across science and engineering.

REPRODUCIBILITY STATEMENT

We provide all the code and configuration files necessary to reproduce our experiments at https://github.com/HazyResearch/precision-ls. In this work, all experiments are done using synthetic data and tasks. All experiments were conducted using PyTorch on NVIDIA A100/H100 GPUs. Detailed hyperparameters (learning rate, batch size, and optimizer settings) and proofs of all theoretical claims are provided in the supplementary materials.

ACKNOWLEDGMENTS

We thank Yasa Baig, Mayee Chen, Rajat Dwaraknath, Sabri Eyuboglu, Chris Fifty, Neel Guha, Hermann Kumbong, Benjamin Spector, Aman Timalsina, Alyssa Unell, Ben Viggiano, Michael Zhang, and Dylan Zinsley for their helpful feedback and discussion during this work.

We gratefully acknowledge the support of NIH under No. U54EB020405 (Mobilize); NSF under Nos. CCF2247015 (Hardware-Aware), CCF1763315 (Beyond Sparsity), CCF1563078 (Volume to Velocity), 1937301 (RTML), DGE-2146755 (GRFP), and PHY-2019786 (IAIFI); US DEVCOM ARL under Nos. W911NF-23-2-0184 (Long-context) and W911NF-21-2-0251 (Interactive Human-AI Teaming); ONR under Nos. N000142312633 (Deep Signal Processing); Stanford HAI under No. 247183; NXP, Xilinx, LETI-CEA, Intel, IBM, Microsoft, NEC, Toshiba, TSMC, ARM, Hitachi, BASF, Accenture, Ericsson, Qualcomm, Analog Devices, Google Cloud, Salesforce, Total, the HAI-GCP Cloud Credits for Research program, the Stanford Data Science Initiative (SDSI), and members of the Stanford DAWN project: Meta, Google, and VMWare. The U.S. Government is authorized to reproduce and distribute reprints for Governmental purposes notwithstanding any copyright notation thereon. Any opinions, findings, and conclusions or recommendations expressed in this material are those of the authors and do not necessarily reflect the views, policies, or endorsements, either expressed or implied, of NIH, NSF, ONR, or the U.S. Government. JL is supported by the Department of Energy Computational Science Graduate Fellowship under Award Number DE-SC0023112. JG and AR's research is supported by NSF grant CCF#2247014. OD is supported by the Hertz Foundation Fellowship, the Stanford Knight-Hennessy Scholarship, and the NSF GRFP.

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

# APPENDIX

The appendix is organized as follows:

- Appendix A provides a more detailed overview of related work.
- Appendix B provides details about our experimental setup.
- Appendix C provides additional experiments and ablation studies.
- Appendix D provides details about our main theoretical results.

## A EXTENDED BACKGROUND

### A.1 LEAST SQUARES

Least squares, $\boldsymbol{Ax} = \boldsymbol{b}$, is well-understood theoretically, and we know of simple numerical algorithms for solving least squares to high precision (Weisberg, 2005; Boyd & Vandenberghe, 2004). We focus on two algorithms: gradient descent and Newton's method.

**Gradient descent**    Given a guess for $\boldsymbol{x}^*$, we minimize the least squares loss

$$\mathcal{L}(\boldsymbol{x}) = \frac{1}{2}\sum_{i=1}^{N}(\boldsymbol{a}_i^T\boldsymbol{x} - b_i)^2 \tag{7}$$

via gradient descent on $\boldsymbol{x}$:

$$\nabla_{\boldsymbol{x}}\mathcal{L}_N = \sum_{i=1}^{N}(\boldsymbol{x}^T\boldsymbol{a}_i - b_i)\boldsymbol{a}_i \tag{8}$$

$$\boldsymbol{x}_{t+1} = \boldsymbol{x}_t - \eta\nabla\mathcal{L}_N(\boldsymbol{x}_t) \tag{9}$$

**Ordinary Least Squares and Newton's method**    In the noiseless, full determined regime, the Bayes-optimal estimator is ordinary least squares (OLS) (Weisberg, 2005):

$$\boldsymbol{x}^{OLS} = (\boldsymbol{A}^T\boldsymbol{A})^{-1}\boldsymbol{A}^T\boldsymbol{b}, \tag{10}$$

where

$$\boldsymbol{A} = \begin{pmatrix} \leftarrow \boldsymbol{a}_1 \rightarrow \\ \vdots \\ \leftarrow \boldsymbol{a}_N \rightarrow \end{pmatrix}, \quad \boldsymbol{b} = \begin{pmatrix} b_1 \\ \vdots \\ b_N \end{pmatrix} \tag{11}$$

Note that this estimator requires a matrix inverse, which is expensive to compute exactly. An alternative is to use Newton's method to approximate the matrix inverse term (Schulz, 1933). To estimate $(\boldsymbol{A}^T\boldsymbol{A})^{-1}$, we can perform the following iterative algorithm:

$$\boldsymbol{M}_{t+1} = \boldsymbol{M}_t(2\boldsymbol{I} - (\boldsymbol{A}^T\boldsymbol{A})\boldsymbol{M}_t) \tag{12}$$

where $\boldsymbol{M}_t$ converges to $(\boldsymbol{A}^T\boldsymbol{A})^{-1}$.

### A.2 RELATED WORK

In this section, we detail prior work on in-context learning, Transformer expressivity, gated convolutional architectures, and algorithm learning.

**In-context learning.**    The capability of Transformers to perform in-context learning on language and pattern matching tasks has been well-documented (Brown et al., 2020; Dasgupta et al., 2022; Wei et al., 2022). More recently, a flurry of work has investigated in-context learning for regression-style tasks. Garg et al. (2022) first formulated the mathematical framework to analyze the estimators Transformers implement in-context, focusing on linear regression and other least squares problems. A number of works further observed empirically that Transformers seem to approximate Bayes-optimal estimators on distributional problems. For example, based on the task distribution, the performance of

in-context Transformers mimics optimally-tuned LASSO on sparse linear regression, ridge regression on noisy dense linear regression, and Bayes-optimal priors for task mixtures (Akyürek et al., 2024; Raventós et al., 2024; Yadlowsky et al., 2023; Ahuja et al., 2023; Bai et al., 2024). Beyond standard least squares problems, other works have investigated the ability of Transformers to in-context solve broader problems of scientific interest like differential equations (Yang et al., 2023b; Chen et al., 2024; Liu et al., 2023a).

Towards explaining these observations, recent works have focused on understanding the expressivity and optimziation landscapes of Transformer variants (typically non-causal linear attention) on linear regression. Linear attention has been shown to be expressive enough to implement numerical algorithms for solving linear regression, including gradient descent (Akyürek et al., 2022; Von Oswald et al., 2023) and Newton's method (Fu et al., 2023; Giannou et al., 2024). Recent work (Ahn et al., 2024; Mahankali et al., 2023; Zhang et al., 2023b) has also begun to investigate the optimization dynamics for linear attention on least squares. Finally, we highlight that recent work (Bai et al., 2024; Huang et al., 2023; Collins et al., 2024; Cheng et al., 2024) makes progress on theoretically understanding non-linear attention, e.g. with softmax or ReLU activations.

Unlike prior work, we investigate the capabilities of standard Transformers, focusing on exploring their capability to perform *high-precision* optimization algorithms. Noting a gap between empirical performance and theoretical claims regarding in-context least squares as gradient descent, we further investigate alternative architectures to softmax attention.

**Expressivity and approximation ability of Transformers.** Although Transformers were initially designed for discrete tasks like language modeling, recent works have investigated the ability of the Transformer architecture to express general *continuous-valued* sequence-to-sequence maps. We briefly mention three classes of prior work:

- **Constructive arguments.** We highlight Giannou et al. (2023), which proposes a looped-Transformer weight construction that implements a basic mathematical instruction set. Using compositions of these instructions, the authors demonstrate that Transformers are expressive enough to implement numerical algorithms, including matrix inversion and SGD on linear models.
- **Universal approximation results.** Several works, such as Yun et al. (2020a;b), provide bounds on the number of parameters and layers required to approximate smooth sequence-to-sequence functions to arbitrary precision using Transformers. However, these results typically require parameters to scale exponentially with respect to problem size, which quickly becomes impractical in practice.
- **Complexity theory results.** Recent works (Chiang et al., 2023; Merrill & Sabharwal, 2023; 2024) prove that *log-precision* Transformers lie in $TC^0$, a limited complexity class of circuits.

**Gated convolutions.** Gated convolutional models are a class of architectures that serve as an efficient alternative to attention. These models, consisting of gating (element-wise multiplication) and long convolutions (filter size equal to sequence length), stem from earlier work (Gu et al., 2021) inspired by the signal processing literature. In this work we focus on the BASECONV model from Arora et al. (2023), but a recent surge of interest in efficient attention replacements has led to a flood of gated convolutional architectures (Poli et al., 2023; Peng et al., 2023; Gu & Dao, 2023).

Recent architectural innovations within the class of gated convolutional models have been largely motivated by language modeling tasks (Fu et al., 2022; Arora et al., 2023). Unlike these prior works, which focus on matching attention's performance on *discrete* tasks, we observe that the connection between gated convolutions and arithmetic circuits implies they are able to exactly express a range of important numerical algorithms for *continuous-valued* tasks. We further investigate their ability to learn these algorithms in-context.

**Algorithm learning.** We mention two lines of work related to learning algorithms using ML:

- **Grokking.** Several works (Power et al., 2022; Nanda et al., 2023; Lee et al., 2024) have observed the ability of Transformers to learn to perfectly perform small discrete algorithmic tasks, e.g. modular arithmetic.

- **Neural Algorithmic Reasoning.** Recent work (Rodionov & Prokhorenkova, 2023; 2024) investigates the ability of graph neural networks to learn fundamental algorithms like breadth-first search (Veličković et al., 2022).

Crucially, we note that these previous works focus on learning *discrete* algorithmic tasks, which Transformers excel at. As far as we know, we are the first to investigate whether Transformers are able to learn *numerical* algorithms, which rely on addressing key challenges with high-precision floating-point arithmetic.

**Precision and scientific ML.** The importance and difficulty of high-precision ML for scientific settings is well-established: although the scientific ML community has made exciting progress in recent years, numerical methods are still known to outperform existing ML methods in precision even on simple PDE benchmarks (McGreivy & Hakim, 2024). Despite this, we are aware of only a few works which directly focus on investigating high precision for ML. We highlight (Michaud et al., 2023; Wang & Lai, 2023), which focus on small MLPs for regression tasks and propose alternate training recipes.

As far as we are aware, we are the first to investigate and isolate effects of model architectures and optimizers on precision in a controlled setting: in-context least squares. We find that typical training recipes for sequence models (e.g. softmax attention, Adam, and standard LR schedulers) encounter surprising precision barriers when applied to numerical tasks.

## B   EXPERIMENTAL SETUP

Here, we provide additional details about our experimental setup.

### B.1   MODEL ARCHITECTURE

We base our Transformer and BASECONV models off the GPT2 family (Radford et al., 2019). Unless otherwise specified, we use the following default settings for Transformers:

| Config | Setting |
|---|---|
| Embedding size | 64 |
| Number of layers | 12 |
| Number of heads | 8 |
| MLPs | True |
| MLP hidden size | $4\times$ embedding size |
| MLP activation | ReLU |
| LayerNorms | True |
| Input dim | 5 |
| Sequence length | 20 |

Table 1: Standard Transformer architecture details.

and the following settings for BASECONVs:

| Config | Setting |
|---|---|
| Embedding size | 64 |
| Number of layers | 3 |
| MLPs | False |
| LayerNorms | False |
| Input dim | 5 |
| Sequence length | 20 |

Table 2: BASECONV architecture details.

Finally, we describe the settings we use for our linear attention experiment (Figure 2):

| Config | Setting |
|---|---|
| Embedding size | 256 |
| Number of layers | 3 |
| Number of heads | 16 |
| MLPs | False |
| LayerNorms | False |
| Input dim | 5 |
| Sequence length | 20 |

Table 3: Linear attention architecture details.

### B.2   OPTIMIZER

We describe two sets of optimizer settings we use throughout this work.

The first, representative of standard training procedures, is inspired by prior in-context learning setups (Garg et al., 2022; Von Oswald et al., 2023).

The second, our training recipe, is for our high-precision experiments, where we find a more aggressive learning rate scheduler is essential. Note we use the adaptive learning rate scheduler and EMA described in Section 5.

| Config | Setting |
|---|---|
| Batch size | 256 |
| Optimizer | Adam |
| Learning rate | $10^{-3}$ |
| Scheduler | StepLR |
| Training iterations | $10^6$ |
| Step rate | $10^4$ |
| Decay rate | 0.9 |

Table 4: Standard optimizer settings.

| Config | Setting |
|---|---|
| Batch size | 1024 |
| Optimizer | Adam |
| Learning rate | $10^{-2}$ |
| Scheduler | AdaptiveLR |
| Training iterations | $2.5 \times 10^6$ |
| Step rate | $3 \times 10^3$ |
| Decay rate | 0.9 |
| EMA decay | 0.98 |
| EMA lambda | 2 |

Table 5: High-precision training recipe settings for BASECONV.

Finally, we describe the optimization settings we used for high-precision linear attention, which we found needed a slightly different learning rate scheduler.

| Config | Setting |
|---|---|
| Batch size | 1024 |
| Optimizer | Adam |
| Learning rate | $10^{-2}$ |
| Scheduler | AdaptiveLR |
| Training iterations | $2.5 \times 10^6$ |
| Step rate | $3 \times 10^3$ |
| Decay rate | 0.9 |
| EMA decay | 0.98 |
| EMA lambda | 2 |

Table 6: High-precision training recipe settings for linear attention.

## B.3 TASKS

Each of our in-context learning tasks can be viewed as a sequence-to-sequence map

$$\mathcal{M} : \mathbb{R}^{N_{in} \times D_{in}} \to \mathbb{R}^{N_{out} \times D_{out}}$$

In this subsection, we provide details about task implementations, specifying the input/output formats for each of the synthetic tasks and in-context least squares variants we implement.

### B.3.1 IN-CONTEXT LEAST SQUARES.

We consider $\mathcal{M}_{LS} : \mathbb{R}^{N \times (D+1)} \to \mathbb{R}^D$, where as above the inputs are formatted as

$$\boldsymbol{u}_{in} := \begin{bmatrix} \boldsymbol{a}_1 & \dots & \boldsymbol{a}_N \\ b_1 & \dots & b_N \end{bmatrix}$$

and the expected output is

$$T_\theta(\boldsymbol{u}_{in})[:\text{-}1, \text{-}1:] := \boldsymbol{x}.$$

### B.3.2 PRIMITIVES.

For each of the following linear algebra primitives, we increase the task size, setting $D = 20$ and $N = 40$.

- READ is defined as $\mathcal{M}_{Read} : \mathbb{R}^{N \times D} \to \mathbb{R}^{N \times D}$, where the inputs are formatted as

$$\boldsymbol{u}_{in} \in \mathbb{R}^{N \times D} := \begin{bmatrix} \boldsymbol{x}_1 & \dots & \boldsymbol{x}_N \end{bmatrix}$$

  and the expected outputs are $T_\theta(\boldsymbol{u}_{in}) \in \mathbb{R}^{N \times D}$ such that

$$T_\theta(\boldsymbol{u}_{in})[k, :] := \begin{cases} \boldsymbol{u}_{in}[i, :] & k = j \\ \boldsymbol{u}_{in}[k, :] & k \neq j \end{cases}$$

  for task parameters $i \neq j \in [N]$.
- LINEAR is defined as $\mathcal{M}_{Linear} : \mathbb{R}^{N \times D} \to \mathbb{R}^{N \times 1}$, where the inputs are formatted as

$$\boldsymbol{u}_{in} \in \mathbb{R}^{N \times D} := \begin{bmatrix} \boldsymbol{x}_1 & \dots & \boldsymbol{x}_N \end{bmatrix}$$

  and the expected outputs are

$$T_\theta(\boldsymbol{u}_{in}) := \begin{bmatrix} \boldsymbol{x}_1^T \boldsymbol{h} & \dots & \boldsymbol{x}_N^T \boldsymbol{h} \end{bmatrix}$$

  where $\boldsymbol{h} \in \mathbb{R}^D$ is a task parameter.
- MULTIPLY is defined as $\mathcal{M}_{Multiply} : \mathbb{R}^{N \times D} \to \mathbb{R}^{N \times D/2}$, where the inputs are formatted as

$$\boldsymbol{u}_{in} \in \mathbb{R}^{N \times D} := \begin{bmatrix} \boldsymbol{x}_1 & \dots & \boldsymbol{x}_N \end{bmatrix}$$

  and the expected outputs are

$$T_\theta(\boldsymbol{u}_{in}) := \left( \boldsymbol{x}_1[:, :D/2] \odot \boldsymbol{x}_1[:, D/2:] \quad \dots \quad \boldsymbol{x}_N[:, :D/2] \odot \boldsymbol{x}_N[:, D/2:] \right).$$

### B.3.3 EXPLICIT GRADIENT UPDATES.

In Section 5, we investigate a simple training setting, in which the model is explicitly trained to predict the gradient of the least squares loss. We proceed to define the task $\mathcal{M}_{gradient}$ : $\mathbb{R}^{(N+1) \times (D^2 + 2D + 1)} \to \mathbb{R}^D$.

The inputs are formatted as

$$\boldsymbol{u}_{in} := \begin{bmatrix} \boldsymbol{a}_1 & \dots & \boldsymbol{a}_N & \boldsymbol{x}_0 \\ b_1 & \dots & b_N & 0 \end{bmatrix}.$$

The expected outputs are

$$T_\theta(\boldsymbol{u}_{in})[\text{-}1:, :\text{D}] := \nabla_{\boldsymbol{w}} \mathcal{L}(\boldsymbol{x}_0).$$

### B.3.4  $k$-TH GRADIENT DESCENT ITERATE.

Finally, toward end-to-end least squares, we investigate a series of increasingly end-to-end tasks in which the model is explicitly trained to predict the $k$-th gradient descent iterate. We proceed to define the task $\mathcal{M}_{iter}^k : \mathbb{R}^{(N+1)\times(D^2+2D+1)} \to \mathbb{R}^D$.

The inputs are formatted as

$$\boldsymbol{u}_{in} := \begin{bmatrix} \boldsymbol{a}_1 & \dots & \boldsymbol{a}_N & \boldsymbol{x}_0 \\ b_1 & \dots & b_N & 0 \end{bmatrix}.$$

The expected outputs are

$$T_\theta(\boldsymbol{u}_{in})[\text{-1:}, :\text{D}] := \boldsymbol{x}_k.$$

### B.4  DATA GENERATION

At each training step, we produce a random training prompt $\boldsymbol{u}_{in}$ by sampling each variable randomly: from the isotropic Gaussian distribution $N(\boldsymbol{0}, \boldsymbol{I})$ for continuous-valued parameters, and from the uniform distribution for discrete parameters. Concretely:

- For the in-context linear regression tasks, input vectors $\boldsymbol{x}_1, \dots, \boldsymbol{x}_N$ are sampled from $N(\boldsymbol{0}^D, \boldsymbol{I}^D)$, and the unknown linear function is determined by $\boldsymbol{w}^*$, also drawn from $N(\boldsymbol{0}^D, \boldsymbol{I}^D)$.
- For the synthetic tasks READ, LINEAR, MULTIPLY (Section 3.3), *each column* of the inputs $\boldsymbol{u}_{in} \in \mathbb{R}^{N\times D}$ is sampled from the isotropic Gaussian distribution $N(\boldsymbol{0}^D, \boldsymbol{I}^D)$. The tasks READ and LINEAR require specifying additional parameters as follows:
    - For READ, at each iteration, $i \neq j \in [N]$ are sampled uniformly.
    - For LINEAR, at each iteration, the affine transformation $\boldsymbol{h}$ is sampled from $N(\boldsymbol{0}^D, 3\boldsymbol{I}^D)$.
- For the explicit gradient task and the $k$-th gradient descent iterate task, the random initialization $\boldsymbol{w}_0$ is also drawn from $N(\boldsymbol{0}^D, \boldsymbol{I}^D)$.

The model is trained to minimize mean squared error over the distribution of prompts.

# C ADDITIONAL EXPERIMENTAL RESULTS

## C.1 ABLATIONS: LINEAR ALGEBRA PRIMITIVES

In Figure 6, we train Transformers and BASECONVS, *with* MLPs, with and without LayerNorms (LN), on the READ, LINEAR, and MULTIPLY primitives from Section B.3.2. We vary the model depth $L \in \{1, 2, 4, 8\}$ and investigate how precision scales with number of layers. In these experiments, we use a standard exponentially decaying LR schedule for Adam.

We show that Transformers and BASECONVS both achieve high precision ($< O(10^{-9})$) on the READ and LINEAR tasks. However, the Transformers struggle to implement MULTIPLY to high precision, and performance scales poorly with model depth. We observe that BASECONV without LayerNorm generally performs the best across all three primitives, consistently outperforming BASECONV with LayerNorm by 2-4 orders of magnitude.

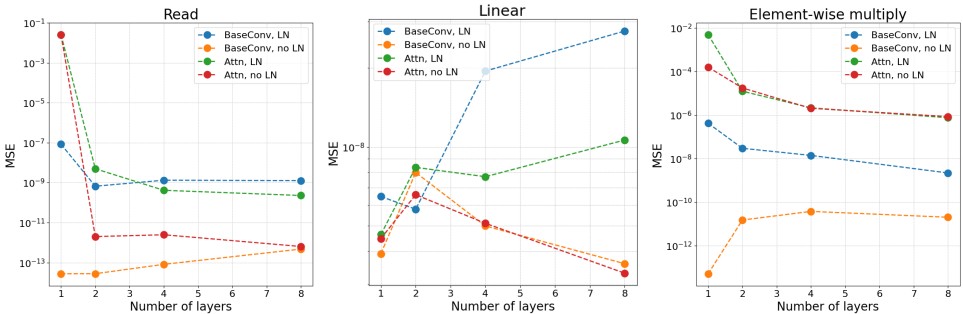

Figure 6: Attention vs. BASECONV, with and without LayerNorms, on synthetic tasks. Precision consistently scales better with depth for BASECONV models than for Transformers. While both models solve READ and LINEAR tasks to at least $10^{-8}$ MSE, the precision of Transformers scales poorly for the MULTIPLY task.

Focusing on 2-layer Transformers and the MULTIPLY task, we additionally find that precision scales poorly with multiple scaling axes, including hidden dimension, number of heads, and MLP upscaling factor (Figure 7).

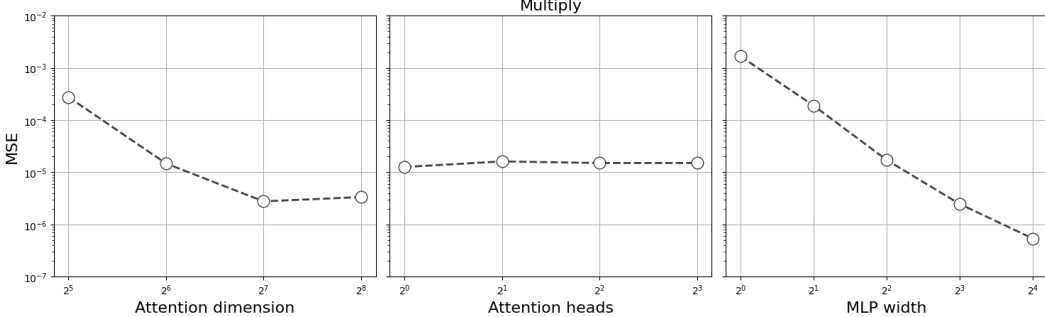

Figure 7: Precision of (2-layer) Transformers on MULTIPLY task scales poorly with attention dimension (left), number of heads (middle), and MLP width (right, where MLP hidden dimension = width $\times$ attention dimension).

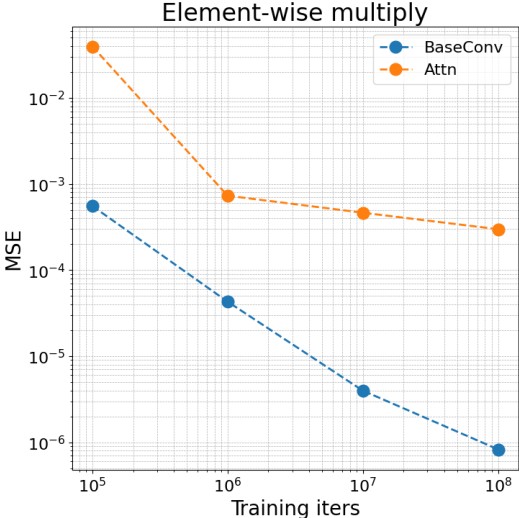

Figure 8: Scaling number of training iterations for 1-layer Transformer vs. BASECONV on the MULTIPLY task. Both models improve precision by 2-3 orders of magnitude as training duration increases by 3 orders of magnitude.

Finally, we investigate the effect of training duration on precision. In Figure 8, we train 1-layer Transformers and BASECONVs, with MLPs and LayerNorms, on the MULTIPLY primitive and vary the number of iterations for which the model is trained. Recall that since new data is sampled at each iteration, we also effectively scale the dataset size proportionally. To keep the learning rates consistent across runs, we scale back the scheduler step size accordingly:

$$num\_iters \in \{10^5, 10^6, 10^7, 10^8\}$$
$$step\_size \in \{10^3, 10^4, 10^5, 10^6\}$$

We observe a power law, particularly clearly for BASECONV, as we scale from $10^5$ to $10^8$ iterations. Both models achieve a 2-3 order of magnitude improvement in precision, but this requires also increasing training duration by 3 orders of magnitude.

## C.2 ABLATIONS: HIGH-PRECISION OPTIMIZATION

In Figure 9, we try directly training on the end-to-end least squares task, simply replacing softmax attention with BASECONV in the standard Transformer architecture. We find we are unable to reach high precision using this training procedure.

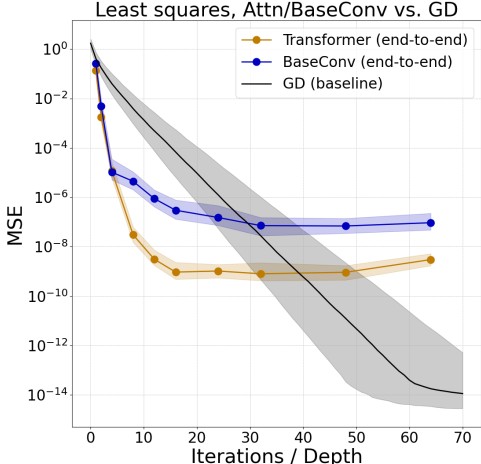

Figure 9: Replacing softmax attention with BASECONV in the standard Transformer architecture and training end-to-end on least squares is not enough to achieve high-precision solutions. BASEC-ONV models trained end-to-end perform as badly as Transformers at small scale, and our largest models perform $100\times$ *worse* than parameter-matched Transformers.

In Figure 10, we ablate the effects of constant and exponentially decaying LR schedulers with Adam (cutting off training after $10^6$ iterations). We find that neither are able to efficiently train to machine precision on the explicit gradients task. For exponentially decaying LR schedule, we find that the LR steprate is a crucial parameter: on the explicit gradient task, a difference of $10,000\times$ between precision saturation thresholds using $1 \times 10^3$ vs $3 \times 10^3$ for example.

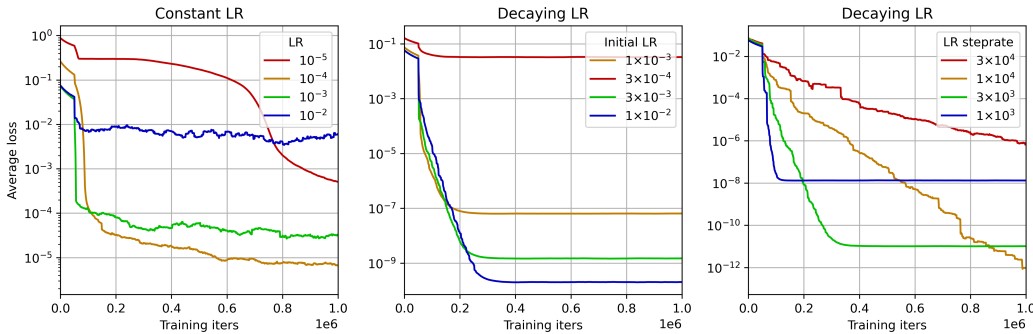

Figure 10: Training with Adam on the explicit gradient task, we ablate LR for constant scheduler (left), initial LR (middle) and LR steprate (right) for decaying scheduler.

In Figure 11, we ablate the effect of applying an EMA over the update vectors from the Adam optimizer. Empirically, we find that this boosts the final MSE by as much as $100,000\times$ on the explicit gradient task.

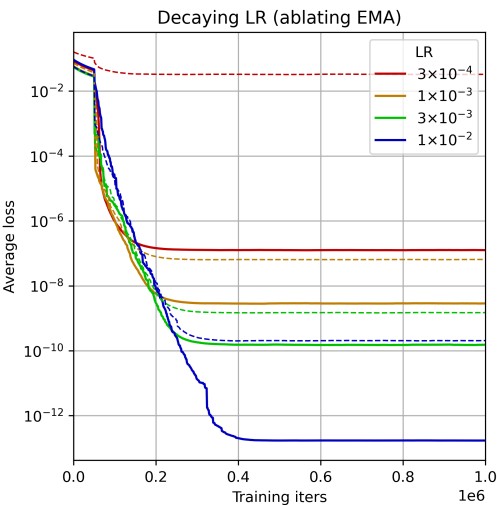

Figure 11: Training on the explicit gradient task, applying EMA over Adam's update vectors consistently boosts final MSE, by up to 5 orders of magnitude.

In Figure 12, we ablate the effect of restoring the MLPs and LayerNorms to BASECONV models. Surprisingly, we find that even these architectural components worsen the model's precision: on the explicit gradient task, by a factor of up to $1,000,000\times$ MSE. We note that due to training instability with the BASECONV+MLP model, we used a less aggressive LR schedule with initial LR $10^{-3}$ and LR steprate $10^4$.

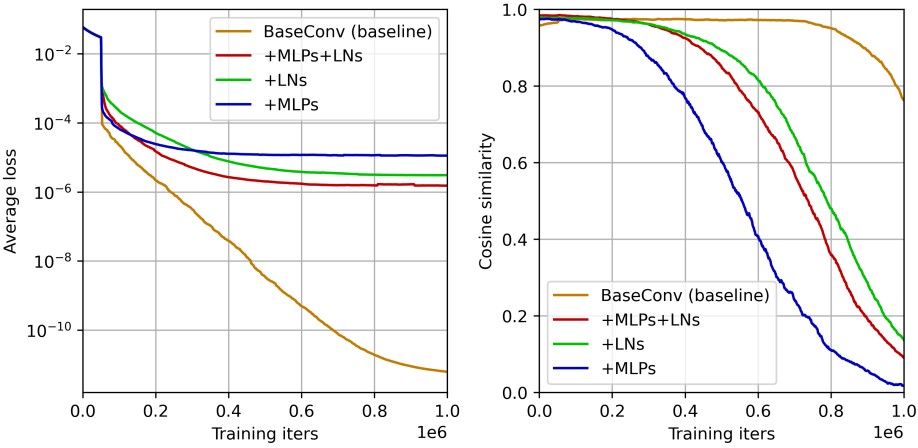

Figure 12: Training on the explicit gradient task, adding MLPs and LNs consistently bottlenecks precision: here, by up to 6 orders of magnitude.

In Figure 13, we evaluate 3-layer BASECONV and linear attention models trained on the explicit gradient task. As in Section 5.2, we apply them iteratively until convergence. We then evaluate on out-of-distribution regression targets, as in Section 3.2.

We surprisingly find that linear attention demonstrates poor numerical generality, despite training to near machine precision on the training distribution. Beyond $\sigma = 4$, the iterations of linear attention diverge.

This result suggests that although different polynomial architectures may equally be able to *express* algorithms, they may *learn* solutions that exhibit vastly different numerical properties.

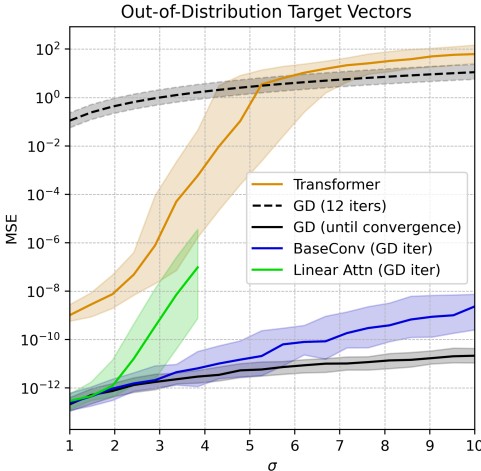

Figure 13: While BASECONV demonstrates improved numerical generality compared to end-to-end trained Transformers, the generalization gap for linear attention is as bad as the Transformer.

### C.3 $k$-TH ITERATE GD

In this section, we investigate how well our proposed techniques can learn the $k$-th GD iterate tasks as defined in Section 5:

$$\{(\boldsymbol{a}_1, b_1), \ldots, (\boldsymbol{a}_N, b_N), \boldsymbol{x}_0\} \to \boldsymbol{x}_k, \text{ where } \boldsymbol{x}_{i+1} = \boldsymbol{x}_i - \eta \nabla \mathcal{L}(\boldsymbol{x}_i), \ i \in [k-1]. \tag{13}$$

Recall that $k = 1$ is equivalent to the explicit gradient task, while taking $k \to \infty$ is equivalent to the standard in-context least squares task. Here, we are interested in understanding how well our techniques extend to larger $k$, towards learning end-to-end least squares. See Appendix B for a more detailed description of the training setup.

Our theoretical results in Section 4 imply that a $k + 2$-layer BASECONV is expressive enough to solve the $k$-th iterate task to machine precision. Thus we train $k + 2$-layer BASECONV models on the $k$-th iterate task for $k \geq 1$ using our training recipe.

| $k$ | 1 | 2 | 3 | 4 |
|---|---|---|---|---|
| MSE | $5.0 \times 10^{-13}$ | $2.5 \times 10^{-11}$ | $2.5 \times 10^{-11}$ | $3.1 \times 10^{-10}$ |

Table 7: We can learn up to $4$ iterations of GD at once with our current training techniques. Model stability becomes a bottleneck with harder tasks.

Training on the $k$-iter GD task, we find our training recipe scales to $k = 4$ before training instability occurs. Adding LayerNorms, we are able to train deeper models, but we find MSE worsens by at least $1,000\times$ for small $k$: see Figure 14.

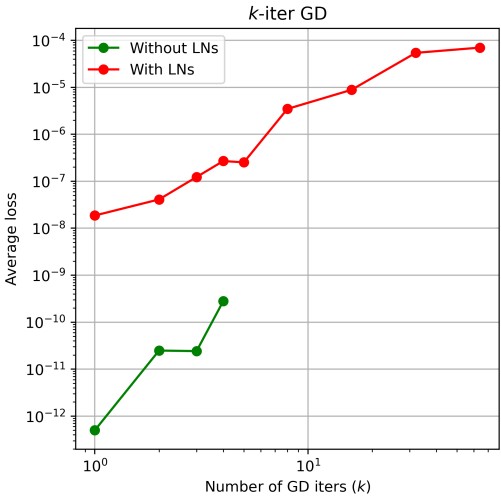

Figure 14: BASECONV with LayerNorms are able to stably scale to deeper models, but LayerNorms present a precision bottleneck: even on small $k$, MSE degrades by over $1,000\times$.

### C.4 IN-CONTEXT ODE SOLVING

In this section, we demonstrate the generality of our insights on the more practical setting of in-context ODE solving. We note that solving differential equations in-context with Transformers is a framework that has been explored in recent papers (Yang et al., 2023a; Herde et al., 2024; Liu et al., 2023a), and thus represents a natural first step towards extending our techniques to realistic scientific ML problems.

**Experimental setup.**    We follow the setup from Liu et al. (2023a):

- We train on a distribution of 1D ODEs over $t \in [-1, 1]$, defined by

$$u'(t) = \alpha_1 c(t) + \alpha_2 u(t) + \alpha_3. \tag{14}$$

  For each operator, we provide 25 in-context examples of forcing functions, initial conditions, and their corresponding solution values at a fixed time $t_{query} \in [-1, 1]$. We then give the model a query forcing function and initial condition, and the goal is to predict the corresponding solution at $t_{query}$.
- We sample our parameters $\alpha_1 \sim \text{Unif}([0.5, 1.5])$, $\alpha_2 \sim \text{Unif}([-1, 1])$, $\alpha_3 \sim \text{Unif}([-1, 1])$.
- Initial conditions are sampled from $u(0) \sim \text{Unif}([-1, 1])$.
- Forcing functions $c(t)$ are sampled from a Gaussian process with RBF kernel $K(x, x') = \exp\left(-\frac{(x-x')^2}{2\ell^2}\right)$, with length-scale parameter $\ell = 1$. We sample each forcing function on 21 equispaced points over $[-1, 1]$.
- ODEs are solved pseudospectrally on $N = 41$ nodes: we find this is sufficient for machine-precision solutions with FLOAT32 datatype.

We find that our observations from least squares transfer to the setting of in-context ODEs:

**Transformers struggle to learn precise solutions.**    We find that a 12-layer, 9M parameter Transformer model only achieves $\approx 10^{-4}$ MSE, almost $10^{10} \times$ worse than the threshold FLOAT32 machine epsilon implies. Furthermore, as with least squares, we observe precision saturation with model size. In Figure 15, we find that scaling the depth of the model by up to $2 \times$ does not improve precision. We further note that precision saturation already seems to occur with 4-layer Transformers. We hypothesize that the depth at which precision saturation begins is dependent on the task difficulty.

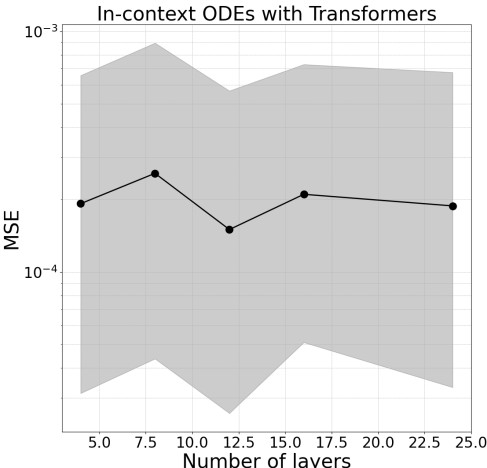

Figure 15: Transformers fail to learn precise algorithms for solving ODEs in-context. As with least squares, precision saturates with deeper models: in our experiments, we observe no significant performance boost between 4-layer and 24-layer Transformers.

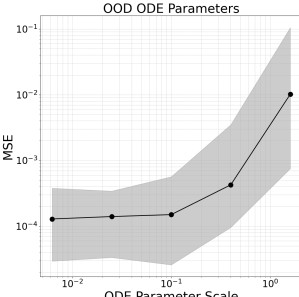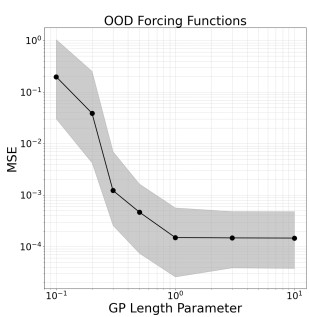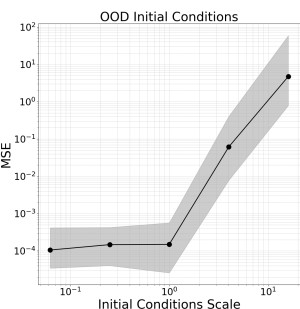

Figure 16: Transformers fail to learn numerically general solutions: performance is brittle to out-of-distribution ODE parameters (left), forcing function smoothness (middle), and initial condition distribution (right).

**Transformers exhibit brittle generalization.** We observe that Transformers are not robust to changes to the distributions of ODE parameters, forcing functions, and initial conditions. We describe our experimental setup below, mirroring Liu et al. (2023a):

- Out-of-distribution ODE parameters. We parameterize out-of-distribution ODEs via a scale parameter $\sigma_{op}$, where $\alpha_1 \sim \text{Unif}([1-\frac{1}{2}\sigma_{op}, 1+\frac{1}{2}\sigma_{op}])$ and $\alpha_2, \alpha_3 \sim \text{Unif}([-\sigma_{op}, \sigma_{op}])$. As we increase $\sigma_{op}$, we sample from a wider distribution of ODE solution operators, including those with larger operator norms and worse-conditioned design matrices.

- Out-of-distribution forcing functions. We vary $\ell$, the length parameter of the Gaussian process from which we sample our forcing functions, which effectively controls their smoothness.

- Out-of-distribution initial conditions. We sample out-of-distribution initial conditions as $u(0) \sim \text{Unif}([-\sigma_{IC}, \sigma_{IC}])$. As we vary $\sigma_{IC}$, we widen the distribution of the solution values at $t = 0$, which increases the overall magnitudes of the solutions.

We note that in all out-of-distribution experiments, the Transformer's MSE explodes to near $O(1)$: refer to Figure 16.

**Our proposed techniques obtain precise and general solutions.** Liu et al. (2023a) shows that in-context ODEs can be reduced to solving least squares problems. Thus, we train a 3-layer BASEC-ONV architecture on the explicit gradient task for the equivalent least squares problem, and apply our model iteratively, as in Section 5. We compare the performance of our iterative model with end-to-end Transformers, least squares solvers, and standard gradient descent applied to the equivalent least squares problem.

We note that our ODEs reduce to least squares problems that are ill-conditioned. In this set of experiments, we find the condition numbers of our design matrices are $O(10^8)$. Since the theoretical convergence rate of gradient descent on least squares depends inversely on the condition number (Boyd & Vandenberghe, 2004), we expect our iterative models and standard gradient descent will require orders of magnitude more iterations than in the least squares problems of Section 5. As such, we limit the number of iterations for our BASECONV model and standard gradient descent to $10,000$. Nonetheless, we find that our BASECONV model learns to high enough precision that we are able to maintain the stability of the iterative algorithm for up to $O(10^5)$ steps. In our experiments, we iteratively apply our BASECONV model until convergence to a fixed point and report final MSEs.

We find that BASECONV learns a precise and general algorithm for in-context ODEs:

- **Precision.** In Figure 17, we show that our BASECONV model, applied iteratively, converges to about $10^{-10}$ MSE, $1,000,000\times$ higher precision than our best Transformers.

- **Generality.** In Figure 18, we find that our BASECONV model exhibits more robust generalization than the Transformer model: in all the out-of-distribution settings we test, our BASECONV model achieves higher precision than Transformers in-distribution. Like above,

we evaluate on out-of-distribution ODE parameters, forcing functions, and initial conditions. In particular, we note that the performance of our BASECONV model almost exactly matches proper gradient descent, even in out-of-distribution settings. Additionally, we find the generalization behavior of our BASECONV model matches the generalization of a proper least squares solver with preconditioning, except for out-of-distribution initial conditions, where we note that the iterative procedure suffers from slow convergence and times out at 10, 000 iterations.

We believe these preliminary results show the promise of our techniques towards learning numerical algorithms for more complex tasks, such as solving PDEs, directly from data.

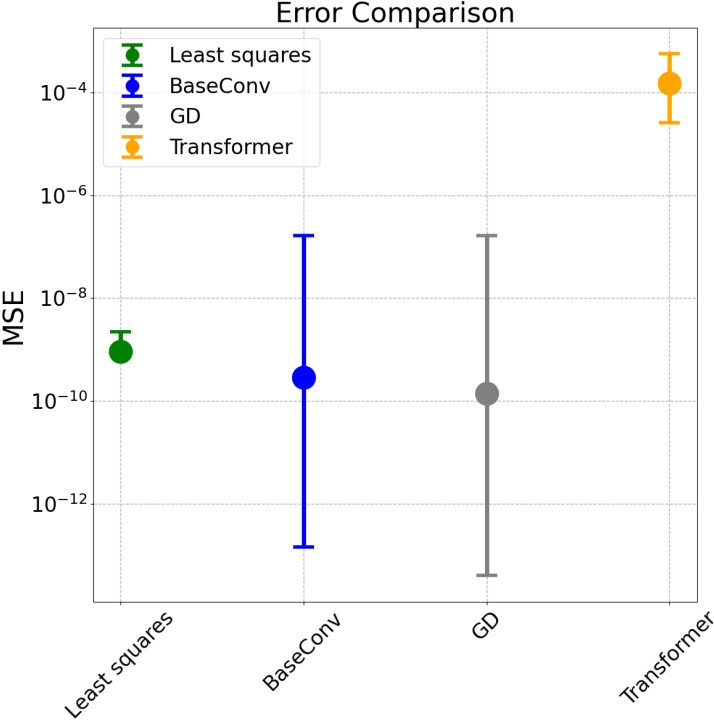

Figure 17: In-distribution error comparison between Transformer, BASECONV, gradient descent, and least squares.

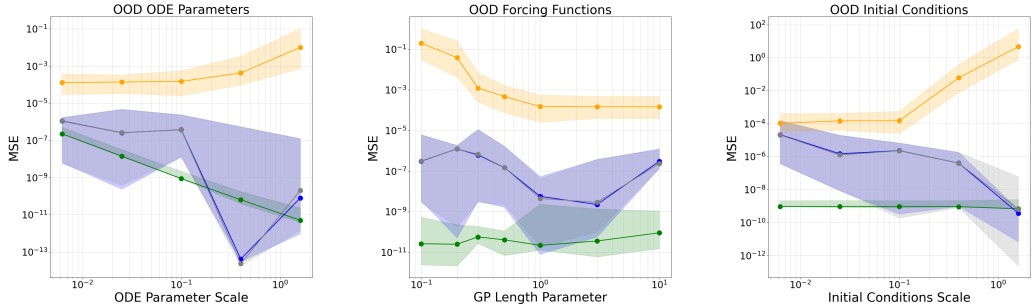

Figure 18: Out-of-distribution error comparison between Transformer (orange), BASECONV (blue), gradient descent (gray), and least squares (green): we evaluate out-of-distribution ODE parameters (left), forcing function smoothness (middle), and initial condition distribution (right). BASECONV learns a numerically general algorithm that closely matches proper gradient descent and least squares.

# D    THEORETICAL RESULTS

This section is organized as follows:

- We detail notation and definitions in Appendix D.1.

- In Appendix D.2, we include theoretical results regarding the primitives from Section 3.3: expressivity results with BASECONV and attention, and iterative algorithms as compositions of primitives.

- In Appendix D.3, we discuss upper and lower bounds for implementing gradient descent on least squares using BASECONV, supplementing Section 4.1.

- In Appendix D.4, we provide theoretical details regarding the universal function approximation properties of BASECONV.

- Finally, in Appendix D.5, we provide technical details about the claims from Section 4.1 that BASECONV can perfectly recover SQUARE and LINEAR.

## D.1    NOTATION

We heavily borrow notation from Appendix H of Arora et al. (2023), which we recollect below. We denote the all 1 row vector of size $k$, given by $[1 \quad 1 \quad \ldots \quad 1 \quad 1]$, and the all 0 row vector of size $k$, given by $[0 \quad 0 \quad \ldots \quad 0 \quad 0]$, as $\mathbf{1}^k$ and $\mathbf{0}^k$, respectively. We also construe the standard basis vector $\mathbf{e}_i$ as a column vector in this appendix, and adhere to the following matrix indexing convention: $\mathbf{M}[i, j]$ is the entry in the $i$th row and the $j$th column, $\mathbf{M}[i, :] \in \mathbb{F}^{1 \times n}$ denotes the $i$th row, and $\mathbf{M}[:, j] \in \mathbb{F}^{m \times 1}$ denotes the $j$th column of $\mathbf{M} \in \mathbb{F}^{m \times n}$, where $\mathbb{F}$ is a field (the reader can assume that $\mathbb{F}$ is the field of real numbers i.e. $\mathbb{F} = \mathbb{R}$). We then use $\mathbf{1}^{m \times n}, \mathbf{0}^{m \times n} \in \mathbb{F}^{m \times n}$ to denote the matrix of all 1s and 0s, respectively. We note that some notation differs from those used in earlier sections.

Next, we denote the *Hadamard product* of vectors $\mathbf{u}, \mathbf{v} \in \mathbb{F}^n$ as $\mathbf{u} \odot \mathbf{v}$; the operation can be extended to matrices by applying the Hadamard product column-wise across the matrices. This is commonly referred to as *(element-wise) gating*. For vectors $\mathbf{u}, \mathbf{v} \in \mathbb{F}^n$, we also denote their *linear (or acyclic) convolution* as $\mathbf{u} * \mathbf{v}$ and *cyclic convolution* as $\mathbf{u} \circledast \mathbf{v}$.

**Polynomial Notation.**    Since convolution is equivalent to operations on polynomials, it is convenient to use them to discuss the inputs and outputs of gated convolution models. Let us define maps $\mathrm{poly} : \mathbb{F}^n \to \mathbb{F}[X]/(X^n)$ such that

$$\mathrm{poly}(\boldsymbol{u}) = \sum_{i=0}^{n-1} \boldsymbol{u}[i] X^i.$$

This allows us to map between vectors and polynomial. Accordingly, we also define $\mathrm{coeff} : \mathbb{F}[X]/(X^{n+1}) \to \mathbb{F}^n$ as the map converting polynomials back to vectors: $\mathrm{coeff}(\boldsymbol{u}(X)) = \boldsymbol{u}$ with $\boldsymbol{u}[i]$ defined as the coefficient in $\boldsymbol{u}(X)$ at degree $i$.

These operations allow us to interpret the convolution of vectors in terms of polynomial multiplication (Heideman & Burrus, 1988). More specifically, we have

$$\boldsymbol{u} * \boldsymbol{v} = \mathrm{coeff} \left( \boldsymbol{u}(X) \cdot \boldsymbol{v}(X) \mod X^n \right)$$

The following notation for a polynomial will be used in this section:

**Definition D.1.** A polynomial $P(X)$ with degree $d$ and some coefficients $\mathbf{c} \in \mathbb{R}^{d+1}$ is defined as,

$$P(X) = \sum_{i=0}^{d} c_i X^i.$$

Further, the degree of $P(X)$ will be denoted as $\deg(P)$.

**Function Approximation.**   In this part, we collect notation and known results about function approximation. We will reference some definitions from Pleśniak (2009); Petersdorff (2015).

The following notation is to denote the $k$th derivative of a function:

**Definition D.2.** For some function $f : \mathbb{R} \to \mathbb{R}$, $f^{(k)} := \frac{d^k}{dx^k} f(x)$ is the $k$th derivative of $f$.

Define a set of univariate functions with a notion of continuity:

**Definition D.3.** We denote $C^k[a, b]$ for $k = 1, 2, \ldots$ the space of univariate functions $f : [a, b] \to \mathbb{R}$, which have derivatives $f^{(1)}, \ldots, f^{(k)}$ that are continuous on the closed interval $[a, b]$.

Next we define a set of multivariate functions with a notion of continuity:

**Definition D.4.** A function $f : [a, b]^n \to \mathbb{R}$ is in $C^k[a, b]^n$ for $k = 1, 2, \ldots$ if all partial derivatives

$$\frac{\partial^\alpha}{\partial x_1^{\alpha_1} \partial x_2^{\alpha_2} \cdots \partial x_n^{\alpha_n}} f(y_1, y_2, \ldots, y_n)$$

exist and are continuous, for every $\alpha_1, \alpha_2, \ldots, \alpha_n \in \mathbb{Z}_{\geq 0}$, such that $\alpha_1 + \alpha_2 + \cdots + \alpha_n \leq k$ and every $(y_1, \ldots y_n) \in [a, b]^n$.

We use the following notation for the set of all univariate polynomials:

**Definition D.5.** For any integer $d \geq 0$, we define

$$\mathcal{P}_d(X) = \{c_0 + c_1 X + \cdots + c_d X^d | c_k \in \mathbb{R}\}.$$

In other words, $P_d(X)$ is the space of univariate polynomials of degree less or equal to $d$.

We use the following notation for multivariate polynomials:

**Definition D.6.** For any integers $n, d \geq 0$ , we define

$$\mathcal{P}_d^n(X_1, \ldots, X_n) = \left\{ \sum_{\boldsymbol{\alpha} = (\alpha_1, \ldots, \alpha_n) \in \mathbb{Z}_{\geq 0}^n} c_\alpha X_1^{\alpha_1} X_2^{\alpha_2} \cdots X_n^{\alpha_n} \middle| c_\alpha \in \mathbb{R}, \sum_{i=0}^n \alpha_i \leq d \right\}.$$

Then $\mathcal{P}_d^n(X_1, \ldots X_n)$ is the space of $n$-variate polynomials of degree less or equal to $d$.

The following notation is for considering the pointwise absolute value of a matrix:

**Definition D.7.** For $M \in \mathbb{R}^{N \times D}$ define,

$$\|\boldsymbol{M}\|_\infty = \max_{\substack{0 \leq i < N \\ 0 \leq j < D}} |M[i, j]| \,.$$

Now lets define the corresponding $\infty-$norm for functions:

**Definition D.8.** For $g : [-1, 1]^{N \times D} \to \mathbb{R}^{N \times D}$, define

$$\|g\|_\infty = \max_{\mathbf{x} \in [-1,1]^{N \times D}} |g(\mathbf{x})| \,.$$

We will use the following version of Jackson's theorem for univariate inputs:

**Theorem D.9** (D. Jackson (1930) Jackson's Theorem for $C^k[-1, 1]$.). *Let $d, k$ be integers with $d + 1 \geq k \geq 0$ and $f \in C^k[-1, 1]$. Then*

$$\inf_{P \in \mathcal{P}_d} \|f - P\|_\infty \leq \left(\frac{\pi}{2}\right)^k \frac{1}{(d+1)d \cdots (d - k + 2)} \left\| f^{(k)} \right\|_\infty. \tag{15}$$

We will use the following version of Jackson's theorem for multivariate inputs:

**Theorem D.10** (Pleśniak (2009) Jackson's Theorem for $C^k[-1, 1]^n$). *Let $d, k$ be integers with $d + 1 \geq k \geq 0$ and $f \in C^k[-1, 1]^n$. Then*

$$\inf_{P \in \mathcal{P}_d^n} \|f - P\|_\infty \leq \frac{c_k}{d^k} \sum_{j=1}^n \left\| \frac{\partial^{k+1}}{\partial x_j^{k+1}} f(\mathbf{x}) \right\|_\infty \tag{16}$$

*where $c_k$ is a positive constant.*

We will use the following definition of univariate smooth functions:

**Definition D.11.** We call a $k$ times differentiable function $f : [-1, 1] \to \mathbb{R}$ to be $(k, L)$-smooth if $\left\| f^{(k)} \right\|_\infty \leq L$.

Next, we observe that given a univariate smooth function, there's a univariate bounded degree polynomial that approximates it to some error, $\epsilon$:

**Corollary D.12.** *For some $(k, L)$-smooth univariate function $f$ (as in [Definition D.11](#)), then there exists a polynomial $P_f(x)$ with*

$$\deg(P_f) \leq O\left( \sqrt[k]{\frac{L}{\epsilon}} \right) + k$$

*such that for all $x \in [-1, 1]$*

$$|f(x) - P_f(x)| \leq \epsilon.$$

*Proof.* We will be a bit more specific on an upper bound of $\deg(P_f)$. We pick:

$$\deg(P_f) = \left\lceil \frac{\pi}{2} \left( \frac{L}{\epsilon} \right)^{\frac{1}{k}} + k \right\rceil. \tag{17}$$

Let $d = \deg(P_f)$ where $P_f$ is the polynomial that achieves the left hand side of [Equation (15)](#). Then we have error at most

$$\left( \frac{\pi}{2} \right)^k \frac{1}{(d+1)d \cdots (d-k+2)} \left\| f^{(k)} \right\|_\infty.$$

Using the definition of a $(k, L)$-smooth univariate function in [Definition D.11](#) we get the error at most

$$\left( \frac{\pi}{2} \right)^k \frac{L}{(d+1)d \cdots (d-k+2)} \leq \left( \frac{\pi}{2} \right)^k \frac{L}{(d-k)^k}$$

where the inequality follows since each $d+1, d, \ldots, d-k+2 \geq (d-k)$.

Plugging in [Equation (17)](#) for $d$ we get the error is at most:

$$\left( \frac{\pi}{2} \right)^k \frac{L}{\left( \frac{\pi}{2} \right)^k \left( \sqrt[k]{\frac{L}{\epsilon}} \right)^k} = \epsilon,$$

as desired. $\qquad \square$

We will use the following definition of multivariate smooth functions that map to a single value:

**Definition D.13.** We call a $k$ times differentiable $f : [-1, 1]^n \to \mathbb{R}$ to be $(k, L)$-smooth if $\left\| \frac{\partial^k}{\partial x_m^k} f(\mathbf{x}) \right\|_\infty \leq L$ for all $1 \leq m \leq n$.

Now we show the corresponding observation for multivariate functions and polynomials:

**Corollary D.14.** *Let $\deg(P_f) = d$. For some $(k, L)$-smooth multivariate function $f$ (as in [Definition D.13](#)), then there exists a polynomial $P_f(\mathbf{x})$ with*

$$\deg(P_f) \leq O_k\left( \sqrt[k]{\frac{n L}{\epsilon}} \right)$$

*such that for all $\mathbf{x} \in [-1, 1]^n$*

$$|f(\mathbf{x}) - P_f(\mathbf{x})| \leq \epsilon.$$

*Proof.* Let $P_f$ be the polynomial we get from the left hand side of [Equation (16)](#). We want to upper bound the error as

$$\frac{c_k}{d^k} \sum_{j=1}^{n} \left\| \frac{\partial^{k+1}}{\partial x_j^{k+1}} f(\mathbf{x}) \right\|_\infty \leq \epsilon,$$

which follows if

$$\frac{c_k}{d^k} \sum_{j=1}^{n} L \le \epsilon$$

since $f$ is $(k, L)$-smooth. The above is the same as

$$\frac{c_k n L}{d^k} \le \epsilon,$$

or equivalently

$$\sqrt[k]{\frac{c_k n L}{\epsilon}} \le d.$$

Picking $d = \left\lceil \sqrt[k]{\frac{c_k n L}{\epsilon}} \right\rceil$ suffices. $\qquad \square$

**Arithmetic Circuit Notation.** We briefly recall arithmetic circuits Peter Bürgisser and Michael Clausen and M. Amin Shokrollah (1997). An *arithmetic circuit* $\mathcal{C}$ with variables $X \triangleq \{x_1, x_2, \ldots, x_n\}$ over a field $\mathbb{F}$ is interpreted as a directed acyclic graph, where the input nodes are labelled by either the variables from $X$ or constants from $\mathbb{F}$ and the internal nodes are labelled by $+$ or $\times$ with the output being the polynomial computed at the output node.

We shall also refer to the *size*[1] of the circuit $\mathcal{C}$ as the number of wires (or edges in $\mathcal{C}$), the *depth* of the circuit as the length of the longest path between an input node and the output node, and the *width* of the circuit as the number of wires that will be intersected by a horizontal 'cut' through the circuit. Moreover, the *degree* of a circuit is defined as the degree of the polynomial computed by the circuit. We summarize this with the following definition:

**Definition D.15.** An arithmetic circuit $\mathcal{C}$ is an $(n, s, \Delta, w)$-*circuit* if $\mathcal{C}$ is an $n$-variate arithmetic circuit of size $s$, depth at most $\Delta$, and width $w$.

**BASECONV Architecture.** In the following definitions we formally define the BASECONV model Arora et al. (2023). To formally define BASECONV, we will need the Kaleidoscope hierarchy Dao et al. (2020) as well.

To start, we define butterfly factors:

**Definition D.16.** A **butterfly factor** of size $k \ge 2$ (denoted as $\overline{\mathbf{B}}_k$) is a matrix of the form $\overline{\mathbf{B}}_k = \begin{bmatrix} \mathbf{D}_1 & \mathbf{D}_2 \\ \mathbf{D}_3 & \mathbf{D}_4 \end{bmatrix}$ where each $\mathbf{D}_i$ is a $\frac{k}{2} \times \frac{k}{2}$ diagonal matrix. We restrict $k$ to be a power of 2.

The following definition is for a butterfly factor matrix, which is made up of the above butterfly factors:

**Definition D.17.** A **butterfly factor matrix** of size $n$ with block size $k$ (denoted as $\overline{\mathbf{B}}_k^{(n)}$) is a block diagonal matrix of $\frac{n}{k}$ (possibly different) butterfly factors of size $k$:

$$\overline{\mathbf{B}}_k^{(n)} = \mathrm{diag}\left( \left[\overline{\mathbf{B}}_k\right]_1, \left[\overline{\mathbf{B}}_k\right]_2, \ldots, \left[\overline{\mathbf{B}}_k\right]_{\frac{n}{k}} \right)$$

Now lets define a butterfly matrix:

**Definition D.18.** A **butterfly matrix** of size $n$ (denoted as $\overline{\mathbf{B}}^{(n)}$) is a matrix that can be expressed as a product of butterfly factor matrices: $\overline{\mathbf{B}}^{(n)} = \overline{\mathbf{B}}_n^{(n)} \overline{\mathbf{B}}_{\frac{n}{2}}^{(n)} \ldots \overline{\mathbf{B}}_2^{(n)}$. Equivalently, we may define $\overline{\mathbf{B}}^{(n)}$ recursively as a matrix that can be expressed in the following form:

$$\overline{\mathbf{B}}^{(n)} = \overline{\mathbf{B}}_n^{(n)} \begin{bmatrix} [\overline{\mathbf{B}}^{(\frac{n}{2})}]_1 & 0 \\ 0 & [\overline{\mathbf{B}}^{(\frac{n}{2})}]_2 \end{bmatrix}$$

(Note that $[\overline{\mathbf{B}}^{(\frac{n}{2})}]_1$ and $[\overline{\mathbf{B}}^{(\frac{n}{2})}]_2$ may be different.)

---

[1]Note that if all the gates of an arithmetic circuit have bounded arity then the number of wires and gates are asymptotically the same but in this appendix we will consider gates with unbounded arity.

Using these butterfly matrices, lets define the Kaleidoscope Hierarchy:

**Definition D.19** (The Kaleidoscope Hierarchy (Dao et al., 2020))**.**

- Define $\mathcal{B}$ as the set of all matrices that can be expressed in the form $\overline{\mathbf{B}}^{(n)}$ (for some $n$).

- Define $(\mathcal{B}\mathcal{B}^*)$ as the set of matrices $\mathbf{M}$ of the form $\mathbf{M} = \mathbf{M}_1 \mathbf{M}_2^*$ for some $\mathbf{M_1}, \mathbf{M}_2 \in \mathcal{B}$.

- Define $(\mathcal{B}\mathcal{B}^*)^w$ as the set of matrices $\mathbf{M}$ that can be expressed as $\mathbf{M} = \mathbf{M}_w \ldots \mathbf{M}_2 \mathbf{M}_1$, with each $\mathbf{M}_i \in (\mathcal{B}\mathcal{B}^*) \, (1 \le i \le w)$. (The notation $w$ represents width.)

- Define $(\mathcal{B}\mathcal{B}^*)^w_e$ as the set of $n \times n$ matrices $\mathbf{M}$ that can be expressed as $\mathbf{M} = \mathbf{S}\mathbf{E}\mathbf{S}^\top$ for some $en \times en$ matrix $\mathbf{E} \in (\mathcal{B}\mathcal{B}^*)^w$, where $\mathbf{S} \in \mathbb{F}^{n \times en} = [\mathbf{I}_n \quad 0 \quad \ldots \quad 0]]$ (i.e. $\mathbf{M}$ is the upper-left corner of $\mathbf{E}$). (The notation $e$ represents expansion relative to $n$.)

Here we now formally define a BASECONV layer:

**Definition D.20** (BASECONV (Arora et al., 2023))**.** Given an input sequence $\mathbf{u} \in \mathbb{R}^{N \times D}$, where $N$ is the sequence length and $D$ is the model dimension, a learned weight matrix $\boldsymbol{W} \in \mathbb{R}^{D \times D}$ and biases $\boldsymbol{B}_1, \boldsymbol{B}_2 \in \mathbb{R}^{N \times D}$ and a matrix of convolution filters $\boldsymbol{H} \in \mathbb{R}^{N \times D}$, a BASECONV layer computes the following:

$$\boldsymbol{y}^{\text{BASECONV}} := (\mathbf{u}\boldsymbol{W} + \boldsymbol{B}_1) \odot (\boldsymbol{H} * \mathbf{u} + \boldsymbol{B}_2) \in \mathbb{R}^{N \times D}, \tag{18}$$

where the $j$th column of $\boldsymbol{H} * \mathbf{u} \in \mathbb{R}^{N \times D}$ is defined as $\boldsymbol{H}[:, j] * \mathbf{u}[:, j]$.

The corresponding pseudocode for a BASECONV layer is as follows:

---
**Algorithm 1** BASECONV$(\mathbf{u}, \boldsymbol{W}, \boldsymbol{B}_1, \boldsymbol{H}, \boldsymbol{B}_2)$

---
**Require:** Input sequence $\mathbf{u} \in \mathbb{R}^{N \times D}$, linear map $\boldsymbol{W} \in \mathbb{R}^{D \times D}$, convolution filter $\boldsymbol{H} \in \mathbb{R}^{N \times D}$, and bias matrices $\boldsymbol{B}_1, \boldsymbol{B}_2 \in \mathbb{R}^{N \times D}$.
1: In parallel for $0 \le n < N : \boldsymbol{x}[n, :] = \mathbf{u}[n, :] \cdot \boldsymbol{W}$
2: In parallel for $0 \le t < D : \boldsymbol{z}[:, t] = \boldsymbol{H}[:, t] * \mathbf{u}[:, t]$

3: In parallel for $0 \le t < D : \boldsymbol{y}[:, t] \leftarrow (\boldsymbol{x}[:, t] + \boldsymbol{B}_1[:, t]) \odot (\boldsymbol{z}[:, t] + \boldsymbol{B}_2[:, t])$.   ▷ See eq. (18)
4: **return** $\boldsymbol{y}$

---

**Remark D.21.** The definition of a BASECONV layer in Equation (19) has the input go through a linear layer before the convolution operation. For this section we will assume the linear layer is the identity matrix, as it is not needed for the results in this section.

**Assumption D.22.** Moving forward we assume the weight matrix $\boldsymbol{W} \in \mathbb{R}^{D \times D}$ in Definition D.20 also has the property $\boldsymbol{W} \in (\mathcal{B}\mathcal{B}^*)^{\text{poly-}\log D}_{\text{poly-}\log D}$. Consequently, each matrix $\boldsymbol{W}$ has $\tilde{\mathcal{O}}(D)$ parameters and runtime for matrix vector multiplication Dao et al. (2020).

In this section, we will establish some additional basic primitives that we expect need to implement via a BASECONV layer: shift and remember. We specify them below:

**Definition D.23.** shift$(\boldsymbol{y}, r, t, f)$
Shift an sequential input of length $N$ up or down by $s$ entries:
INPUT: $\boldsymbol{y} \in \mathbb{R}^{N \times D}, s \ge 0$.
OUTPUT: $\boldsymbol{z} \in \mathbb{R}^{N \times D}$ where $\boldsymbol{z}^+ = $ shift_down$(\boldsymbol{y}, s)$ and $\boldsymbol{z}^- = $ shift_up$(\boldsymbol{y}, s)$

$$
\boldsymbol{y} \equiv \begin{pmatrix} \leftarrow \boldsymbol{y}_0 \rightarrow \\ \vdots \\ \leftarrow \boldsymbol{y}_{i-1} \rightarrow \\ \leftarrow \boldsymbol{y}_i \rightarrow \\ \vdots \\ \leftarrow \boldsymbol{y}_{N-1} \rightarrow \end{pmatrix} \quad \boldsymbol{z}^{+} \equiv \begin{pmatrix} \leftarrow \boldsymbol{0} \rightarrow \\ \vdots \\ \leftarrow \boldsymbol{0} \rightarrow \\ \leftarrow \boldsymbol{y}_0 \rightarrow \\ \vdots \\ \leftarrow \boldsymbol{y}_{N-1-s} \rightarrow \end{pmatrix} \quad \boldsymbol{z}^{-} \equiv \begin{pmatrix} \leftarrow \boldsymbol{y}_s \rightarrow \\ \vdots \\ \leftarrow \boldsymbol{y}_{N-1} \rightarrow \\ \leftarrow \boldsymbol{0} \rightarrow \\ \vdots \\ \leftarrow \boldsymbol{0} \rightarrow \end{pmatrix}
$$

The following proposition is defining the convolution Kernel that computes the $\texttt{shift\_down}\left(\cdot, \lfloor \frac{N}{2} \rfloor\right)$ primitive:

**Proposition D.24.** *Define $\boldsymbol{H} \in \mathbb{R}^{2N \times D}$ as*

$$
\boldsymbol{H}[k, :] = \begin{cases} \mathbf{1}^{D} & \text{if } k = N \\ 0 & \text{otherwise} \end{cases}.
$$

*For any $\mathbf{u} \in \mathbb{R}^{2N \times D}$, $\boldsymbol{H} * \mathbf{u}$ will result in*

$$
\boldsymbol{H} * \begin{pmatrix} \mathbf{u}_1 \\ \mathbf{u}_2 \end{pmatrix} \rightarrow \begin{pmatrix} \mathbf{0}^{N \times D} \\ \mathbf{u}_1 \end{pmatrix},
$$

*where $\mathbf{u}_1, \mathbf{u}_2 \in \mathbb{R}^{N \times D}$.*

*Proof.* The convolution operation: $\boldsymbol{H} * \begin{pmatrix} \mathbf{u}_1 \\ \mathbf{u}_2 \end{pmatrix}$ where each column of $\boldsymbol{H}$ is convolved with each column of $\mathbf{u}$ can be restated as a polynomial multiplication. For column i, $0 \le i < 2N$,

$$
\boldsymbol{H}[:, i] * \begin{pmatrix} \mathbf{u}_1 \\ \mathbf{u}_2 \end{pmatrix}[:, i] = \text{coeff}((X^N \cdot \mathbf{u}[:, i](X)) \mod X^{2N}).
$$

Note that the columns of $\boldsymbol{H}$ are all $\mathbf{e}_N$ basis vectors and $\text{poly}(\mathbf{e}_N) = X^N$.

When we multiply the term through the input polynomial we get,

$$
\text{coeff}\left(X^N \cdot \left(\mathbf{u}[0][i] + \mathbf{u}[1][i]X + \cdots + \mathbf{u}[2N-1][i]X^{2N-1}\right) \mod X^{2N}\right)
$$
$$
= \text{coeff}(\mathbf{u}[0][i]X^N + \mathbf{u}[1][i]X^{N+1} + \cdots + \mathbf{u}[2N-1][i]X^{3N-1} \mod X^{2N}).
$$

With the lower order terms all becoming zeros, the above is same as

$$
\text{coeff}((0 + 0X + \cdots 0X^{N-1}
$$
$$
+ \mathbf{u}[0][i]X^N + \mathbf{u}[1][i]X^{N+1} + \cdots + \mathbf{u}[2N-1][i]X^{3N-1}) \mod X^{2N}).
$$

After we take the $\mod X^{2N}$ we get

$$
\text{coeff}(0 + 0X + \cdots + 0X^{N-1} + \mathbf{u}[0][i]X^N + \cdots + \mathbf{u}[N-1][i]X^{2N-1}),
$$

which implies that $\boldsymbol{H} * \begin{pmatrix} \mathbf{u}_1 \\ \mathbf{u}_2 \end{pmatrix}$ is

$$
\begin{pmatrix} \mathbf{0}^{N \times D} \\ \mathbf{u}_1 \end{pmatrix},
$$

as desired. $\square$

We also define the following primitive:

**Definition D.25.** `remember`$(\boldsymbol{y}, r, t, f)$

INPUT: $\boldsymbol{y} \in \mathbb{R}^{N' \times d'}, r \in \mathbb{Z}, t \in \mathbb{Z}, f : \mathbb{R}^{t-r} \to \mathbb{R}^{t-r+s}, \boldsymbol{v}_1 \in \mathbb{R}^r, \boldsymbol{x} \in \mathbb{R}^{t-r}$, where $\boldsymbol{y}$ is defined as below.

OUTPUT: $\boldsymbol{z} \in \mathbb{R}^{N' \times d'}$, which is defined as follows:

$$
\boldsymbol{y} \equiv \begin{pmatrix} \leftarrow \boldsymbol{v_1} \rightarrow \\ \hline \leftarrow \boldsymbol{x} \rightarrow \\ \hline \boldsymbol{0}^{s \times d'} \\ \hline \leftarrow \boldsymbol{v_2} \rightarrow \\ \hline \boldsymbol{0} \\ \hline \vdots \\ \hline \boldsymbol{0} \end{pmatrix}
\qquad
\boldsymbol{z} \equiv \begin{pmatrix} \leftarrow \boldsymbol{v_1} \rightarrow \\ \hline \leftarrow f(\boldsymbol{x}) \rightarrow \\ \hline \leftarrow \boldsymbol{v_2} \rightarrow \\ \hline \boldsymbol{0} \\ \hline \vdots \\ \hline \boldsymbol{0} \end{pmatrix}
$$

We will need the following BASECONV implementation of `remember`:

**Proposition D.26** (Arora et al. (2024), The Remembering Primitive). *For any* $\boldsymbol{x} \in \mathbb{R}^{n \times d'}, \boldsymbol{v}_1 \in \mathbb{R}^{r \times d'}, \boldsymbol{v}_2 \in \mathbb{R}^{m-r}$ *where* $n = t - r$ *contained in some* $\boldsymbol{y} \in \mathbb{R}^{N' \times d'}$ *such that* $\boldsymbol{v}_1$ *is in the first* $r$ *rows,* $\boldsymbol{x}$ *is in the next* $n$ *rows, 0s fill up the next* $s$ *rows, and* $\boldsymbol{v}_2$ *are in the next* $m - r$ *rows, for some* $3n + 3m + 2s + 2t \le N'$ *so that for* $\boldsymbol{h} \in \mathbb{R}^{n \times d}$ *and* $\boldsymbol{W} \in \mathbb{R}^{d' \times d'}$ *with* $\boldsymbol{x} * \boldsymbol{h} \in \mathbb{R}^{(n+s) \times d'}$ *and* $\boldsymbol{v} * \boldsymbol{h} \in \mathbb{R}^{(m+t) \times d'}$, *where* $\boldsymbol{v} \in \mathbb{R}^{m \times d'}$ *is defined as* $\boldsymbol{v}_2 + $ `shift_down`$(\boldsymbol{v}_1, m - r)$, *there exists a* $(N', 8, d', N', d') - $ BASECONV *that computes* `remember`$(\boldsymbol{y}, r, t, f)$, *where* $f$ *can be implemented in 1 layer of* BASECONV *through the parameters* $\boldsymbol{W} \in \mathbb{R}^{d' \times d'}, \boldsymbol{h} \in \mathbb{R}^{N' \times d'}, \boldsymbol{b}_1 \in \mathbb{R}^{N' \times d'}, \boldsymbol{b}_2 \in \mathbb{R}^{N' \times d'}$ *as defined below:*

$$
f(\boldsymbol{u}) = \left( \begin{pmatrix} \boldsymbol{uW} \\ \boldsymbol{0}^{s \times d'} \end{pmatrix} + \begin{pmatrix} \boldsymbol{b}_1 \\ \boldsymbol{1}^{s \times d'} \end{pmatrix} \right) \odot \left( \boldsymbol{u} * \boldsymbol{h} + \begin{pmatrix} \boldsymbol{b}_2 \\ \boldsymbol{0}^{s \times d'} \end{pmatrix} \right)
$$

We will also need the following generalization of the above result:

**Corollary D.27** (Arora et al. (2023)). *Let* $\boldsymbol{y}$ *be as in Proposition D.26 but now let* $f$ *be implemented with* BASECONV$(N, L, D, N, D)$. *Then* `remember`$(\boldsymbol{y}, r, t, f)$ *where* $t - r = n$ *can be implemented with* BASECONV *via* $(N, O(L), D, N, D) - $ BASECONV.

The rest of Appendix D will use this $5-$tuple notation for BASECONV:

**Definition D.28.** Lets define a 5-tuple notation for a BASECONV layer as $(N, \ell, D, N', D') - $ BASECONV with $\ell$ layers such that:

1. Input and output are $N \times D$ matrices.

2. Each layer is defined by Definition D.20 where $N$ and $D$ are replaced by $N'$ and $D'$. I.e. each layer takes in $N' \times D'$ matrices and output $N' \times D'$ matrices. We refer to the tuple $(N', D')$ as the *inner dimension* of the model.

3. The matrices are projected from $(N, D) \to (N', D')$ (and vice-versa) via a linear projection.

We state the following bounds on parameters and runtime for a single BASECONV layer:

**Proposition D.29** (Arora et al. (2023)). *An* $(N, 1, D, N, D) - $ BASECONV *requires* $\tilde{O}(ND)$ *parameters and runtime.*

We state the following result that says arithmetic circuit can be represented as a BASECONV model:

**Theorem D.30** (Arora et al. (2023), Theorem H.21). *For any* $(ND, s, \Delta, w)$-*arithmetic circuit* $\mathcal{C}$, *there exists an equivalent* $(N, \Delta', D, N', D') - $ BASECONV *with* $\Delta' = \mathcal{O}(\Delta \log w)$, $N' = \mathcal{O}(w), D' = D$ *that simulates* $\mathcal{C}$.

## D.2 Primitives

In this section, we provide theoretical results about primitives.

- In Appendix D.2.1, we implement the three primitives (READ, LINEAR, and MULTIPLY) from Section 3.3 using BASECONV, each using a single layer.
- Next, in Appendix D.2.2 and D.2.3, we briefly sketch how the three primitives READ, LINEAR, and MULTIPLY can be used in composition to exactly express gradient descent and Newton's method iterations on least squares (see Appendix A).
- Finally, in Appendix D.2.4, we provide a proof that a single layer of causal softmax attention cannot exactly represent the entry-wise squaring function. As a corollary, since entry-wise square is a special case of MULTIPLY, this implies that attention cannot exactly express the MULTIPLY task for all arguments.

**BASECONV parameterization** We recount the parameterization of BASECONV from Equation 2:

$$
\boldsymbol{y} := \left( \underbrace{(\boldsymbol{u} \cdot \boldsymbol{W}_{gate} + \boldsymbol{b}_{gate})}_{\text{Linear Projection}} \odot \underbrace{(\boldsymbol{h} * (\boldsymbol{u} \cdot \boldsymbol{W}_{in} + \boldsymbol{b}_{in}) + \boldsymbol{b}_{conv})}_{\text{Convolution}} \right) \cdot \boldsymbol{W}_{out} + \boldsymbol{b}_{out}
\tag{19}
$$
$$
:= W_{out}(W_{gate}(\mathbf{u}) \odot Conv(W_{in}(\mathbf{u})))
$$

where $W_{in}, W_{gate}, W_{out}$ are linear projections $\mathbb{R}^D \to \mathbb{R}^D$.

### D.2.1 1-LAYER BASECONV CAN IMPLEMENT LINEAR ALGEBRA PRIMITIVES

Below, we formally define the linear algebra primitives we discuss in Section 3.3, and we describe our BASECONV weight constructions.

**Read** The READ operator, which maps inputs $\mathbf{u} \in \mathbb{R}^{N \times d}$ to outputs $\mathbf{y} \in \mathbb{R}^{N \times d}$, is:

$$
\text{READ}(i, j, a, b)(\mathbf{u}) = \begin{cases} \mathbf{u}[k, a : b] & k \neq j \\ \mathbf{u}[i, a : b] & k = j \end{cases}.
\tag{20}
$$

Our implementation requires the use of the positional encodings and residual connections within the BASECONV architecture. Concretely, consider the input

$$
\boldsymbol{u}_{in} = \begin{pmatrix} \boldsymbol{e}_1 & \boldsymbol{e}_2 & \dots & \boldsymbol{e}_N \\ \boldsymbol{u}[1, :] & \boldsymbol{u}[2, :] & \dots & \boldsymbol{u}[N, :] \end{pmatrix},
$$

where the basis vector $\boldsymbol{e}_k$ represents the positional encoding for the $k$-th entry of the sequence. Define the output of the BASECONV layer *with residual connection*:

$$
\boldsymbol{y} := W_{out}(W_{gate}(\mathbf{u}) \odot Conv(W_{in}(\mathbf{u})) + \boldsymbol{u}).
$$

Then the following weight construction is equivalent to $\text{READ}(i, j, a, b)$:

- $W_{gate}(\boldsymbol{u}[k, :]) := \boldsymbol{u}[k, j]\mathbf{1}^D$
- $Conv(W_{in}(\boldsymbol{u}))[k, :] := \boldsymbol{u}[k + i - j, :] - \boldsymbol{u}[k, :]$
- $W_{out} := proj(a : b)$.

In particular, $W_{gate}$ is defined such that

$$
W_{gate}(\boldsymbol{u}[k, :]) = \begin{cases} \mathbf{1}^D & k = j \\ \mathbf{0}^D & k \neq j \end{cases}.
$$

Thus

$$
W_{gate}(\mathbf{u}) \odot Conv(W_{in}(\mathbf{u})) = \begin{cases} \boldsymbol{u}[k + i - j, :] - \boldsymbol{u}[k, :] = \boldsymbol{u}[i, :] - \boldsymbol{u}[j, :] & k = j \\ \mathbf{0}^D & k \neq j \end{cases}.
$$

Finally,

$$
W_{gate}(\mathbf{u}) \odot Conv(W_{in}(\mathbf{u})) + \boldsymbol{u} = \begin{cases} \boldsymbol{u}[i, :] & k = j \\ \boldsymbol{u}[k, :] & k \neq j \end{cases}
$$

so the final output of this layer will be exactly equivalent to $\text{READ}(i, j, a, b)$.

**Linear transformation** The LINEAR operator, which maps inputs $\mathbf{u} \in \mathbb{R}^{N \times d_{in}}$ to outputs $\mathbf{y} \in \mathbb{R}^{N \times d_{out}}$, is:

$$\text{LINEAR}(\boldsymbol{H})(\mathbf{u}) = \mathbf{u}\boldsymbol{H} \tag{21}$$

where $\boldsymbol{H} : \mathbb{R}^{d_{in}} \to \mathbb{R}^{d_{out}}$ is a linear map.

Define $Conv(W_{in}(\mathbf{u})) = \mathbf{1}_D$, $W_{gate} = I$, and $W_{out} = \boldsymbol{H}$. Then

$$W_{gate}(\mathbf{u}) \odot Conv(W_{in}(\mathbf{u})) = \boldsymbol{u}$$

so

$$W_{out}(W_{gate}(\mathbf{u}) \odot Conv(W_{in}(\mathbf{u}))) = \boldsymbol{u}\boldsymbol{H}.$$

Thus the output of this layer is exactly equivalent to $\text{LINEAR}(\boldsymbol{H})$.

**Element-wise multiply** The MULTIPLY operator, which maps inputs $\mathbf{u} \in \mathbb{R}^{N \times d_{in}}$ to outputs $\mathbf{y} \in \mathbb{R}^{N \times d_{out}}$, is:

$$\text{MULTIPLY}(a, b, d_{out})(\mathbf{u}) = \mathbf{u}[:, a : a + d_{out}] \odot \mathbf{u}[:, b : b + d_{out}] \tag{22}$$

Define $Conv = $ Identity, $W_{in} = proj(a : a + d_{out})$, $W_{gate} = proj(b : b + d_{out})$, and $W_{out} = \boldsymbol{I}$.

Then

$$W_{gate}(\mathbf{u}) \odot Conv(W_{in}(\mathbf{u})) = \mathbf{u}[:, a : a + d_{out}] \odot \mathbf{u}[:, b : b + d_{out}].$$

Since $W_{out} = \boldsymbol{I}$, the output of this layer will be equivalent to $\text{MULTIPLY}(a, b, d_{out})$.

### D.2.2 GRADIENT DESCENT

We assume our input is of the form

$$\boldsymbol{u} = \begin{pmatrix} \boldsymbol{a}_1 & \dots & \boldsymbol{a}_N & \boldsymbol{x}_0 \\ b_1 & \dots & b_N & 0 \end{pmatrix}.$$

Our goal is to compute the gradient update

$$\boldsymbol{x}_1 := \boldsymbol{x}_0 - \eta \sum_{i=1}^{N} (\boldsymbol{x}_0^T \boldsymbol{a}_i - b_i) \boldsymbol{a}_i. \tag{23}$$

Intuitively, our argument proceeds similarly to the causal gradient descent construction from Appendix D.3.1:

- First, we repeatedly apply READ and LINEAR to move the information $\{\boldsymbol{a}_i, b_i\} \forall i$ into e.g. the final entry of the sequence. Without loss of generality, we omit the rest of the sequence, and assume we have access to a large enough embedding dimension that we can make use of arbitrary amounts of memory.

  After this phase, our $\boldsymbol{u}$ is of the form

  $$\dots \begin{pmatrix} \boldsymbol{x}_0 & 0 & \boldsymbol{a}_1 & \dots & \boldsymbol{a}_N & b_1 & \dots & b_N & \dots \end{pmatrix}^T.$$

- Next, we use MULTIPLY and LINEAR to compute and store $\{\boldsymbol{x}_0^T \boldsymbol{a}_i\}$ for all $i$. We will end up with

  $$\boldsymbol{u} = \dots \begin{pmatrix} \boldsymbol{x}_0 & 0 & \{\boldsymbol{a}_i\}_i & \{b_i\}_i & \{\boldsymbol{x}_0^T \boldsymbol{a}_i\}_i & \dots \end{pmatrix}.$$

- We use LINEAR to compute and store $\{\boldsymbol{x}_0^T \boldsymbol{a}_i - b_i\}$ for all $i$:

  $$\boldsymbol{u} = \dots \begin{pmatrix} \boldsymbol{x}_0 & 0 & \{\boldsymbol{a}_i\}_i & \{b_i\}_i & \{\boldsymbol{x}_0^T \boldsymbol{a}_i\}_i & \{\boldsymbol{x}_0^T \boldsymbol{a}_i - b_i\}_i & \dots \end{pmatrix}.$$

- We use MULTIPLY and LINEAR to compute and store $\{(\boldsymbol{x}_0^T \boldsymbol{a}_i - b_i) \boldsymbol{a}_i\}$ for all $i$:

  $$\boldsymbol{u} = \dots \begin{pmatrix} \boldsymbol{x}_0 & 0 & \{\boldsymbol{a}_i\}_i & \{b_i\}_i & \{\boldsymbol{x}_0^T \boldsymbol{a}_i\}_i & \{(\boldsymbol{x}_0^T \boldsymbol{a}_i - b_i)\boldsymbol{a}_i\}_i & \dots \end{pmatrix}.$$

- Finally, we can use LINEAR to compute the gradient update:

  $$\boldsymbol{u} = \dots \begin{pmatrix} \boldsymbol{x}_0 - \eta \sum_{i=1}^{N} (\boldsymbol{x}_0^T \boldsymbol{a}_i - b_i)\boldsymbol{a}_i & 0 & \{\boldsymbol{a}_i\}_i & \{b_i\}_i & \{\boldsymbol{x}_0^T \boldsymbol{a}_i\}_i & \{(\boldsymbol{x}_0^T \boldsymbol{a}_i - b_i)\boldsymbol{a}_i\}_i & \dots \end{pmatrix}.$$

### D.2.3 NEWTON'S METHOD

We assume our input is of the form

$$
\boldsymbol{u} = \begin{pmatrix} \boldsymbol{a}_1 & \ldots & \boldsymbol{a}_N & \boldsymbol{M}_0[1,:] & \ldots & \boldsymbol{M}_0[D,:] \\ b_1 & \ldots & b_N & 0 & \ldots & 0 \end{pmatrix}.
$$

Our goal is to compute the Newton's iterate:

$$
\boldsymbol{M}_1 := \boldsymbol{M}_0(2\boldsymbol{I} - (\boldsymbol{a}^T\boldsymbol{a})\boldsymbol{M}_0), \tag{24}
$$

where

$$
\boldsymbol{a} = \begin{pmatrix} \leftarrow \boldsymbol{a}_1 \rightarrow \\ \vdots \\ \leftarrow \boldsymbol{a}_N \rightarrow \end{pmatrix}, \quad \boldsymbol{b} = \begin{pmatrix} b_1 \\ \vdots \\ b_N \end{pmatrix}. \tag{25}
$$

For any matrix $\boldsymbol{M} \in \mathbb{R}^{n \times p}$, let $flt$ denote the `flatten` operation, so that $flt(\boldsymbol{M})$ represent a vectorized version of $\boldsymbol{M}$: $flt(\boldsymbol{M}) \in \mathbb{R}^{np}$.

We proceed similarly to the argument from Appendix D.2.2.

- First, we repeatedly apply READ and LINEAR to move all information $\{\boldsymbol{a}_i\}_i \; \forall i$ and $flt(\boldsymbol{M})$ to e.g. the final entry of the sequence. We omit the rest of the sequence for notational ease, and we assume we have access to a large enough embedding dimension that we can make use of arbitrary amounts of memory.

  After this phase, we have

  $$
  \boldsymbol{u} = \ldots (flt(\boldsymbol{M}_0) \quad \{\boldsymbol{a}_i\}_i \quad \ldots).
  $$

- Using LINEAR, we can copy and rearrange the $\boldsymbol{a}_i$'s to construct copies of $flt(\boldsymbol{a})$ and $flt(\boldsymbol{a}^T)$:

  $$
  \boldsymbol{u} = \ldots \big(flt(\boldsymbol{M}_0) \quad \{\boldsymbol{a}_i\}_i \quad flt(\boldsymbol{a}^T) \quad flt(\boldsymbol{a}) \quad \ldots\big).
  $$

- Now, note that we can represent the matrix multiplication $\boldsymbol{a}^T\boldsymbol{a}$ as a linear combination of the entries of the element-wise multiplication $flt(\boldsymbol{a}^T) \odot flt(\boldsymbol{a})$. This means that we can obtain $flt(\boldsymbol{a}^T\boldsymbol{a})$ using a single application of MULTIPLY and LINEAR:

  $$
  \boldsymbol{u} = \ldots \big(flt(\boldsymbol{M}_0) \quad \{\boldsymbol{a}_i\}_i \quad flt(\boldsymbol{a}^T) \quad flt(\boldsymbol{a}) \quad flt(\boldsymbol{a}^T\boldsymbol{a}) \ldots\big).
  $$

- By the same argument, we can obtain $flt((\boldsymbol{a}^T\boldsymbol{a})\boldsymbol{M}_0)$ using another application of MULTIPLY and LINEAR:

  $$
  \boldsymbol{u} = \ldots \big(flt(\boldsymbol{M}_0) \quad \{\boldsymbol{a}_i\}_i \quad flt(\boldsymbol{a}^T) \quad flt(\boldsymbol{a}) \quad flt((\boldsymbol{a}^T\boldsymbol{a})\boldsymbol{M}_0) \ldots\big).
  $$

- Finally, we have that $flt(\boldsymbol{M}_1) := 2flt(\boldsymbol{M}_0) - flt((\boldsymbol{a}^T\boldsymbol{a})\boldsymbol{M}_0)$ can be obtained using LINEAR once more:

  $$
  \boldsymbol{u} = \ldots \big(flt(\boldsymbol{M}_1) \quad \{\boldsymbol{a}_i\}_i \quad flt(\boldsymbol{a}^T) \quad flt(\boldsymbol{a}) \quad flt((\boldsymbol{a}^T\boldsymbol{a})\boldsymbol{M}_0) \ldots\big).
  $$

### D.2.4 Softmax attention can't implement element-wise squaring.

In this section, we consider the following parameterization of *softmax attention*:

$$\text{Attn}(\boldsymbol{u}) = \text{softmax}\left((\boldsymbol{u}\boldsymbol{W_Q})(\boldsymbol{u}\boldsymbol{W_K})^T + \boldsymbol{M}\right)(\boldsymbol{u}\boldsymbol{W_V} + \boldsymbol{B}), \tag{26}$$

where $\boldsymbol{u} \in \mathbb{R}^{N \times D}$, $\boldsymbol{W_Q}, \boldsymbol{W_K}, \boldsymbol{W_V} \in \mathbb{R}^{D \times D}$, $\boldsymbol{B} \in \mathbb{R}^{N \times D}$, and $\boldsymbol{M} \in \mathbb{R}^{N \times N}$ is the causal attention mask:

$$\boldsymbol{M}_{ij} = \begin{cases} -\infty & i < j \\ 0 & \text{otherwise} \end{cases} \tag{27}$$

**Theorem D.31.** *One-layer single-headed causal softmax attention cannot exactly represent the entry-wise squaring function* SQUARE $: \mathbb{R}^{N \times D} \to \mathbb{R}^{N \times D}$ *s.t.*

$$\text{SQUARE}(\boldsymbol{u})_{ij} = \boldsymbol{u}_{ij}^2$$

*for all* $\boldsymbol{u} \in \mathbb{R}^{N \times D}$.

*Proof.* We proceed by contradiction. Let's assume there exists $\boldsymbol{W_Q}, \boldsymbol{W_K}, \boldsymbol{W_V}, \boldsymbol{B} \in \mathbb{R}^{D \times D}$ and $\boldsymbol{B} \in \mathbb{R}^{N \times D}$ such that $\forall \boldsymbol{u} \in \mathbb{R}^{N \times D}$,

$$\text{softmax}\left((\boldsymbol{u}\boldsymbol{W_Q})(\boldsymbol{u}\boldsymbol{W_K})^T + \boldsymbol{M}\right)(\boldsymbol{u}\boldsymbol{W_V} + \boldsymbol{B}) = \text{SQUARE}(\boldsymbol{u}). \tag{28}$$

Consider the set of inputs $\boldsymbol{u} \in \mathbb{R}^{N \times D}$ with at most one non-zero entry, defined as

$$\boldsymbol{u}[i, j] = \begin{cases} \boldsymbol{u}_{ab} & (i, j) = (a, b) \\ 0 & \text{else} \end{cases} \tag{29}$$

for an arbitrary choice of $a \in [N]$, $b \in [D]$. Then:

$$\boldsymbol{Q} := \boldsymbol{u}\boldsymbol{W_Q} = \begin{pmatrix} \boldsymbol{0}^N \\ \vdots \\ \boldsymbol{0}^N \\ \boldsymbol{u}_{ab}\boldsymbol{W_Q}[b, :] \\ \vdots \\ \boldsymbol{0}^N \end{pmatrix} \tag{30}$$

where $\boldsymbol{Q}$'s rows are all $\boldsymbol{0}^N$ except for the $a$-th, which is $\boldsymbol{u}_{ab}\boldsymbol{W_Q}[b, :]$.

Similarly:

$$\boldsymbol{K} := \boldsymbol{u}\boldsymbol{W_K} = \begin{pmatrix} \boldsymbol{0}^N \\ \vdots \\ \boldsymbol{0}^N \\ \boldsymbol{u}_{ab}\boldsymbol{W_K}[b, :] \\ \vdots \\ \boldsymbol{0}^N \end{pmatrix} \tag{31}$$

and

$$\boldsymbol{V} := \boldsymbol{u}\boldsymbol{W_V} = \begin{pmatrix} \boldsymbol{0}^N \\ \vdots \\ \boldsymbol{0}^N \\ \boldsymbol{u}_{ab}\boldsymbol{W_V}[b,:] \\ \vdots \\ \boldsymbol{0}^N \end{pmatrix} \tag{32}$$

Then the pre-softmax attention matrix, $\boldsymbol{A}' = \boldsymbol{Q}\boldsymbol{K}^T$, satisfies

$$\boldsymbol{A}'_{ij} = \begin{cases} \boldsymbol{u}_{ab}^2(\boldsymbol{W_Q}\boldsymbol{W_K}^T)[b,b] & (i,j) = (a,a) \\ 0 & \text{otherwise} \end{cases}. \tag{33}$$

Define

$$C := \boldsymbol{u}_{ab}^2(\boldsymbol{W_Q}\boldsymbol{W_K}^T)[b,b]. \tag{34}$$

Now consider what happens after we apply the softmax operator. Recall that the softmax operator is defined as

$$\text{softmax}(\boldsymbol{z})[i] = \frac{\exp(\boldsymbol{z}[i])}{\sum_{j=1}^D \exp(\boldsymbol{z}[j])} \tag{35}$$

for $\boldsymbol{z} \in \mathbb{R}^D$. Then $\boldsymbol{A} := \text{softmax}(\boldsymbol{A}' + \boldsymbol{M})$ satisfies

$$\boldsymbol{A}_{ij} = \begin{cases} \frac{1}{i} & i \neq a \\ \frac{1}{\exp(C)+a-1} & i = a, j \neq a \\ \frac{\exp(C)}{\exp(C)+a-1} & (i,j) = (a,a) \end{cases} \tag{36}$$

Now let's consider the output of softmax attention:

$$\boldsymbol{O} = \boldsymbol{A}(\boldsymbol{V} + \boldsymbol{B}) \tag{37}$$

such that $\boldsymbol{O} = \text{SQUARE}(\boldsymbol{u})$.

Note that for $i \neq a$:

$$\boldsymbol{O}[i,:] = \frac{1}{i}\sum_{k=1}^i (\boldsymbol{V} + \boldsymbol{B})[k,:] \tag{38}$$

and this must also be equal to $\boldsymbol{0}^N = \text{SQUARE}(\boldsymbol{u})[i,:]$. We consider three cases:

- First, consider $i < a$ in order from $i = 1, \ldots, a - 1$. Since this equality is true for all $i < a$, we can verify that $(\boldsymbol{V} + \boldsymbol{B})[i,:]$ must equal $\boldsymbol{0}^N$ for all $i < a$.

- Next, looking at $i = a + 1$, we have

$$\frac{1}{a+1}\left((\boldsymbol{V} + \boldsymbol{B})[a,:] + (\boldsymbol{V} + \boldsymbol{B})[a+1,:]\right) = \boldsymbol{0}^N \tag{39}$$

so we must have

$$(\boldsymbol{V} + \boldsymbol{B})[a,:] = -(\boldsymbol{V} + \boldsymbol{B})[a+1,:] \tag{40}$$

- Finally, from $i \geq a + 1$, we can again conclude that $(\boldsymbol{V} + \boldsymbol{B})[i,:]$ must equal $\boldsymbol{0}^N$ for all $i > a + 1$.

This means the only rows of $\boldsymbol{V} + \boldsymbol{B}$ that might not be zero are $(\boldsymbol{V} + \boldsymbol{B})[a, :]$ and $(\boldsymbol{V} + \boldsymbol{B})[a + 1, :]$. Thus looking at the $a$-th row:

$$\frac{\exp(C)}{\exp(C) + a - 1}(\boldsymbol{V} + \boldsymbol{B})[a, :] = \text{SQUARE}(\boldsymbol{u})[a, :]$$

$$= \begin{bmatrix} 0 & \dots & \boldsymbol{u}_{ab}^2 & \dots & 0 \end{bmatrix}$$

Recall that from above,

$$\boldsymbol{V}[a, :] = \boldsymbol{u}_{ab} \boldsymbol{W_V}[b, :] \tag{41}$$

Then analyzing entry-wise, we have:

$$\frac{\exp(C)}{\exp(C) + a - 1} \left( \boldsymbol{u}_{ab} \boldsymbol{W_V}[b, j] + \boldsymbol{B}[a, j] \right) = 0 \tag{42}$$

for all $j \neq b$, and

$$\frac{\exp(C)}{\exp(C) + a - 1} \left( \boldsymbol{u}_{ab} \boldsymbol{W_V}[b, b] + \boldsymbol{B}[a, b] \right) = \boldsymbol{u}_{ab}^2. \tag{43}$$

We now plug back in our expression for $C$ and simplifying the latter equation. For ease of notation, denote $A := (\boldsymbol{W_Q} \boldsymbol{W_K}^T)_{bb}$, $V := \boldsymbol{W_V}[b, b]$, and $B := \boldsymbol{B}[a, b]$. Then the expression simplifies to:

$$V \exp(A \boldsymbol{u}_{ab}^2) \boldsymbol{u}_{ab} + B \exp(A \boldsymbol{u}_{ab}^2) = \exp(A \boldsymbol{u}_{ab}^2) \boldsymbol{u}_{ab}^2 + (a - 1) \boldsymbol{u}_{ab}^2$$

This must hold for all non-zero values of $\boldsymbol{u}_{ab}$. We can take $V = B = 0$, but we are still left with

$$- \exp(A \boldsymbol{u}_{ab}^2) \boldsymbol{u}_{ab}^2 = (a - 1) \boldsymbol{u}_{ab}^2$$

$$- \exp(A \boldsymbol{u}_{ab}^2) = a - 1$$

However, there is no choice of $A$ such that this statement holds. This completes the proof by contradiction.

As a corollary, we have

**Corollary D.32.** *One-layer single-headed causal softmax attention cannot exactly represent the entry-wise multiply function* MULTIPLY $: \mathbb{R}^{N \times D} \to \mathbb{R}^{N \times d_{out}}$ *s.t.*

$$\text{MULTIPLY}(a, b, d_{out})(\mathbf{u}) = \mathbf{u}[:, a : a + d_{out}] \odot \mathbf{u}[:, b : b + d_{out}] \tag{44}$$

*for all* $\boldsymbol{u} \in \mathbb{R}^{N \times D}$ *and all choices of* $a$, $b$, $d_{out}$.

*Proof.* Note that for $a = 0$, $b = 0$, and $d_{out} = D$,

$$\text{SQUARE}(\boldsymbol{u}) = \text{MULTIPLY}(a, b, d_{out})(\boldsymbol{u}).$$

Since softmax attention cannot exactly represent SQUARE for all $\boldsymbol{u}$, it also cannot represent MULTIPLY for all $\boldsymbol{u}$. □

### D.3 UPPER AND LOWER BOUNDS WITH BASECONV FOR GRADIENT DESCENT

In this section, we detail upper and lower bounds for implementing gradient descent using BASEC-ONV, as discussed in Section 4.1.

- **Upper bounds.** We provide two explicit constructions for implementing iterations gradient descent on linear regression: one for *non-causal* BASECONV requiring $O(1)$ layers and $O(D)$ state size, and one for *causal* BASECONV requiring $O(1)$ layers and $O(D^2)$ state size.

- **Lower bounds.** In Appendix D.3.2, we prove that our constructions are asymptotically optimal with respect to layers and state size.

#### D.3.1 UPPER BOUNDS: BASECONV CAN IMPLEMENT GRADIENT DESCENT FOR LINEAR REGRESSION

In this section, we provide weight constructions for exactly implementing gradient descent on linear regression. Recall:

$$\mathcal{L}_N = \frac{1}{2N} \sum_{i=1}^{N} (\boldsymbol{x}^T \boldsymbol{a}_i - \boldsymbol{b}_i)^2 \tag{45}$$

so

$$\nabla_{\boldsymbol{x}} \mathcal{L}_N = \frac{1}{N} \sum_{i=1}^{N} (\boldsymbol{x}^T \boldsymbol{a}_i - \boldsymbol{b}_i) \boldsymbol{a}_i \tag{46}$$

$$= \frac{1}{N} \left( \sum_{i=1}^{N} \boldsymbol{b}_i \boldsymbol{a}_i - \left( \sum_{i=1}^{N} \boldsymbol{a}_i \boldsymbol{a}_i^T \right) \boldsymbol{x} \right) \tag{47}$$

**Non-causal BASECONV** This weight construction uses Equation 46 to compute the gradient descent update.

We note that non-causal constructions for in-context linear regression are standard in the literature: e.g. Von Oswald et al. (2023); Ahn et al. (2024).

We start with input:

$$\boldsymbol{b} \equiv \begin{pmatrix} \boldsymbol{a}_1 & \dots & \boldsymbol{a}_N & \boldsymbol{a}_q \\ \boldsymbol{b}_1 & \dots & \boldsymbol{b}_N & 0 \end{pmatrix}$$

We define the initial embedding:

$$\begin{pmatrix} \boldsymbol{a}_1 & \dots & \boldsymbol{a}_N & \boldsymbol{0}^D \\ \boldsymbol{b}_1 & \dots & \boldsymbol{b}_N & 0 \\ \boldsymbol{x}_0 & \dots & \boldsymbol{x}_0 & \boldsymbol{x}_0 \\ \boldsymbol{0}^D & \dots & \boldsymbol{0}^D & \boldsymbol{0}^D \\ \boldsymbol{0}^D & \dots & \boldsymbol{0}^D & \boldsymbol{0}^D \\ \boldsymbol{0}^D & \dots & \boldsymbol{0}^D & \boldsymbol{a}_q \\ 0 & \dots & 0 & 0 \end{pmatrix}$$

We drop the bottom two rows of the block matrix representation for now and show how to perform the gradient descent update with the rest of the embedding.

Layer 1:

$$
\underbrace{\begin{pmatrix} \leftarrow \boldsymbol{a}_i \rightarrow \\ \leftarrow \boldsymbol{b}_i \rightarrow \\ \leftarrow \boldsymbol{x}_0 \rightarrow \\ \leftarrow \boldsymbol{a}_i \rightarrow \\ \leftarrow \boldsymbol{0}^D \rightarrow \end{pmatrix}}_{conv(in\_proj(\cdot))} \odot \underbrace{\begin{pmatrix} \leftarrow \boldsymbol{1}^D \rightarrow \\ \leftarrow \boldsymbol{1} \rightarrow \\ \leftarrow \boldsymbol{1}^D \rightarrow \\ \leftarrow \boldsymbol{x}_0 \rightarrow \\ \leftarrow \boldsymbol{0}^D \rightarrow \end{pmatrix}}_{gate\_proj(\cdot)} = \begin{pmatrix} \leftarrow \boldsymbol{a}_i \rightarrow \\ \leftarrow \boldsymbol{b}_i \rightarrow \\ \leftarrow \boldsymbol{x}_0 \rightarrow \\ \leftarrow \boldsymbol{a}_i \odot \boldsymbol{x}_0 \rightarrow \\ \leftarrow \boldsymbol{0}^D \rightarrow \end{pmatrix}
$$

$$
\begin{pmatrix} \leftarrow \boldsymbol{a}_i \rightarrow \\ \leftarrow \boldsymbol{b}_i \rightarrow \\ \leftarrow \boldsymbol{x}_0 \rightarrow \\ \leftarrow \boldsymbol{a}_i \odot \boldsymbol{x}_0 \rightarrow \\ \leftarrow \boldsymbol{0}^D \rightarrow \end{pmatrix} \underbrace{\rightarrow}_{out\_proj(\cdot)} \begin{pmatrix} \leftarrow \boldsymbol{a}_i \rightarrow \\ \leftarrow \boldsymbol{b}_i \rightarrow \\ \leftarrow \boldsymbol{x}_0 \rightarrow \\ \leftarrow \boldsymbol{a}_i \odot \boldsymbol{x}_0 \rightarrow \\ \leftarrow (\boldsymbol{x}_0^T \boldsymbol{a}_i - \boldsymbol{b}_i)\boldsymbol{1}^D \rightarrow \end{pmatrix}
$$

Layer 2:

$$
\underbrace{\begin{pmatrix} \leftarrow \boldsymbol{a}_i \rightarrow \\ \leftarrow \boldsymbol{b}_i \rightarrow \\ \leftarrow \boldsymbol{x}_0 \rightarrow \\ \leftarrow \boldsymbol{a}_i \odot \boldsymbol{x}_0 \rightarrow \\ \leftarrow (\boldsymbol{x}_0^T \boldsymbol{a}_i - \boldsymbol{b}_i)\boldsymbol{1}^D \rightarrow \end{pmatrix}}_{conv(in\_proj(\cdot))} \odot \underbrace{\begin{pmatrix} \leftarrow \boldsymbol{1}^D \rightarrow \\ \leftarrow \boldsymbol{1} \rightarrow \\ \leftarrow \boldsymbol{1}^D \rightarrow \\ \leftarrow \boldsymbol{1}^D \rightarrow \\ \leftarrow \boldsymbol{a}_i \rightarrow \end{pmatrix}}_{gate\_proj(\cdot)} = \begin{pmatrix} \leftarrow \boldsymbol{a}_i \rightarrow \\ \leftarrow \boldsymbol{b}_i \rightarrow \\ \leftarrow \boldsymbol{x}_0 \rightarrow \\ \leftarrow \boldsymbol{a}_i \odot \boldsymbol{x}_0 \rightarrow \\ \leftarrow (\boldsymbol{x}_0^T \boldsymbol{a}_i - \boldsymbol{b}_i)\boldsymbol{a}_i \rightarrow \end{pmatrix}
$$

$$
\begin{pmatrix} \leftarrow \boldsymbol{a}_i \rightarrow \\ \leftarrow \boldsymbol{b}_i \rightarrow \\ \leftarrow \boldsymbol{x}_0 \rightarrow \\ \leftarrow \boldsymbol{a}_i \odot \boldsymbol{x}_0 \rightarrow \\ \leftarrow (\boldsymbol{x}_0^T \boldsymbol{a}_i - \boldsymbol{b}_i)\boldsymbol{a}_i \rightarrow \end{pmatrix} \underbrace{\rightarrow}_{out\_proj(\cdot)=Identity} \begin{pmatrix} \leftarrow \boldsymbol{a}_i \rightarrow \\ \leftarrow \boldsymbol{b}_i \rightarrow \\ \leftarrow \boldsymbol{x}_0 \rightarrow \\ \leftarrow \boldsymbol{a}_i \odot \boldsymbol{x}_0 \rightarrow \\ \leftarrow (\boldsymbol{x}_0^T \boldsymbol{a}_i - \boldsymbol{b}_i)\boldsymbol{a}_i \rightarrow \end{pmatrix}
$$

Layer 3:

$$
\begin{pmatrix} \leftarrow \boldsymbol{a}_i \rightarrow \\ \leftarrow \boldsymbol{b}_i \rightarrow \\ \leftarrow \boldsymbol{x}_0 \rightarrow \\ \leftarrow \boldsymbol{a}_i \odot \boldsymbol{x}_0 \rightarrow \\ \leftarrow (\boldsymbol{x}_0^T \boldsymbol{a}_i - \boldsymbol{b}_i)\boldsymbol{a}_i \rightarrow \end{pmatrix} \underbrace{\rightarrow}_{conv(in\_proj(\cdot))} \begin{pmatrix} \leftarrow \boldsymbol{a}_i \rightarrow \\ \leftarrow \boldsymbol{b}_i \rightarrow \\ \leftarrow \boldsymbol{x}_0 \rightarrow \\ \leftarrow \boldsymbol{a}_i \odot \boldsymbol{x}_0 \rightarrow \\ \leftarrow \sum_{i=1}^{N}(\boldsymbol{x}_0^T \boldsymbol{a}_i - \boldsymbol{b}_i)\boldsymbol{a}_i \rightarrow \end{pmatrix}
$$

$$\begin{pmatrix} \leftarrow \boldsymbol{a}_i \rightarrow \\ \leftarrow \boldsymbol{b}_i \rightarrow \\ \leftarrow \boldsymbol{x}_0 \rightarrow \\ \leftarrow \boldsymbol{a}_i \odot \boldsymbol{x}_0 \rightarrow \\ \leftarrow \underbrace{\sum_{i=1}^{N} (\boldsymbol{x}_0^T \boldsymbol{a}_i - \boldsymbol{b}_i) \boldsymbol{a}_i}_{= \nabla_{\boldsymbol{x}} \mathcal{L}(\boldsymbol{x}_0)} \rightarrow \end{pmatrix} \underbrace{\rightarrow}_{out\_proj(\cdot)} \begin{pmatrix} \leftarrow \boldsymbol{a}_i \rightarrow \\ \leftarrow \boldsymbol{b}_i \rightarrow \\ \leftarrow \boldsymbol{x}_0 - \eta \nabla_{\boldsymbol{x}} \mathcal{L}(\mathbf{w_0}) \rightarrow \\ \leftarrow \mathbf{0}^D \rightarrow \\ \leftarrow \mathbf{0}^D \rightarrow \end{pmatrix}$$

**Causal BASECONV** This weight construction uses Equation 47 to compute the gradient descent update.

We start with input:

$$\boldsymbol{b} \equiv \begin{pmatrix} \boldsymbol{a}_1 & \dots & \boldsymbol{a}_N & \mathbf{0}^D \\ \boldsymbol{b}_1 & \dots & \boldsymbol{b}_N & 0 \\ \mathbf{0}^D & \dots & \mathbf{0}^D & \boldsymbol{x}_0 \end{pmatrix}$$

We use two BASECONV layers to construct an initial embedding, after which each gradient descent update step will only require a single BASECONV layer.

In the following construction, we use $flt$ to denote the $\texttt{flatten}$ operation, which maps an $M \times N$ matrix to a $MN$-entry vector with the same elements.

Layer 1:

$$\underbrace{\begin{pmatrix} \boldsymbol{a}_1 & \dots & \boldsymbol{a}_N & \mathbf{0}^D \\ \boldsymbol{b}_1 & \dots & \boldsymbol{b}_N & 0 \\ \mathbf{0}^D & \dots & \mathbf{0}^D & \boldsymbol{x}_0 \\ \boldsymbol{a}_1 & \dots & \boldsymbol{a}_N & \mathbf{0}^D \\ flt(\boldsymbol{a}_1(\mathbf{1}^D)^T) & \dots & flt(\boldsymbol{a}_N(\mathbf{1}^D)^T) & flt(\mathbf{0}^D(\mathbf{0}^D)^T) \end{pmatrix}}_{conv(in\_proj(\cdot))} \odot \underbrace{\begin{pmatrix} \leftarrow \mathbf{1}^D \rightarrow \\ \leftarrow \mathbf{1} \rightarrow \\ \leftarrow \mathbf{1}^D \rightarrow \\ \boldsymbol{b}_1\mathbf{1}^D & \dots & \boldsymbol{b}_N\mathbf{1}^D & \mathbf{0}^D \\ flt(\mathbf{1}^D\boldsymbol{a}_1^T) & \dots & flt(\mathbf{1}^D\boldsymbol{a}_N^T) & flt(\mathbf{0}^D(\mathbf{0}^D)^T) \end{pmatrix}}_{gate\_proj(\cdot)} =$$

$$\begin{pmatrix} \boldsymbol{a}_1 & \dots & \boldsymbol{a}_N & \mathbf{0}^D \\ \boldsymbol{b}_1 & \dots & \boldsymbol{b}_N & 0 \\ \mathbf{0}^D & \dots & \mathbf{0}^D & \boldsymbol{x}_0 \\ \boldsymbol{b}_1\boldsymbol{a}_1 & \dots & \boldsymbol{b}_1\boldsymbol{a}_N & \mathbf{0}^D \\ flt(\boldsymbol{a}_1\boldsymbol{a}_1^T) & \dots & flt(\boldsymbol{a}_N\boldsymbol{a}_N^T) & flt(\mathbf{0}^D(\mathbf{0}^D)^T) \end{pmatrix} \underbrace{\rightarrow}_{out\_proj=Identity} \begin{pmatrix} \boldsymbol{a}_1 & \dots & \boldsymbol{a}_N & \mathbf{0}^D \\ \boldsymbol{b}_1 & \dots & \boldsymbol{b}_N & 0 \\ \mathbf{0}^D & \dots & \mathbf{0}^D & \boldsymbol{x}_0 \\ \boldsymbol{b}_1\boldsymbol{a}_1 & \dots & \boldsymbol{b}_1\boldsymbol{a}_N & \mathbf{0}^D \\ flt(\boldsymbol{a}_1\boldsymbol{a}_1^T) & \dots & flt(\boldsymbol{a}_N\boldsymbol{a}_N^T) & flt(\mathbf{0}^D(\mathbf{0}^D)^T) \end{pmatrix}$$

Layer 2:

$$
\underbrace{\begin{pmatrix}
\boldsymbol{a}_1 & \ldots & \boldsymbol{a}_N & \boldsymbol{0}^D \\
\hline
\boldsymbol{b}_1 & \ldots & \boldsymbol{b}_N & 0 \\
\hline
\boldsymbol{0}^D & \ldots & \boldsymbol{0}^D & \boldsymbol{x}_0 \\
\hline
\leftarrow \sum_{i=1}^N \boldsymbol{b}_i \boldsymbol{a}_i \rightarrow \\
\hline
\leftarrow \sum_{i=1}^N flt(\boldsymbol{a}_i \boldsymbol{a}_i^T) \rightarrow
\end{pmatrix}}_{conv(in\_proj(\cdot))}
\odot
\underbrace{\begin{pmatrix}
\leftarrow \boldsymbol{1}^D \rightarrow \\
\hline
\leftarrow \boldsymbol{1} \rightarrow \\
\hline
\leftarrow \boldsymbol{1}^D \rightarrow \\
\hline
\leftarrow \boldsymbol{1}^D \rightarrow \\
\hline
\leftarrow \boldsymbol{1}^{D^2} \rightarrow
\end{pmatrix}}_{gate\_proj(\cdot)}
=
\begin{pmatrix}
\boldsymbol{a}_1 & \ldots & \boldsymbol{a}_N & \boldsymbol{0}^D \\
\hline
\boldsymbol{b}_1 & \ldots & \boldsymbol{b}_N & 0 \\
\hline
\boldsymbol{0}^D & \ldots & \boldsymbol{0}^D & \boldsymbol{x}_0 \\
\hline
\leftarrow \sum_{i=1}^N \boldsymbol{b}_i \boldsymbol{a}_i \rightarrow \\
\hline
\leftarrow \sum_{i=1}^N flt(\boldsymbol{a}_i \boldsymbol{a}_i^T) \rightarrow
\end{pmatrix}
$$

$$
\begin{pmatrix}
\boldsymbol{a}_1 & \ldots & \boldsymbol{a}_N & \boldsymbol{0}^D \\
\hline
\boldsymbol{b}_1 & \ldots & \boldsymbol{b}_N & 0 \\
\hline
\boldsymbol{0}^D & \ldots & \boldsymbol{0}^D & \boldsymbol{x}_0 \\
\hline
\leftarrow \sum_{i=1}^N \boldsymbol{b}_i \boldsymbol{a}_i \rightarrow \\
\hline
\leftarrow \sum_{i=1}^N flt(\boldsymbol{a}_i \boldsymbol{a}_i^T) \rightarrow
\end{pmatrix}
\underbrace{\rightarrow}_{out\_proj=Identity}
\begin{pmatrix}
\boldsymbol{a}_1 & \ldots & \boldsymbol{a}_N & \boldsymbol{0}^D \\
\hline
\boldsymbol{b}_1 & \ldots & \boldsymbol{b}_N & 0 \\
\hline
\boldsymbol{0}^D & \ldots & \boldsymbol{0}^D & \boldsymbol{x}_0 \\
\hline
\leftarrow \sum_{i=1}^N \boldsymbol{b}_i \boldsymbol{a}_i \rightarrow \\
\hline
\leftarrow \sum_{i=1}^N flt(\boldsymbol{a}_i \boldsymbol{a}_i^T) \rightarrow
\end{pmatrix}
$$

Now, we use a single BASECONV layer to implement a gradient descent update.

$$
\underbrace{\begin{pmatrix}
\boldsymbol{a}_1 & \ldots & \boldsymbol{a}_N & \boldsymbol{0}^D \\
\hline
\boldsymbol{b}_1 & \ldots & \boldsymbol{b}_N & 0 \\
\hline
\boldsymbol{0}^D & \ldots & \boldsymbol{0}^D & \boldsymbol{x}_0 \\
\hline
\boldsymbol{0}^D & \ldots & \boldsymbol{0}^D & \boldsymbol{1}^D \\
\hline
\leftarrow \sum_{i=1}^N \boldsymbol{b}_i \boldsymbol{a}_i \rightarrow \\
\hline
\leftarrow \sum_{i=1}^N flt(\boldsymbol{a}_i \boldsymbol{a}_i^T) \rightarrow \\
\hline
\leftarrow \sum_{i=1}^N \boldsymbol{b}_i \boldsymbol{a}_i \rightarrow \\
\hline
\leftarrow \sum_{i=1}^N flt(\boldsymbol{a}_i \boldsymbol{a}_i^T) \rightarrow
\end{pmatrix}}_{conv(in\_proj(\cdot))}
\odot
\underbrace{\begin{pmatrix}
\leftarrow \boldsymbol{1}^D \rightarrow \\
\hline
\leftarrow \boldsymbol{1} \rightarrow \\
\hline
\leftarrow \boldsymbol{1}^D \rightarrow \\
\hline
\leftarrow \boldsymbol{1}^D \rightarrow \\
\hline
\leftarrow \boldsymbol{1}^D \rightarrow \\
\hline
\leftarrow \boldsymbol{1}^{D^2} \rightarrow \\
\hline
\boldsymbol{0}^D & \ldots & \boldsymbol{0}^D & \boldsymbol{1}^D \\
\hline
\boldsymbol{0}^{D^2} & \ldots & \boldsymbol{0}^{D^2} & flt(\boldsymbol{1}^D \boldsymbol{x}_0^T)
\end{pmatrix}}_{gate\_proj(\cdot)}
=
\begin{pmatrix}
\boldsymbol{a}_1 & \ldots & \boldsymbol{a}_N & \boldsymbol{0}^D \\
\hline
\boldsymbol{b}_1 & \ldots & \boldsymbol{b}_N & 0 \\
\hline
\boldsymbol{0}^D & \ldots & \boldsymbol{0}^D & \boldsymbol{x}_0 \\
\hline
\boldsymbol{0}^D & \ldots & \boldsymbol{0}^D & \boldsymbol{1}^D \\
\hline
\leftarrow \sum_{i=1}^N \boldsymbol{b}_i \boldsymbol{a}_i \rightarrow \\
\hline
\leftarrow \sum_{i=1}^N flt(\boldsymbol{a}_i \boldsymbol{a}_i^T) \rightarrow \\
\hline
\boldsymbol{0}^D & \ldots & \boldsymbol{0}^D & \sum_{i=1}^N \boldsymbol{b}_i \boldsymbol{a}_i \\
\hline
\boldsymbol{0}^{D^2} & \ldots & \boldsymbol{0}^{D^2} & \sum_{i=1}^N flt(\boldsymbol{a}_i(\boldsymbol{a}_i \odot \boldsymbol{x}_0)^T)
\end{pmatrix}
$$

Note that the gradient

$$
\nabla_{\boldsymbol{x}} \mathcal{L}(\boldsymbol{x}_0) = \sum_{i=1}^N \boldsymbol{b}_i \boldsymbol{a}_i - \left( \sum_{i=1}^N \boldsymbol{a}_i \boldsymbol{a}_i^T \right) \boldsymbol{w_0}
$$

can be written as a linear combination of the vector

$$
\begin{pmatrix}
\sum_{i=1}^N \boldsymbol{b}_i \boldsymbol{a}_i \\
\hline
\sum_{i=1}^N flt(\boldsymbol{a}_i(\boldsymbol{a}_i \odot \boldsymbol{x}_0)^T)
\end{pmatrix}
$$

so we can write a weight construction for $out\_proj$ that updates $w_0 \to w_0 - \eta \nabla_{\boldsymbol{x}} \mathcal{L}(\boldsymbol{x}_0)$:

$$
\begin{pmatrix}
\boldsymbol{a}_1 & \cdots & \boldsymbol{a}_N & \boldsymbol{0}^D \\
\boldsymbol{b}_1 & \cdots & \boldsymbol{b}_N & 0 \\
\boldsymbol{0}^D & \cdots & \boldsymbol{0}^D & \boldsymbol{x}_0 \\
\boldsymbol{0}^D & \cdots & \boldsymbol{0}^D & \boldsymbol{1}^D \\
\leftarrow \sum_{i=1}^{N} \boldsymbol{b}_i \boldsymbol{a}_i \rightarrow \\
\leftarrow \sum_{i=1}^{N} flt(\boldsymbol{a}_i \boldsymbol{a}_i^T) \rightarrow \\
\boldsymbol{0}^D & \cdots & \boldsymbol{0}^D & \sum_{i=1}^{N} \boldsymbol{b}_i \boldsymbol{a}_i \\
\boldsymbol{0}^{D^2} & \cdots & \boldsymbol{0}^{D^2} & \sum_{i=1}^{N} flt(\boldsymbol{a}_i(\boldsymbol{a}_i \odot \boldsymbol{x}_0)^T)
\end{pmatrix}
\xrightarrow[out\_proj]{}
\begin{pmatrix}
\boldsymbol{a}_1 & \cdots & \boldsymbol{a}_N & \boldsymbol{0}^D \\
\boldsymbol{b}_1 & \cdots & \boldsymbol{b}_N & 0 \\
\boldsymbol{0}^D & \cdots & \boldsymbol{0}^D & \boldsymbol{x}_0 - \eta \nabla_{\boldsymbol{x}} \mathcal{L}(\boldsymbol{x}_0) \\
\boldsymbol{0}^D & \cdots & \boldsymbol{0}^D & \boldsymbol{1}^D \\
\leftarrow \sum_{i=1}^{N} \boldsymbol{b}_i \boldsymbol{a}_i \rightarrow \\
\leftarrow \sum_{i=1}^{N} flt(\boldsymbol{a}_i \boldsymbol{a}_i^T) \rightarrow \\
\boldsymbol{0}^D & \cdots & \boldsymbol{0}^D & \sum_{i=1}^{N} \boldsymbol{b}_i \boldsymbol{a}_i \\
\boldsymbol{0}^{D^2} & \cdots & \boldsymbol{0}^{D^2} & \sum_{i=1}^{N} flt(\boldsymbol{a}_i(\boldsymbol{a}_i \odot \boldsymbol{x}_0)^T)
\end{pmatrix}
$$

### D.3.2 LOWER BOUNDS: BASECONV CONSTRUCTIONS ARE ASYMPTOTICALLY OPTIMAL

Note that the non-causal weight construction in Appendix D.3.1 requires $O(1)$ layers and $O(D)$ state size, while the causal weight construction in Appendix D.3.1 requires $O(1)$ layers and $O(D^2)$ state size. Clearly the $O(D)$ state size requirement for non-causal models is tight, since one needs to store the gradient $\nabla_{\boldsymbol{x}} \mathcal{L} \in \mathbb{R}^D$. In this section, we prove that the $O(D^2)$ state size requirement for causal models is also asymptotically tight.

**Theorem D.33.** *Any single-pass (causal) algorithm computing the gradient*

$$
\nabla_{\boldsymbol{x}} \mathcal{L} = \sum_{j=1}^{N} b_j \boldsymbol{a}_j - \left( \sum_{j=1}^{N} \boldsymbol{a}_j \boldsymbol{a}_j^T \right) \boldsymbol{x}
$$

*given inputs $\{(\boldsymbol{a}_1, b_1), \ldots, (\boldsymbol{a}_N, b_N); \boldsymbol{x}\}$, with $(\boldsymbol{a}_i, b_i) \in \mathbb{R}^{(D+1)N}$ and $\boldsymbol{x} \in \mathbb{R}^D$, requires $\Omega(D^2)$ state size in the worst case, where $b_j \in \mathbb{R}$ and $\boldsymbol{a}_j, \boldsymbol{x} \in \mathbb{R}^D$.*

*Proof.* For simplicity, we pick $N = D$ for large enough $D$.

Since we can compute $\sum_{j=1}^{D} b_j \boldsymbol{a}_j$ in $O(D)$ space, we focus on computing the expensive $\left( \sum_{j=1}^{N} \boldsymbol{a}_j \boldsymbol{a}_j^T \right) \boldsymbol{x}$ term. Assume there exists a single-pass algorithm $\mathcal{A}$ that computes $\left( \sum_{j=1}^{N} \boldsymbol{a}_j \boldsymbol{a}_j^T \right) \boldsymbol{x}$ exactly for all choices of $\boldsymbol{a}_1, \ldots, \boldsymbol{a}_D, \boldsymbol{x} \in \mathbb{R}^D$. Now consider the following two claims:

1. Define $\boldsymbol{s}_D$ to be the state of the algorithm after seeing $\boldsymbol{a}_1, \ldots, \boldsymbol{a}_D$. Then we claim that $\boldsymbol{s}_D$ must have enough information to exactly reconstruct $\boldsymbol{M}_D := \sum_{j=1}^{D} \boldsymbol{a}_j \boldsymbol{a}_j^T$.

   This follows since the algorithm must be correct for any value $\boldsymbol{x} \in \mathbb{R}^D$ takes on. In particular, setting $\boldsymbol{x} = \boldsymbol{e}_i$ for $i \in [D]$, we observe that the algorithm must be able to exactly recover $\boldsymbol{M}_D \boldsymbol{e}_i = \boldsymbol{M}_D[:, i]$, $i \in [D]$.

2. The space of matrices

   $$
   \left\{ \sum_{j=1}^{D} \boldsymbol{a}_j \boldsymbol{a}_j^T \right\}
   $$

   over all choices of $\boldsymbol{a}_j \in \mathbb{R}^D$, $j \in [d]$ contains the set of all real symmetric matrices in $\mathbb{R}^{D \times D}$.

This holds since for any real symmetric matrix $\boldsymbol{A}$, we can obtain a set of possible $\boldsymbol{a}_j$'s via its eigendecomposition Strang (2012):

$$\boldsymbol{A} = \boldsymbol{Q}\boldsymbol{\Lambda}\boldsymbol{Q}^T = \sum_{j=1}^{D} \boldsymbol{a}_j\boldsymbol{a}_j^T$$

where $\boldsymbol{a}_j = \sqrt{\lambda_j}\boldsymbol{Q}[:,j]$.

From the first claim, we conclude that $\boldsymbol{s}_D$ must contain enough information to be able to recover $\boldsymbol{M}_D$ for any possible value $\boldsymbol{M}_D$ can take on (over all choices of $\boldsymbol{a}_1, \ldots, \boldsymbol{a}_D \in \mathbb{R}^D$). From the second claim, we have that the space of possible values of $\boldsymbol{M}_D$ includes the set of all possible real symmetric matrices. Since we know that this set requires $\frac{(D)(D+1)}{2}$ parameters to represent, we can conclude that $|\boldsymbol{s}_D| \geq \frac{(D)(D+1)}{2} \geq \Omega(D^2)$. $\qquad\square$

### D.4 BASECONV AND JACKSON'S THEOREM

In this section we prove BASECONV's ability to approximate arbitrary univariate and multivariate smooth functions.

#### D.4.1 UNIVARIATE FUNCTION APPROXIMATION

We start with a special case of smooth functions that apply entry-wise univariate smooth functions:

**Definition D.34.** Let $\overline{f} : [-1, 1] \to \mathbb{R}$ be a $(k, L)$-smooth univariate function. Then define

$$f : [-1, 1]^{N \times D} \to \mathbb{R}^{N \times D}$$

as follows. For all $0 \le i < N, 0 \le j < D$, and $\mathbf{u} \in [-1, 1]^{N \times D}$:

$$(f(\mathbf{u}))[i, j] = \overline{f}(\mathbf{u}[i, j]).$$

Now we will state a simple observation on BASECONV's ability to approximate these functions.

**Lemma D.35.** *For any smooth function $f$ as defined in Definition D.34, let $g(\mathbf{x}) = P_{\overline{f}}(\mathbf{x})$ with $P_{\overline{f}}$ being the polynomial from Corollary D.12. Then for all $\mathbf{x} \in [-1, 1]^{N \times D}$,*

$$\|g(\mathbf{x}) - f(\mathbf{x})\|_\infty \le \epsilon.$$

*Proof.* Follows from Definitions D.7 and D.34 and Corollary D.12. □

Next we will state a construction of an arithmetic circuit for a function that applies a univariate polynomial to all entries in $[-1, 1]^{N \times D}$:

**Lemma D.36.** *Let $P(X)$ be a degree $d$ univariate polynomial. Then there is a $(ND, O(ND), O(d), ND)$-circuit to compute $P(\mathbf{u})$ where $P(\mathbf{u})$ is defined as follows. For an input $\mathbf{u} \in [-1, 1]^{N \times D}$,*

$$P(\mathbf{u})[i, j] = P(\mathbf{u}[i, j]).$$

*Proof.* Let the univariate polynomial be

$$P(X) = \sum_{i=0}^{d} c_i X^i$$

where coefficients $c_i \in \mathbb{R}$.

Next we state the natural arithmetic circuit to compute $P(x)$ for $x \in \mathbb{R}$ in Algorithm 2:

---

**Algorithm 2** circuit $\mathcal{C}_P(x)$:

1: $s_0 \leftarrow c_0$
2: $m_0 \leftarrow 1$
3: **for** $j = 1, 2, \ldots, d$ **do**
4:     $m_j \leftarrow m_{j-1} \cdot x$        ▷ Multiplication gate
5:     $t_j \leftarrow c_j \cdot m_j$        ▷ Multiplication gate
6:     $s_j \leftarrow s_{j-1} + t_j$        ▷ Addition gate
7: **return** $s_d$        ▷ $s_d$ is the output gate

---

Next we apply the above circuit in parallel to form the circuit that computes $P(\mathbf{u})$ in Algorithm 3:

---

**Algorithm 3** Circuit for $P(\mathbf{u})$:

1: **for** $i = 0, 1, \ldots, N - 1$ **do**
2:     **for** $j = 0, 1, \ldots, D - 1$ **do**
3:         $\mathbf{z}[i, j] = \mathcal{C}_P(\mathbf{u}[i, j])$        ▷ Do this in parallel
4: **return** $\mathbf{z}$        ▷ $\mathbf{z}$ is the output matrix

---

Looking at Algorithm 2, the depth of the circuit is $3d$, or $O(d)$, since that is the bound on iterations of the for loop, and each iteration we compute 3 sequential operations. Therefore it's a $(1, O(d), O(d, O(1))$-*circuit*.

For Algorithm 3, The width is $O(ND)$, since we have our input of size $N \times D$, which goes through the circuit in parallel, as stated in Algorithm 3. Therefore we have an $(ND, O(ND), O(d), O(ND))$-*circuit* that computes $P(\mathbf{u})$. □

Since BASECONV has the ability to represent any arithmetic circuit, we get the following:

**Corollary D.37.** *We can implement $P(\mathbf{u})$ (where $P(\mathbf{u})$ is as defined in Lemma D.36 ) when* $\deg(P) = d$ *with a* $(N, O(d \log(ND)), D, O(ND), D) - $ BASECONV.

*Proof.* Follows from Lemma D.36 giving us the $(ND, O(ND), O(d), O(ND))$-*circuit* for an arbitrary polynomial and Theorem D.30 gives us the BASECONV model to implement the circuit. □

We will prove a tighter bound showing we can represent $P(\mathbf{u})$ using a constant number of BASECONV layers (for constant $\deg(P)$):

**Theorem D.38.** *We can implement $P(\mathbf{u})$ when* $\deg(P) = d$ *with an* $(O(N), O(d), D, O(N), D) - $ BASECONV *model.*

*Proof.* We will convert the steps done in Algorithm 2 to layers of BASECONV. Since Algorithm 3 is essentially running Algorithm 2 in parallel over all entries of input $\mathbf{u} \in [-1, 1]^{N \times D}$, the latter happens automatically in our BASECONV implementation.

For this proof, define
$$P_j(X) = X^j$$
and let $\boldsymbol{C}_i$ be the matrix of size $N \times D$ and all the entries are $c_i$.

We expand the input to our BASECONV layers as follows,
$$\mathbf{u} = \begin{pmatrix} \mathbf{u}' \\ \mathbf{0}^{3N \times D} \end{pmatrix}.$$

This means that the size of the internal dimension of our BASECONV layers will be $(4N, D)$.

To begin iterations of the for loop we need to store initial values into the extra space in $\mathbf{u}$. Taking us from
$$\mathbf{u} = \begin{pmatrix} \mathbf{u}' \\ \mathbf{0}^{N \times D} \\ \mathbf{0}^{N \times D} \\ \mathbf{0}^{N \times D} \end{pmatrix} \rightarrow \begin{pmatrix} \mathbf{u} \\ \mathbf{1}^{N \times D} \\ \mathbf{1}^{N \times D} \\ \boldsymbol{C}_0 \end{pmatrix} =: \mathbf{u}_0$$

We do this via $\text{BASECONV}(\mathbf{u}', \boldsymbol{I}^{D \times D}, \begin{pmatrix} \mathbf{0}^{N \times D} \\ \mathbf{1}^{N \times D} \\ \mathbf{1}^{N \times D} \\ \boldsymbol{C}_0 \end{pmatrix}, \mathbf{0}^{4N \times D}, \mathbf{1}^{4N \times D})$ which computes

$$\left( \left( \begin{pmatrix} \mathbf{u} \\ \mathbf{0}^{N \times D} \\ \mathbf{0}^{N \times D} \\ \mathbf{0}^{N \times D} \end{pmatrix} \boldsymbol{I}^{D \times D} + \begin{pmatrix} \mathbf{0}^{N \times D} \\ \mathbf{1}^{N \times D} \\ \mathbf{1}^{N \times D} \\ \boldsymbol{C}_0 \end{pmatrix} \right) \odot \left( \mathbf{0}^{4N \times D} * \begin{pmatrix} \mathbf{u} \\ \mathbf{0}^{N \times D} \\ \mathbf{0}^{N \times D} \\ \mathbf{0}^{N \times D} \end{pmatrix} + \mathbf{1}^{4N \times D} \right) \right).$$

The above simplifies to
$$\left( \left( \begin{pmatrix} \mathbf{u} \\ \mathbf{0}^{N \times D} \\ \mathbf{0}^{N \times D} \\ \mathbf{0}^{N \times D} \end{pmatrix} + \begin{pmatrix} \mathbf{0}^{N \times D} \\ \mathbf{1}^{N \times D} \\ \mathbf{1}^{N \times D} \\ \boldsymbol{C}_0 \end{pmatrix} \right) \odot \left( \mathbf{1}^{4N \times D} \right), \right.$$

which gives us
$$\begin{pmatrix} \mathbf{u} \\ \mathbf{1}^{N \times D} \\ \mathbf{1}^{N \times D} \\ \boldsymbol{C}_0 \end{pmatrix} =: \mathbf{u}_0,$$

as desired

This was done with a $(4N, 1, D, 4N, D) - \text{BASECONV}$ layer.

Our goal is, at the end of iteration $j$ to compute $\mathbf{u}_j \in \mathbb{R}^{4N \times D}$ such that,

$$\mathbf{u}_j = \left( \begin{array}{c} \mathbf{u} \\ \hline P_j(\mathbf{u}) \\ \hline \boldsymbol{C}_j \odot P_j(\mathbf{u}) \\ \hline \boldsymbol{C}_0 + \boldsymbol{C}_1 \odot P_1(\mathbf{u}) + \cdots + \boldsymbol{C}_j \odot P_j(\mathbf{u}) \end{array} \right).$$

We will view the above matrix in terms of the variables in the Algorithm 2 as follows

$$\left( \begin{array}{c} \mathbf{u} \\ \hline P_j(\mathbf{u}) \\ \hline \boldsymbol{C}_j \odot P_j(\mathbf{u}) \\ \hline \boldsymbol{C}_0 + \boldsymbol{C}_1 \odot P_1(\mathbf{u}) + \cdots + \boldsymbol{C}_j \odot \mathbf{u}^j \end{array} \right) =: \left( \begin{array}{c} \mathbf{u} \\ \hline \mathbf{m}_j \\ \hline \mathbf{t}_j \\ \hline \mathbf{s}_j \end{array} \right).$$

The for loop runs for values of $1 \leq j \leq d$ which the remainder of this proof will replicate. There are three lines in the for loop in Algorithm 2 which we will cover how these operations happen in constant number of BASECONV layers.

In line 4, the first line in the for loop computes

$$\mathbf{u}_{j-1} = \left( \begin{array}{c} \mathbf{u} \\ \hline \mathbf{m}_{j-1} \\ \hline \mathbf{t}_{j-1} \\ \hline \mathbf{s}_{j-1} \end{array} \right) \rightarrow \left( \begin{array}{c} \mathbf{u} \\ \hline \mathbf{m}_j \\ \hline \mathbf{t}_{j-1} \\ \hline \mathbf{s}_{j-1} \end{array} \right) =: \mathbf{u}_j^{(1)}.$$

Note that $\mathbf{m}_j = \mathbf{m}_{j-1} \odot \mathbf{u}$.

We use the `remember` primitive to compute $\mathbf{u}_j^{(1)}$ from $\mathbf{u}_{j-1}$. Define $f : \mathbb{R}^{2N \times D} \rightarrow \mathbb{R}^{2N \times D}$ as follows

$$f \left( \begin{array}{c} \mathbf{u} \\ \mathbf{m}_{j-1} \end{array} \right) = \left( \begin{array}{c} \mathbf{u} \\ \mathbf{m}_{j-1} \odot \mathbf{u} \end{array} \right).$$

If we can compute $f$ with BASECONV layers then we can compute $\mathbf{u}_j^{(1)}$ for $\mathbf{u}_{j-1}$ by calling `remember`$(\mathbf{u}_j, 0, 2N - 1, f)$.

We show $\text{BASECONV} \left( \left( \begin{array}{c} \mathbf{u} \\ \mathbf{m}_j \end{array} \right), \boldsymbol{I}^{D \times D}, \mathbf{0}^{2N \times D}, \boldsymbol{H}, \left( \begin{array}{c} \mathbf{1}^{N \times D} \\ \mathbf{0}^{N \times D} \end{array} \right) \right)$ maps

$$\left( \begin{array}{c} \mathbf{u} \\ \mathbf{m}_{j-1} \end{array} \right) \rightarrow \left( \begin{array}{c} \mathbf{u} \\ \mathbf{m}_j \end{array} \right),$$

where $\boldsymbol{H}$ is defined as in Proposition D.24. We plug the matrices into the BASECONV layer as follows:

$$\left( \left( \begin{array}{c} \mathbf{u} \\ \mathbf{m}_{j-1} \end{array} \right) \cdot \boldsymbol{I}^{D \times D} + \mathbf{0}^{2N \times D} \right) \odot \left( \boldsymbol{H} * \left( \begin{array}{c} \mathbf{u} \\ \mathbf{m}_{j-1} \end{array} \right) + \left( \begin{array}{c} \mathbf{1}^{N \times D} \\ \mathbf{0}^{N \times D} \end{array} \right) \right).$$

We know from Proposition D.24 that this convolution operation is a shift down by $N$ rows. Therefore the above simplifies to

$$\left( \left( \begin{array}{c} \mathbf{u} \\ \mathbf{m}_{j-1} \end{array} \right) \cdot \boldsymbol{I}^{D \times D} + \mathbf{0}^{2N \times D} \right) \odot \left( \left( \begin{array}{c} \mathbf{0}^{N \times D} \\ \mathbf{u} \end{array} \right) + \left( \begin{array}{c} \mathbf{1}^{N \times D} \\ \mathbf{0}^{N \times D} \end{array} \right) \right),$$

which simplifies to

$$\left( \begin{array}{c} \mathbf{u} \\ \mathbf{m}_{j-1} \end{array} \right) \odot \left( \begin{array}{c} \mathbf{1}^{N \times D} \\ \mathbf{u} \end{array} \right) = \left( \begin{array}{c} \mathbf{u} \\ \mathbf{m}_{j-1} \odot \mathbf{u} \end{array} \right) = f \left( \begin{array}{c} \mathbf{u} \\ \mathbf{m}_j \end{array} \right),$$

as desired. Therefore by Proposition D.26, line 4 can be computed by $(4N, 8, D, 4N, D) - \text{BASECONV}$.

For line 5 of the for loop we need to compute

$$\mathbf{u}_j^{(1)} = \left(\frac{\begin{array}{c}\mathbf{u}\\\hline\mathbf{m}_j\\\hline\mathbf{t}_{j-1}\\\hline\mathbf{s}_{j-1}\end{array}}{}\right) \rightarrow \left(\frac{\begin{array}{c}\mathbf{u}\\\hline\mathbf{m}_j\\\hline\mathbf{t}_j\\\hline\mathbf{s}_{j-1}\end{array}}{}\right) =: \mathbf{u}_j^{(2)}.$$

Note that $\mathbf{t}_j = \boldsymbol{C}_j \odot \mathbf{m}_j$.

To do this we will use three BASECONV layers. We use the `remember` primitive to compute $\mathbf{u}_j^{(2)}$ from $\mathbf{u}_j^{(1)}$. Define $g : \mathbb{R}^{2N \times D} \rightarrow \mathbb{R}^{2N \times D}$ as follows,

$$g\begin{pmatrix}\mathbf{m}_j\\\mathbf{t}_{j-1}\end{pmatrix} = \begin{pmatrix}\mathbf{m}_j\\\boldsymbol{C}_j \odot \mathbf{m}_j\end{pmatrix}.$$

If we can compute $g$ with BASECONV layers then we can compute $\mathbf{u}_j^{(2)}$ for $\mathbf{u}_{j-1}$ by calling `remember`$(\mathbf{u}_j^{(1)}, N, 3N - 1, g)$.

Indeed, we show the $g$ can be computed by first computing BASECONV $\left(\begin{pmatrix}\mathbf{m}_j\\\mathbf{t}_{j-1}\end{pmatrix}, \boldsymbol{I}^{D\times D}, \mathbf{0}^{2N\times D}, \mathbf{0}^{2N\times D}, \begin{pmatrix}\mathbf{1}^{N\times D}\\\mathbf{0}^{N\times D}\end{pmatrix}\right)$:

$$\left(\left(\begin{pmatrix}\mathbf{m}_j\\\mathbf{t}_{j-1}\end{pmatrix} \cdot \boldsymbol{I}^{D\times D} + \mathbf{0}^{2N\times D}\right) \odot \left(\mathbf{0}^{2N\times D} * \begin{pmatrix}\mathbf{m}_j\\\mathbf{t}_{j-1}\end{pmatrix} + \begin{pmatrix}\mathbf{1}^{N\times D}\\\mathbf{0}^{N\times D}\end{pmatrix}\right)\right),$$

which simplifies to

$$\left(\begin{pmatrix}\mathbf{m}_j\\\mathbf{t}_{j-1}\end{pmatrix}\right) \odot \left(\begin{pmatrix}\mathbf{1}^{N\times D}\\\mathbf{0}^{N\times D}\end{pmatrix}\right).$$

This results in

$$\begin{pmatrix}\mathbf{m}_j\\\mathbf{0}^{N\times D}\end{pmatrix}.$$

We pass into the next layer, BASECONV $\left(\begin{pmatrix}\mathbf{m}_j\\\mathbf{0}^{N\times D}\end{pmatrix}, \boldsymbol{I}^{D\times D}, \begin{pmatrix}\mathbf{0}^{N\times D}\\\mathbf{1}^{N\times D}\end{pmatrix}, \boldsymbol{H}, \begin{pmatrix}\mathbf{1}^{N\times D}\\\mathbf{0}^{N\times D}\end{pmatrix}\right)$ where $\boldsymbol{H}$ is defined as in Proposition D.24:

$$\left(\left(\begin{pmatrix}\mathbf{m}_j\\\mathbf{0}^{N\times D}\end{pmatrix} \cdot \boldsymbol{I}^{D\times D} + \begin{pmatrix}\mathbf{0}^{N\times D}\\\mathbf{1}^{N\times D}\end{pmatrix}\right) \odot \left(\boldsymbol{H} * \begin{pmatrix}\mathbf{m}_j\\\mathbf{0}^{N\times D}\end{pmatrix} + \begin{pmatrix}\mathbf{1}^{N\times D}\\\mathbf{0}^{N\times D}\end{pmatrix}\right)\right).$$

Since the kernel $\boldsymbol{H}$ is as in Proposition D.24, this simplifies to

$$\left(\begin{pmatrix}\mathbf{m}_j\\\mathbf{1}^{N\times D}\end{pmatrix} \odot \left(\begin{pmatrix}\mathbf{0}^{N\times D}\\\mathbf{m}_j\end{pmatrix} + \begin{pmatrix}\mathbf{1}^{N\times D}\\\mathbf{0}^{N\times D}\end{pmatrix}\right)\right).$$

The above simplifies further to

$$\begin{pmatrix}\mathbf{m}_j\\\mathbf{1}^{N\times D}\end{pmatrix} \odot \begin{pmatrix}\mathbf{1}^{N\times D}\\\mathbf{m}_j\end{pmatrix},$$

which results in:

$$\begin{pmatrix}\mathbf{m}_j\\\mathbf{m}_j\end{pmatrix}.$$

We pass the above to BASECONV $\left(\begin{pmatrix}\mathbf{m}_j\\\mathbf{m}_j\end{pmatrix}, \boldsymbol{I}^{D\times D}, \mathbf{0}^{2N\times D}, \mathbf{0}^{2N\times D}, \begin{pmatrix}\mathbf{1}^{N\times D}\\\boldsymbol{C}_j\end{pmatrix}\right)$:

$$\left(\left(\begin{pmatrix}\mathbf{m}_j\\\mathbf{m}_j\end{pmatrix} \cdot \boldsymbol{I}^{D\times D} + \mathbf{0}^{2N\times D}\right) \odot \left(\mathbf{0}^{2N\times D} * \begin{pmatrix}\mathbf{m}_j\\\mathbf{m}_j\end{pmatrix} + \begin{pmatrix}\mathbf{1}^{N\times D}\\\boldsymbol{C}_j\end{pmatrix}\right)\right)$$

which simplifies to

$$\begin{pmatrix}\mathbf{m}_j\\\mathbf{m}_j\end{pmatrix} \odot \begin{pmatrix}\mathbf{1}^{N\times D}\\\boldsymbol{C}_j\end{pmatrix}.$$

The above results in

$$\begin{pmatrix}\mathbf{m}_j\\\boldsymbol{C}_j \odot \mathbf{m}_j\end{pmatrix} = g\begin{pmatrix}\mathbf{m}_j\\\mathbf{t}_{j-1}\end{pmatrix},$$

as desired.

Therefore by Corollary D.27, line 5 was computed by $(4N, O(1), D, 4N, D) - \text{BASECONV}$.

For line 6, the final line of the for loop, we want

$$\mathbf{u}_j^{(2)} = \begin{pmatrix} \mathbf{u} \\ \hline \mathbf{m}_j \\ \hline \mathbf{t}_j \\ \hline \mathbf{s}_{j-1} \end{pmatrix} \rightarrow \begin{pmatrix} \mathbf{u} \\ \hline \mathbf{m}_j \\ \hline \mathbf{t}_j \\ \hline \mathbf{s}_j \end{pmatrix} =: \mathbf{u}_j.$$

Note that $\mathbf{s}_j = \mathbf{s}_{j-1} + \mathbf{t}_j$

Define function $h : \mathbb{R}^{2N \times D} \rightarrow \mathbb{R}^{2N \times D}$ as follows,

$$h \begin{pmatrix} \mathbf{t}_j \\ \mathbf{s}_{j-1} \end{pmatrix} = \begin{pmatrix} \mathbf{t}_j \\ \mathbf{s}_{j-1} + \mathbf{t}_j \end{pmatrix}.$$

If we can compute $h$ with BASECONV layers then we can compute $\mathbf{u}_j$ for $\mathbf{u}_{j-1}$ by calling `remember`$(\mathbf{u}_j^{(2)}, 2N, 4N-1, h)$.

Indeed we show that $h$ can be computed by computing $\text{BASECONV}\left( \begin{pmatrix} \mathbf{t}_j \\ \mathbf{s}_{j-1} \end{pmatrix}, \mathbf{0}^{D \times D}, \mathbf{1}^{2N \times D}, \overline{\mathbf{H}}, \mathbf{0}^{2N \times D} \right)$, where kernel $\overline{\mathbf{H}} \in \mathbb{R}^{2N \times D}$ is defined as:

$$\overline{\mathbf{H}}[k, :] \equiv \begin{cases} \mathbf{1}^D & \text{if } k \in \{0, N\} \\ \mathbf{0}^D & \text{otherwise.} \end{cases}$$

.

This layer computes

$$\left( \begin{pmatrix} \mathbf{t}_j \\ \mathbf{s}_{j-1} \end{pmatrix} \cdot \mathbf{0}^{2N \times D} + \mathbf{1}^{2N \times D} \right) \odot \left( \overline{\mathbf{H}} * \begin{pmatrix} \mathbf{t}_j \\ \mathbf{s}_{j-1} \end{pmatrix} + \mathbf{0}^{2N \times D} \right).$$

This simplifies to

$$\left( \mathbf{1}^{2N \times D} \right) \odot \left( \overline{\mathbf{H}} * \begin{pmatrix} \mathbf{t}_j \\ \mathbf{s}_{j-1} \end{pmatrix} \right) = \left( \overline{\mathbf{H}} * \begin{pmatrix} \mathbf{t}_j \\ \mathbf{s}_{j-1} \end{pmatrix} \right).$$

Now we compute this convolution for column $i$, $0 \le i < 2N$. For notational convenience, let $\begin{pmatrix} \mathbf{t}_j \\ \mathbf{s}_{j-1} \end{pmatrix}$ be noted as matrix $\mathbf{V}$. Then we have:

$$\overline{\mathbf{H}}[:, i] * \mathbf{V}[:, i] = \text{coeff}\left( (1 + X^N) \mathbf{V}[:, i](X) \mod X^{2N} \right),$$

where $(1 + X^N)$ is the polynomial representation of the columns of $\overline{\mathbf{H}}$ (since there's a one in the 0th index and a one in the $N$th index of each column).

The expression simplifies to

$$\text{coeff} \mathbf{V}[:, i](X) + \mathbf{V}[:, i](X) X^N \mod X^{2N},$$

which can be broken down to

$$\text{coeff} \left( \left( \mathbf{V}[0][i] + \mathbf{V}[1][i]X + \cdots + \mathbf{V}[2N-1][i]X^{2N-1} \right) \mod X^{2N} \right)$$
$$+ \text{coeff} \left( \left( \mathbf{V}[0][i]X^N + \mathbf{V}[1][i]X^{N+1} + \cdots + \mathbf{V}[2N-1][i]X^{3N-1} \right) \mod X^{2N} \right)$$

with the lower order terms in the second coefficient vector being zeros,

$$\text{coeff} \left( \left( \mathbf{V}[0][i] + \mathbf{V}[1][i]X + \cdots + \mathbf{V}[2N-1][i]X^{2N-1} \right) \mod X^{2N} \right)$$
$$+ \text{coeff} \left( \left( 0 + 0X + \cdots + 0X^{N-1} + \mathbf{V}[0][i]X^N + \cdots + \mathbf{V}[2N-1][i]X^{3N-1} \right) \mod X^{2N} \right)$$

After taking $\mod X^{2N}$ we get

$$\text{coeff} \left( \mathbf{V}[0][i] + \mathbf{V}[1][i]X + \cdots + \mathbf{V}[2N-1][i]X^{2N-1} \right)$$
$$+ \text{coeff} \left( 0 + 0X + \cdots 0X^{N-1} \mathbf{V}[0][i]X^N + \cdots \mathbf{V}[N-1][i]X^{2N-1} \right)$$

The first set of coefficients is the input matrix as is. And the second one is the input matrix shifted down as seen in Proposition D.24. Therefore when we add these vectors we are doing

$$\begin{pmatrix} \mathbf{t}_j \\ \mathbf{s}_{j-1} \end{pmatrix} + \begin{pmatrix} \mathbf{0}^{N \times D} \\ \mathbf{t}_j \end{pmatrix} = h \begin{pmatrix} \mathbf{t}_j \\ \mathbf{s}_{j-1} \end{pmatrix},$$

as desired. Therefore by Proposition D.26, line 6 is computed with by $(4N, 1, D, 4N, D) -$ BASECONV.

The $\mathbf{s}_d$ matrix gives us $\boldsymbol{C}_0 + \boldsymbol{C}_1 \odot \mathbf{m}_1 + \cdots + \boldsymbol{C}_d \odot \mathbf{m}_d$. Recalling that

$$\boldsymbol{C}_0 + \boldsymbol{C}_1 \odot \mathbf{m}_1 + \cdots + \boldsymbol{C}_d \odot \mathbf{m}_d \equiv \sum_{j=0}^{d} \boldsymbol{C}_j \odot \mathbf{u}^j = P(\mathbf{u}),$$

and hence $\mathbf{s}_d$ is our desired output.

We have $d$ layers, each consisting of $O(1)$ BASECONV layers. Giving us $O(d)$ many layers to implement Algorithm 2.

Therefore, via the ability to stack BASECONV layers to do function composition, the for loop was computed by a $(4N, O(d), D, 4N, D) -$ BASECONV , as desired. $\quad\square$

The following states BASECONV's ability to approximate a univariate smooth function:

**Proposition D.39.** *Let $f$ be the $(k, L)$ -smooth function defined in Definition D.34. Then there is a* $\left(N, O\left(\sqrt[k]{\frac{L}{\epsilon}}\right) + k, D, (ND), D\right) -$ BASECONV *model that approximates $f$ within error $\epsilon$.*

*Proof.* Follows from Corollary D.12, Lemma D.35, and Theorem D.38. $\quad\square$

### D.4.2 MULTIVARIATE FUNCTION APPROXIMATION

We consider the following multivariate functions:

**Definition D.40.** For $0 \leq 1 < N, 0 \leq j < D$, let $\bar{f}_{i,j} : [-1, 1]^{N \times D} \to \mathbb{R}$ be a $(k, L)$-smooth multivariate function. Then define

$$f(\mathbf{x}) : [-1, 1]^{N \times D} \to \mathbb{R}^{N \times D}$$

as follows. For all $0 \leq i < N, 0 \leq j < D, \mathbf{u} \in [-1, 1]^{N \times D}$ define

$$f(\mathbf{u})[i, j] := \bar{f}_{i,j}(\mathbf{u}).$$

**Lemma D.41.** *For any smooth function $f$ as defined in Definition D.40, let $g(X_1, \ldots, X_{N \times D}) = P_{\bar{f}}(X_1, \ldots, X_{N \times D})$ be the polynomial from Corollary D.14. Then for all $\mathbf{x} \in [-1, 1]^{N \times D}$,*

$$\|g(\mathbf{x}) - f(\mathbf{x})\|_\infty \leq \epsilon.$$

*Proof.* Follows from Definitions D.7 and D.40 and Corollary D.14. $\quad\square$

Next we will state a construction for an arithmetic circuit for a function that takes a $[-1, 1]^{N \times D}$ variable input:

**Lemma D.42.** *Let $P(\boldsymbol{X})$ be a degree $d$ multivariate polynomial. Then there is a* $\left(n, O(d \cdot n^d), O(d \log(n)), O(n^d)\right)$-circuit *to compute $P(\mathbf{u})$ on any input $\mathbf{u} \in [-1, 1]^n$.*

*Proof.* Let the multivariate polynomial be as defined in Definition D.6. We build the circuit to compute this in Algorithm 4,

---

**Algorithm 4** circuit $\mathcal{C}_P(\mathbf{x})$:

1: **for** $\boldsymbol{\alpha} = (\alpha_1, \ldots, \alpha_n) \in \mathbb{Z}_{\geq 0}^n$ such that $\sum_{i=1}^n \alpha_i \leq d$ **do**
2:      $m_{\boldsymbol{\alpha}} \leftarrow 1$
3:      **for** $i = 1, 2, \ldots, n$ **do**                                  ▷ Done in parallel
4:          **if** $\alpha_i \neq 0$ **then**
5:              $m_{\boldsymbol{\alpha}} \leftarrow m_{\boldsymbol{\alpha}} \cdot x_i^{\alpha_i}$
6:      $t_{\boldsymbol{\alpha}} \leftarrow c_{\boldsymbol{\alpha}} \cdot m_{\boldsymbol{\alpha}}$
7: **for** $\boldsymbol{\alpha} = (\alpha_1, \ldots, \alpha_n) \in \mathbb{Z}_{\geq 0}^n$ such that $\sum_{i=1}^n \alpha_i \leq d$ **do**
8:      $s \leftarrow \sum t_{\boldsymbol{\alpha}}$                                       ▷ Done in parallel
9: **return** s

---

We compute the for loop starting on line 3 by making multiplications in parallel. Therefore obtaining a depth of $O(\log(d))$. We also have the for loop starting on line 7, making pairwise addition operations, resulting in a depth of $O(d \log(n))$.      $\square$

We again use the result that BASECONV can represent any arithmetic circuit to get:

**Corollary D.43.** *We can implement $P(\mathbf{u})$ (where $P(\mathbf{u})$ is as defined in Lemma D.42) when* $\deg(P(X_1, \ldots, X_{ND})) = d$ *with a* $\big(N, O(d \log(ND)), D, O((ND)^d), D\big) - $ BASECONV *where* $\mathbf{u} \in [-1, 1]^{N \times D}$.

*Proof.* Lemma D.42 gives us the arithetmic circuit that computes this polynomial. Then via Theorem D.30 we get a $\big(N, O(d \log(ND)), D, O((ND)^d), D\big) - $ BASECONV model to implement the circuit.      $\square$

Finally we state BASECONV's ability to approximate multivariate smooth functions:

**Proposition D.44.** *Let $f$ be the function defined in Definition D.40. Then there is a* $\big(N, O(d \log(ND)), D, O((ND)^d), D\big) - $ BASECONV *model that approximates $f$ to within error $\epsilon$, with* $d = O_k\big(\sqrt[k]{\frac{NDL}{\epsilon}}\big)$.

*Proof.* We get the existence of a polynomial that approximates $f$ for some $\epsilon$ from Corollary D.14. Then via Corollary D.43 we get that we can represent any polynomial, implying $\big(N, O(d \log(ND)), D, O((ND)^d), D\big) - $ BASECONV represents any polynomial that approximates the multivariate smooth function $f$.      $\square$

## D.5 ZERO POPULATION GRADIENT BASECONV ON PRIMITIVES RECOVERS EXACT SOLUTION

In this section, we prove we can recover the functions LINEAR and MULTIPLY exactly given the expected gradients of their respective loss functions being 0 along with some necessary assumptions.

### D.5.1 NOTATION

We start by defining additional notation for this subsection.

For readability, we will redefine how we index an entry of a 2 dimensional matrix - note that we are using 0 indexing $[N] = \{0, 1, \ldots, N - 1\}$. For an entry of matrix $A[i, j]$ where $i$ is the row number and $j$ is the column number, we denote it as $A_{i,j}$. Now recall our BASECONV layer in 18, we will define the parameters of the layer as follows. We have the weight matrix, $W = \{W_{i,j}\} \in \mathbb{R}^{d \times d}$, the kernel matrix, $K = \{K_{i,j}\} \in \mathbb{R}^{N \times d}$, the first bias matrix $B^{(1)} = \{B^{(1)}_{i,j}\} \in \mathbb{R}^{N \times d}$ and the second bias matrix, $B^{(2)} = \{B^{(2)}_{i,j}\} \in \mathbb{R}^{N \times d}$. We denote the array of these parameters as $\theta = (W, K, B^{(1)}, B^{(2)})$. Therefore we have the BASECONV layer operation,

$$Z = \text{BASECONV}(\theta, \mathbf{u}, c, d_{out}) \stackrel{\text{def}}{=} \left(\mathbf{u}W + B^{(1)}\right) \odot \left(K * \mathbf{u} + B^{(2)}\right)[:, c : c + d_{out} - 1] \quad (48)$$

for some integers $d_{out} \in [d]$ (with $d_{out} \neq 0$) and $c \in [d - d_{out}]$ that we use to truncate columns of the output layer to match function input and output size as stated below.

Recall that $\mathbf{u}$ is the input to a BASECONV layer, $\mathbf{u} = \{\mathbf{u}_{i,j}\} \in \mathbb{R}^{N \times d}$.

Moving onto our target function:
$$f : \mathbb{R}^{N \times d} \to \mathbb{R}^{N \times d_{out}}.$$

Naturally, for $i \in [N]$ and $j \in [d_{out}]$, we'll denote $(f(\mathbf{u}))[i, j]$ by $f(\mathbf{u})_{i,j}$.

Next, we define the training input distribution.

### D.5.2 TRAINING INPUT DISTRIBUTION

1. Let $\Delta$ be the training distribution on $\mathbb{R}^{N \times d}$ such that:

**Assumption D.45.** Given a monomial, $\Pi_k \left(\mathbf{u}_{i_k, j_k}\right)^{m_k}$, if $m_k$ is odd for some $k$ then $\mathbb{E}\left[\Pi_k \mathbf{u}_{i_k, j_k}^{m_k}\right] = 0$. Otherwise, $\mathbb{E}\left[\Pi_k \left(\mathbf{u}_{i_k, j_k}\right)^{m_k}\right] > 0$.

**Assumption D.46.** Assume that the training data is generated as

- $\mathbf{u} \sim \Delta$ as input

- Output is $\mathbf{y} = f(\mathbf{u}) + \mathcal{E}$ where $\mathcal{E} = \{\mathcal{E}_{i,j}\} \in \mathbb{R}^{N \times d_{out}}$ is the random error matrix such that

  - The distributions on $\mathcal{E}$ and $\Delta$ are independent. (Call the distribution on $\mathcal{E}$ to be $\Delta_\mathcal{E}$)
  - $\mathbb{E}[\mathcal{E}_{i,j}] = 0$ for all $(i, j) \in [N] \times [d_{out}]$

**Loss function**

- Define for $i \in [N]$ and $j \in [d_{out}]$

$$\overline{L_{i,j}}(\mathbf{u}, \theta, \mathcal{E}) = (Z_{i,j} - \mathbf{y}_{i,j})^2 = \left(Z_{i,j} - f(\mathbf{u})_{i,j} - \mathcal{E}_{i,j}\right)^2 \quad (49)$$

- $L(\mathbf{u}) = \sum_{i=0}^{N-1} \sum_{j=0}^{d_{out}-1} \overline{L_{i,j}}(\mathbf{u}, \theta, \mathcal{E})$

- Training loss, $\overline{L^{(t)}}(\theta) = \overline{L^{(t)}} = \mathbb{E}_{\substack{\mathbf{u} \sim \Delta \\ \mathcal{E} \sim \Delta_\mathcal{E}}} [L(\mathbf{u})]$

- $\nabla_\theta \overline{L^{(t)}}(\theta) = \mathbb{E}_{\mathbf{u}, \mathcal{E}} \sum_{i=0}^{N-1} \sum_{j=0}^{d_{out}-1} \nabla_\theta \overline{L}_{i,j}(\mathbf{u})$

**The Goal** Given a target function $f$, what can we infer for $\boldsymbol{\theta} = \left(\boldsymbol{W}, \boldsymbol{K}, \boldsymbol{B}^{(1)}, \boldsymbol{B}^{(2)}\right)$ from $\nabla_{\boldsymbol{\theta}} \overline{L}(\boldsymbol{\theta}) = \mathbf{0}$?

1. Ideally, we would like to assume that $f$ can be represented exactly by 1-layer BASECONV.

2. For now, let's assume that $f(\mathbf{u})_{i,j}$ only depends on $\mathbf{u}_{i,:}$

   This includes as special cases:

   - $f(\mathbf{u}) = \mathbf{u}_{:,a:a+d_{out}-1} \odot \mathbf{u}_{:,b:b+d_{out}-1}$ for some integers $a, b \in [d_{out}]$
   - $f(\mathbf{u}) = \mathbf{u} \cdot \overline{\boldsymbol{W}}$ for $\overline{\boldsymbol{W}} \in \mathbb{R}^{d \times d}$

We want to prove that when the gradients of the expected loss function are 0, then the set of parameters that satisfy the condition perform exactly these functions.

### D.5.3 A GENERIC PARTIAL DERIVATIVE

Let's try and reason as much as we can for a generic partial derivative. Let $x \in \boldsymbol{\theta} = \left(\boldsymbol{W}, \boldsymbol{K}, \boldsymbol{B}^{(1)}, \boldsymbol{B}^{(2)}\right)$. Then from Equation (49), we have that for any $(i, j) \in [N] \times [d_{out}]$:

$$
\begin{aligned}
\frac{\partial \overline{L}_{i,j}}{\partial x} &= 2\left(\boldsymbol{Z}_{i,j} - f(\mathbf{u})_{i,j} - \mathcal{E}_{i,j}\right) \frac{\partial \boldsymbol{Z}_{i,j}}{\partial x} \\
&= 2\left(\boldsymbol{T}_{i,j}^{(1)} - \boldsymbol{T}_{i,j}^{(2)} - \boldsymbol{T}_{i,j}^{(3)}\right),
\end{aligned}
$$

where

$$
\begin{aligned}
\boldsymbol{T}_{i,j}^{(1)} &= \boldsymbol{Z}_{i,j} \frac{\partial \boldsymbol{Z}_{i,j}}{\partial x}. \\
\boldsymbol{T}_{i,j}^{(2)} &= f(\mathbf{u})_{i,j} \frac{\partial \boldsymbol{Z}_{i,j}}{\partial x}. \\
\boldsymbol{T}_{i,j}^{(3)} &= \mathcal{E}_{i,j} \frac{\partial \boldsymbol{Z}_{i,j}}{\partial x}.
\end{aligned}
$$

**Proposition D.47.** $\mathbb{E}_{\mathbf{u},\mathcal{E}}[\boldsymbol{T}_{i,j}^{(3)}] = 0$.

*Proof.* Follows from the facts that $\Delta$ and $\Delta_{\mathcal{E}}$ are independent, and $\mathbb{E}[\mathcal{E}_{i,j}] = 0$. $\qquad \square$

From now on, we will ignore the term $\boldsymbol{T}_{i,j}^{(3)}$ because of Proposition D.47 we can (in expectation) assume that $\boldsymbol{T}_{i,j}^{(3)} = 0$.

### D.5.4 SETTING THE GRADIENTS TO 0

In this section we will prove the gradients of the loss function are 0, under some given assumptions on the input data and parameters, when the functions we're learning are MULTIPLY and LINEAR. In order to do so we need to have another restriction on the target function $f$ which is that $f$ must be defined with a linear map, $\overline{\boldsymbol{W}} \in \mathbb{R}^{d \times d}$ that has non-zero columns. We make further assumptions on $\boldsymbol{W}, \boldsymbol{B}^{(1)}, \boldsymbol{K}, \boldsymbol{B}^{(2)}$, which we will justify later. We note that this assumption is satisfied for both the MULTIPLY and LINEAR functions.

**Assumption D.48.** For all $j \in [d_{out}]$ and $c \in [d - d_{out}]$, $(i)$ either $(\boldsymbol{K}_{:,j+c} \neq \mathbf{0})$ or $\left(\boldsymbol{B}_{:,j+c}^{(2)} \neq \mathbf{0}\right)$ and $(ii)$ $\boldsymbol{W}_{:,j+c} \neq \mathbf{0}$. Further, we have $\boldsymbol{B}^{(1)} = \mathbf{0}$.

The target function $f : \mathbb{R}^{N \times d} \to \mathbb{R}^{N \times d_{out}}$ is

1. Implementable with 1-layer $BC\left(\overline{\boldsymbol{W}}, \overline{\boldsymbol{K}}, \overline{\boldsymbol{B}}^{(1)}, \overline{\boldsymbol{B}}^{(2)}\right)$ such that for all $j \in [d_{out}]$ and $c \in [d - d_{out}], \overline{\boldsymbol{W}}_{:,j+c} \neq \mathbf{0}$

2. $f(\mathbf{u})_{i,j}$ only depends on $\mathbf{u}_{i,:}$.

We make another assumption to assist with the following theorems,

**Assumption D.49.** For all $j \in \{c, \ldots, c + d_{out} - 1\}$, $\langle \mathbf{W}_{:,j}, \overline{\mathbf{W}}_{:,j} \rangle \neq 0$ and $\overline{\mathbf{W}}_{:,j} \neq \mathbf{0}$.

The main results are as follows. First for the MULTIPLY function we have

**Theorem D.50.** *Given Assumptions D.45, D.46, D.48, and a function*
$$f(\mathbf{u}, a, b, d_{out}) = \mathbf{u}_{:,a:a+d_{out}-1} \odot \mathbf{u}_{:,b:b+d_{out}-1},$$
*where $a, b \in [d - d_{out}]$ and with $a \leq b$ and $c = a^2$. Let $\boldsymbol{\theta}_0$ be such that, $\mathbb{E}\nabla_{\boldsymbol{\theta}}\overline{L}|_{\theta \leftarrow \theta_0} = \mathbf{0}$ then* BASECONV$(\mathbf{u}, \boldsymbol{\theta}_0)[:, c : c + d_{out}] = f(\mathbf{u})$.

We prove a similar result for LINEAR function:

**Theorem D.51.** *Given Assumptions D.45, D.46, D.48, D.49, and a function*
$$f(\mathbf{u}) = \mathbf{u}\overline{W}.$$
*Let $\boldsymbol{\theta}_0$ be such that, $\mathbb{E}\nabla_{\boldsymbol{\theta}}\overline{L}|_{\theta \leftarrow \theta_0} = \mathbf{0}$ with $c = 0$ and $d_{out} = d$. Then* BASECONV$(\mathbf{u}, \boldsymbol{\theta}_0, 0, d) = f(\mathbf{u})$.

The following is to provide information about each entry in the output of a BASECONV layer.

**Lemma D.52.** *For all $(i, j) \in [N] \times [d_{out}]$, the entries of a resulting layer of* BASECONV, $\mathbf{Z}$, *are:*
$$\mathbf{Z}_{i,j} = \left( \left( \sum_{\ell=0}^{d-1} \mathbf{u}_{i,\ell} \cdot \mathbf{W}_{\ell,j+c} \right) + \mathbf{B}_{i,j+c}^{(1)} \right) \cdot \left( \left( \sum_{k=0}^{i} \mathbf{K}_{k,j+c} \cdot \mathbf{u}_{i-k,j+c} \right) + \mathbf{B}_{i,j+c}^{(2)} \right) \quad (50)$$

*Proof.* To begin, from Equation (48), we know that a layer of BASECONV yields a matrix $\mathbf{Z}$ as,
$$\mathbf{Z} = \left( \mathbf{u} \cdot \mathbf{W} + \mathbf{B}^{(1)} \right) \odot \left( \mathbf{K} * \mathbf{u} + \mathbf{B}^{(2)} \right) [:, c : c + d_{out} - 1].$$

Looking at the $\mathbf{u} \cdot \mathbf{W}$ operation, we know that for a row $i \in [N]$ and column $j \in [d_{out}]$, the vector dot product is computed as
$$\left\langle \mathbf{u}_{i,:}^{\top}, \mathbf{W}_{:,j+c} \right\rangle = \sum_{\ell=0}^{d-1} \mathbf{u}_{i,\ell} \cdot \mathbf{W}_{\ell,j+c}.$$
Meaning each entry in the resulting matrix is defined as such
$$(\mathbf{u} \cdot \mathbf{W})_{i,j+c} = \sum_{\ell=0}^{d-1} \mathbf{u}_{i,\ell} \cdot \mathbf{W}_{\ell,j+c}.$$
To sum the matrix $\mathbf{B}^{(1)}$ to this operation, we simply add the corresponding index giving us
$$\left( \mathbf{u} \cdot \mathbf{W} + \mathbf{B}^{(1)} \right)_{i,j+c} = \left( \sum_{\ell=0}^{d-1} \mathbf{u}_{i,\ell} \cdot \mathbf{W}_{\ell,j+c} \right) + \mathbf{B}_{i,j+c}^{(1)}. \quad (51)$$
Then, for the convolution operation between $\mathbf{K}$ and $\mathbf{u}$, that's computed column by column, we have for all $j \in [d_{out}]$:
$$(\mathbf{K} * \mathbf{u})_{:,j+c} = \mathbf{K}_{:,j+c} * \mathbf{u}_{:,j+c},$$
i.e. for any $i \in [N]$,
$$(\mathbf{K}_{:,j+c} * \mathbf{u}_{:,j+c})[i] = \sum_{k=0}^{i} \mathbf{K}_{k,j+c} \cdot \mathbf{u}_{i-k,j+c}.$$
Finally, to sum $\mathbf{B}^{(2)}$ we add the corresponding entry giving us
$$\left( \mathbf{K} * \mathbf{u} + \mathbf{B}^{(2)} \right) = \left( \sum_{k=0}^{i} \mathbf{K}_{k,j+c} \cdot \mathbf{u}_{i-k,j+c} \right) + \mathbf{B}_{i,j+c}^{(2)}. \quad (52)$$
Combining Equations (51) and (52), gives us Equation (50) as expected. □

---

[2]These assumptions are without loss of generality.

### D.5.5 Some partial derivatives are always zero

To simplify future computations in this section, we will state a simple lemma on some partial derivatives that always go to 0.

**Lemma D.53.** *Fix* $i \in [N], j \in [d_{out}], j' \in [d], c \in [d - d_{out}]$. *Then for any* $j' \neq j + c, 0 \leq \ell < N$, *and* $0 \leq k < N$ *we have,*

$$\frac{\partial \boldsymbol{Z}_{i,j}}{\partial \boldsymbol{W}_{\ell,j'}} = \frac{\partial \boldsymbol{Z}_{i,j}}{\partial \boldsymbol{K}_{k,j'}} = 0.$$

*Further, any* $(i, j + c) \neq (i', j')$ *we have,*

$$\frac{\partial \boldsymbol{Z}_{i,j}}{\partial \boldsymbol{B}^{(1)}_{i',j'}} = \frac{\partial \boldsymbol{Z}_{i,j}}{\partial \boldsymbol{B}^{(2)}_{i',j'}} = 0.$$

*Proof.* Follows from Equation (50) and definition of partial derivatives. $\square$

### D.5.6 Generic form of partial derivatives plus a consequence

Given Lemma D.52 we can conclude the following.

**Lemma D.54.** *For* $0 \leq i < N, 0 \leq j < d_{out},$ *and* $0 \leq c < d - d_{out},$ *any entry* $x \in \{\boldsymbol{W}_{i,j+c}, \boldsymbol{B}^{(1)}_{i,j+c}\}$,

$$\frac{\partial \boldsymbol{Z}_{i,j}}{\partial x} = \left( (\boldsymbol{K}_{:,j+c} * \mathbf{u}_{:,j+c}) [i] + \boldsymbol{B}^{(2)}_{i,j+c} \right) \frac{\partial}{\partial x} \left( \langle \mathbf{u}_{i,:}^\top, \boldsymbol{W}_{:,j+c} \rangle + \boldsymbol{B}^{(1)}_{i,j+c} \right)$$

*then for any entry* $x \in \{\boldsymbol{K}_{i,j+c}, \boldsymbol{B}^{(2)}_{i,j+c}\}$,

$$\frac{\partial \boldsymbol{Z}_{i,j}}{\partial x} = \left( \langle \mathbf{u}_{i,:}^\top, \boldsymbol{W}_{:,j+c} \rangle + \boldsymbol{B}^{(1)}_{i,j+c} \right) \cdot \frac{\partial}{\partial x} \left( (\boldsymbol{K}_{:,j+c} * \mathbf{u}_{:,j+c}) [i] + \boldsymbol{B}^{(2)}_{i,j+c} \right).$$

A consequence of Lemma D.54 is the following.

**Corollary D.55.** *Let* $\boldsymbol{\theta} = \left( \boldsymbol{W}, \boldsymbol{K}, \boldsymbol{B}^{(1)}, \boldsymbol{B}^{(2)} \right) = \boldsymbol{0}$. *Then for all parameter variables x, we have*

$$\frac{\partial \boldsymbol{Z}_{i,j}}{\partial x} = 0.$$

*Specifically,*

$$\nabla_{\boldsymbol{\theta}} \overline{L} (\boldsymbol{\theta}) |_{\boldsymbol{\theta} = \boldsymbol{0}} = \boldsymbol{0}.$$

Corollary D.55 implies that initializing $\boldsymbol{\theta} = \boldsymbol{0}$ is not a good choice for initializing parameters since it is a local minima.

We can exactly figure out the partial derivatives in Lemma D.54 by the following.

**Lemma D.56.** *Fix* $i \in [N], j' \in [d]$. *Then for any* $0 \leq \ell < N$ *we have,*

$$\frac{\partial}{\partial \boldsymbol{W}_{\ell,j'}} \left( \langle \mathbf{u}_{i,:}^\top, \boldsymbol{W}_{:,j'} \rangle + \boldsymbol{B}^{(1)}_{i,j'} \right) = \mathbf{u}_{i,\ell}$$

*and*

$$\frac{\partial}{\partial \boldsymbol{B}^{(1)}_{i,j'}} \left( \langle \mathbf{u}_{i,:}^\top, \boldsymbol{W}_{:,j'} \rangle + \boldsymbol{B}^{(1)}_{i,j'} \right) = 1.$$

*Also,*

$$\frac{\partial}{\partial \boldsymbol{B}^{(2)}_{i,j'}} \left( (\boldsymbol{K}_{:,j'} * \mathbf{u}_{:,j'}) [i] + \boldsymbol{B}^{(2)}_{i,j'} \right) = 1.$$

*Next, for any $0 \leq k \leq i$, we have*

$$\frac{\partial}{\partial \boldsymbol{K}_{k,j'}} \left( (\boldsymbol{K}_{:,j'} * \mathbf{u}_{:,j'}) [i] + \boldsymbol{B}_{i,j'}^{(2)} \right) = \mathbf{u}_{i-k,j'}$$

*and for all $k > i$,*

$$\frac{\partial}{\partial \boldsymbol{K}_{k,j'}} \left( (\boldsymbol{K}_{:,j'} * \mathbf{u}_{:,j'}) [i] + \boldsymbol{B}_{i,j'}^{(2)} \right) = 0.$$

*Proof.* Let's begin by looking at

$$\frac{\partial}{\partial \boldsymbol{W}_{\ell,j'}} \left( \langle \mathbf{u}_{i,:}^\top, \boldsymbol{W}_{:,j'} \rangle + \boldsymbol{B}_{i,j'}^{(1)} \right).$$

Expanding this out gives us

$$\frac{\partial}{\partial \boldsymbol{W}_{\ell,j'}} \left( \sum_{\ell'=0}^{d-1} \mathbf{u}_{i,\ell'} \cdot \boldsymbol{W}_{\ell',j'} + \boldsymbol{B}_{i,j'}^{(1)} \right).$$

When we take the partial derivative of this with respect to $\boldsymbol{W}_{\ell,j'}$, the $\boldsymbol{B}_{i,j'}^{(1)}$ term goes to 0. And the term

$$\frac{\partial}{\partial \boldsymbol{W}_{\ell,j'}} \left( \sum_{\ell'=0}^{d-1} \mathbf{u}_{i,\ell'} \cdot \boldsymbol{W}_{\ell',j'} \right) = \mathbf{u}_{i,\ell},$$

as desired, since $\boldsymbol{W}_{\ell,j'}$ only shows up in the summation when $\ell' = \ell$.

Next, let us look at

$$\frac{\partial}{\partial \boldsymbol{B}_{i,j'}^{(1)}} \left( \langle \mathbf{u}_{i,:}^\top, \boldsymbol{W}_{:,j'} \rangle + \boldsymbol{B}_{i,j'}^{(1)} \right)$$

Since $\boldsymbol{B}_{i,j'}^{(1)}$ doesn't show up in the dot product of the vectors, we know that piece goes to zero, giving us

$$\frac{\partial}{\partial \boldsymbol{B}_{i,j'}^{(1)}} \left( \langle \mathbf{u}_{i,:}^\top, \boldsymbol{W}_{:,j'} \rangle + \boldsymbol{B}_{i,j'}^{(1)} \right) = \frac{\partial \boldsymbol{B}_{i,j'}^{(1)}}{\partial \boldsymbol{B}_{i,j'}^{(1)}} = 1,$$

as desired.

Next, for any $0 \leq k \leq i$ we have

$$\frac{\partial}{\partial \boldsymbol{K}_{k,j'}} \left( (\boldsymbol{K}_{:,j'} * \mathbf{u}_{:,j'}) [i] + \boldsymbol{B}_{i,j'}^{(2)} \right).$$

The $\boldsymbol{B}_{i,j'}^{(2)}$ term goes to 0 as we're taking the partial derivative with respect to $\boldsymbol{K}_{k,j'}$. So we have

$$\frac{\partial}{\partial \boldsymbol{K}_{k,j'}} ((\boldsymbol{K}_{:,j'} * \mathbf{u}_{:,j'}) [i]) = \frac{\partial}{\partial \boldsymbol{K}_{k,j'}} \sum_{k'=0}^{i} \boldsymbol{K}_{k',j'} \mathbf{u}_{i-k',j'} = \mathbf{u}_{i-k,j'}$$

as desired, since $\boldsymbol{K}_{k,j'}$ only shows up in the summation when $k' = k$.

Next, for $k > i$ we have

$$\frac{\partial}{\partial \boldsymbol{K}_{k,j'}} ((\boldsymbol{K}_{:,j'} * \mathbf{u}_{:,j'}) [i]) = \frac{\partial}{\partial \boldsymbol{K}_{k,j'}} \sum_{k'=0}^{i} \boldsymbol{K}_{k',j'} \mathbf{u}_{i-k',j'}, = 0 \tag{53}$$

as desired, since $\boldsymbol{K}_{k,j'}$ will never show up in the summation as $k' < k$.

Finally, let us look at the fourth piece,

$$\frac{\partial}{\partial \boldsymbol{B}_{i,j'}^{(2)}} \left( (\boldsymbol{K}_{:,j'} * \mathbf{u}_{:,j'}) [i] + \boldsymbol{B}_{i,j'}^{(2)} \right).$$

The term $\boldsymbol{B}^{(2)}_{i,j'}$ doesn't appear in the result of the convolution operation, therefore that piece goes to 0, giving us

$$\frac{\partial}{\partial \boldsymbol{B}^{(2)}_{i,j'}} \left( \left( \boldsymbol{K}_{:,j'} * \mathbf{u}_{:,j'} \right)[i] + \boldsymbol{B}^{(2)}_{i,j'} \right) = \frac{\partial \boldsymbol{B}^{(2)}_{i,j'}}{\partial \boldsymbol{B}^{(2)}_{i,j'}} = 1,$$

as desired.

$\square$

**Definition D.57.** For the rest of the section, we will redefine $\boldsymbol{\theta} = \left( \boldsymbol{W}, \boldsymbol{K}, \boldsymbol{B}^{(2)} \right)$. Note that we are just removing $\boldsymbol{B}^{(1)}$ since it is all zeros as per Assumption D.48.

**Lemma D.58.** *Given Assumption D.45 and recall that $\boldsymbol{B}^{(1)} = \boldsymbol{0}$. Fix $i \in [N], j \in [d_{out}], c \in [d - d_{out}]$. Then we have*

$$\mathbb{E}\left[ \boldsymbol{Z}_{i,j} \frac{\partial \boldsymbol{Z}_{i,j}}{\partial \boldsymbol{B}^{(2)}_{i,j+c}} \right] = \boldsymbol{B}^{(2)}_{i,j+c} \sum_{\ell'=0}^{d-1} \mathbb{E}\left[ \mathbf{u}^2_{i,\ell'} \right] \boldsymbol{W}^2_{\ell',j+c}.$$

*Next, for any $0 \le k \le i$ we have*

$$\mathbb{E}\left[ \boldsymbol{Z}_{i,j} \frac{\partial \boldsymbol{Z}_{i,j}}{\partial \boldsymbol{K}_{k,j+c}} \right] = \boldsymbol{K}_{k,j+c} \sum_{\ell'=0}^{d-1} \boldsymbol{W}^2_{\ell',j+c} \mathbb{E}\left[ \mathbf{u}^2_{i,\ell'} \cdot \mathbf{u}^2_{i-k,j+c} \right].$$

*For $k > i$,*

$$\mathbb{E}\left[ \boldsymbol{Z}_{i,j} \frac{\partial \boldsymbol{Z}_{i,j}}{\partial \boldsymbol{K}_{k,j+c}} \right] = 0.$$

*Finally, for any $0 \le \ell \le d - 1$,*

$$\mathbb{E}\left[ \boldsymbol{Z}_{i,j} \frac{\partial \boldsymbol{Z}_{i,j}}{\partial \boldsymbol{W}_{\ell,j+c}} \right] = \boldsymbol{W}_{\ell,j+c} \sum_{k'=0}^{i} \boldsymbol{K}^2_{k',j+c} \mathbb{E}\left[ \mathbf{u}^2_{i-k',j+c} \cdot \mathbf{u}^2_{i,\ell} \right] + \left( \boldsymbol{B}^{(2)}_{i,j+c} \right)^2 \boldsymbol{W}_{\ell,j+c} \mathbb{E}\left[ \mathbf{u}^2_{i,\ell} \right].$$

*Proof.* Given Lemma D.54 and Lemma D.56 (along with the fact that $\boldsymbol{B}^{(1)} = \boldsymbol{0}$) we have

$$\mathbb{E}\left[ \boldsymbol{Z}_{i,j} \frac{\partial \boldsymbol{Z}_{i,j}}{\partial \boldsymbol{B}^{(2)}_{i,j+c}} \right] = \mathbb{E}\left[ \left( \boldsymbol{K}_{:,j+c} * \mathbf{u}_{:,j+c}[i] + \boldsymbol{B}^{(2)}_{i,j+c} \right) \left\langle \mathbf{u}^\top_{i,:}, \boldsymbol{W}_{:,j+c} \right\rangle^2 \right]$$

$$= \sum_{\ell'=0}^{d-1} \sum_{\ell''=0}^{d-1} \sum_{k'=0}^{i} \mathbb{E}\left[ \mathbf{u}_{i,\ell'} \mathbf{u}_{i-k',j+c} \mathbf{u}_{i,\ell''} \right] \boldsymbol{W}_{\ell',j+c} \boldsymbol{W}_{\ell'',j+c} \boldsymbol{K}_{k',j+c}$$

$$+ \boldsymbol{B}^{(2)}_{i,j+c} \sum_{\ell'=0}^{d-1} \sum_{\ell''=0}^{d-1} \mathbb{E}\left[ \mathbf{u}_{i,\ell'} \mathbf{u}_{i,\ell''} \right] \boldsymbol{W}_{\ell',j+c} \boldsymbol{W}_{\ell'',j+c}.$$

In the above, the first summation goes to 0 since for all $\ell', \ell'', k$, by Assumption D.45, the expected value of the product of three $\mathbf{u}$'s will be 0 since there's an odd number of them. Again, by Assumption D.45, the second summation will be non-zero if and only if $\ell' = \ell''$. Therefore we get the following,

$$\mathbb{E}\left[ \boldsymbol{Z}_{i,j} \frac{\partial \boldsymbol{Z}_{i,j}}{\partial \boldsymbol{B}^{(2)}_{i,j}} \right] = \boldsymbol{B}^{(2)}_{i,j+c} \sum_{\ell'=0}^{d-1} \mathbb{E}\left[ \mathbf{u}^2_{i,\ell'} \right] \boldsymbol{W}^2_{\ell',j+c}$$

as desired.

Moving onto the next piece, using Lemma D.54 and Lemma D.56 (along with the fact that $\boldsymbol{B}^{(1)} = \boldsymbol{0}$) we have for $0 \leq k \leq i$,

$$
\begin{aligned}
\mathbb{E}\left[\boldsymbol{Z}_{i,j} \frac{\partial \boldsymbol{Z}_{i,j}}{\partial \boldsymbol{K}_{k,j+c}}\right] &= \mathbb{E}\left[\left(\boldsymbol{K}_{:,j+c} * \mathbf{u}_{:,j+c}[i] + \boldsymbol{B}_{i,j+c}^{(2)}\right) \left\langle \mathbf{u}_{i,:}^{\top}, \boldsymbol{W}_{:,j+c}\right\rangle^2 \mathbf{u}_{i-k,j+c}\right] \\
&= \sum_{\ell'=0}^{d-1}\sum_{\ell''=0}^{d-1}\sum_{k'=0}^{i} \mathbb{E}\left[\mathbf{u}_{i,\ell'}\mathbf{u}_{i,\ell''}\mathbf{u}_{i-k',j+c}\mathbf{u}_{i-k,j+c}\right] \boldsymbol{W}_{\ell',j+c}\boldsymbol{W}_{\ell'',j+c}\boldsymbol{K}_{k',j+c} \\
&\quad + \boldsymbol{B}_{i,j+c}^{(2)} \sum_{\ell'=0}^{d-1}\sum_{\ell''=0}^{d-1} \mathbb{E}\left[\mathbf{u}_{i,\ell'}\mathbf{u}_{i,\ell''}\mathbf{u}_{i-k,j+c}\right] \boldsymbol{W}_{\ell',j+c}\boldsymbol{W}_{\ell'',j+c}.
\end{aligned}
$$

By Assumption D.45, only expected values of terms with square monomials are non-zero. Specifically, the first summation has the $\mathbf{u}_{i-k,j}$ term, therefore, we need $k' = k$ to get an even exponent. This is the same reasoning for $\ell' = \ell''$. Therefore, the first summation is non-zero if and only if $k' = k$ and $\ell' = \ell''$. The second summation will be 0 since for all $\ell', \ell''$ the expected value of the $\mathbf{u}$'s is 0 since there's an odd number of them, there will always be an odd exponent. So we get

$$
\mathbb{E}\left[\boldsymbol{Z}_{i,j} \frac{\partial \boldsymbol{Z}_{i,j}}{\partial \boldsymbol{K}_{k,j+c}}\right] = \boldsymbol{K}_{k,j+c} \sum_{\ell'=0}^{d-1} \boldsymbol{W}_{\ell',j+c}^2 \mathbb{E}\left[\mathbf{u}_{i,\ell'}^2 \cdot \mathbf{u}_{i-k,j+c}^2\right]
$$

as desired.

When $k > i$,

$$
\mathbb{E}\left[\boldsymbol{Z}_{i,j} \frac{\partial \boldsymbol{Z}_{i,j}}{\partial \boldsymbol{K}_{k,j+c}}\right] = 0
$$

since we index the convolution piece at $i$, $\partial \boldsymbol{K}_{k,j+c}$ for $k > i$ will never be in the piece we're taking the derivative of. Moving onto the final piece, given Lemma D.54 and Lemma D.56 and $\boldsymbol{B}^{(1)} = \boldsymbol{0}$ we have

$$
\begin{aligned}
\mathbb{E}\left[\boldsymbol{Z}_{i,j} \frac{\partial \boldsymbol{Z}_{i,j}}{\partial \boldsymbol{W}_{\ell,j+c}}\right] &= \mathbb{E}\left[\left((\boldsymbol{K}_{:,j+c} * \mathbf{u}_{:,j+c})[i] + \left(\boldsymbol{B}_{i,j+c}^{(2)}\right)\right)^2 \left\langle \mathbf{u}_{i,:}^{\top}, \boldsymbol{W}_{:,j+c}\right\rangle \mathbf{u}_{i,\ell}\right] \\
&= \sum_{k'=0}^{i}\sum_{k''=0}^{i}\sum_{\ell'=0}^{d-1} \mathbb{E}\left[\mathbf{u}_{i-k',j+c}\mathbf{u}_{i,\ell'}\mathbf{u}_{i-k'',j+c}\mathbf{u}_{i,\ell}\right] \boldsymbol{K}_{k',j+c}\boldsymbol{K}_{k'',j+c}\boldsymbol{W}_{\ell',j+c} \\
&\quad + 2\boldsymbol{B}_{i,j+c}^{(2)} \sum_{k'=0}^{i}\sum_{\ell'=0}^{d-1} \mathbb{E}\left[\mathbf{u}_{i-k',j+c}\mathbf{u}_{i,\ell'}\mathbf{u}_{i,\ell}\right] \boldsymbol{K}_{k',j+c}\boldsymbol{W}_{\ell',j+c} \\
&\quad + \left(\boldsymbol{B}_{i,j+c}^{(2)}\right)^2 \sum_{\ell'=0}^{d-1} \mathbb{E}\left[\mathbf{u}_{i,\ell'}\mathbf{u}_{i,\ell}\right] \boldsymbol{W}_{\ell',j+c}.
\end{aligned}
$$

We again use Assumption D.45 to simplify the summations. The first summation has the $\mathbf{u}_{i,\ell}$ term, therefore to get an even exponent on it we need $\ell' = \ell$. This is the same reasoning for $k' = k''$. Therefore the first summation will be non-zero if and only if $\ell' = \ell$ and $k' = k''$. The second summation will be 0 for all $k', \ell'$ since we're taking the expected value of an odd number of $\mathbf{u}$ products, there will always be an odd exponent. The third term will be non-zero if and only if $\ell' = \ell$ to get an even exponent on $\mathbf{u}$'s entry. Therefore we have,

$$
\mathbb{E}\left[\boldsymbol{Z}_{i,j} \frac{\partial \boldsymbol{Z}_{i,j}}{\partial \boldsymbol{W}_{\ell,j+c}}\right] = \boldsymbol{W}_{\ell,j+c} \sum_{k'=0}^{i} \boldsymbol{K}_{k',j+c}^2 \mathbb{E}\left[\mathbf{u}_{i-k',j+c}^2 \cdot \mathbf{u}_{i,\ell}^2\right] + \left(\boldsymbol{B}_{i,j+c}^{(2)}\right)^2 \boldsymbol{W}_{\ell,j+c} \mathbb{E}\left[\mathbf{u}_{i,\ell}^2\right]
$$

as desired.

$\square$

### D.5.7   LINEAR

The following lemma will be for when the function we are considering is a linear map.

**Lemma D.59.** *With $c = 0$ and $d_{out} = d$, fix $i \in [N]$, $j \in [d]$, and $\overline{W} \in \mathbb{R}^{d \times d}$. Then we have*

$$\mathbb{E}\left[\langle \mathbf{u}_{i,:}^\top, \overline{W}_{:,j} \rangle \frac{\partial Z_{i,j}}{\partial B_{i,j}^{(2)}}\right] = \sum_{\ell=0}^{d-1} W_{\ell,j} \overline{W}_{\ell,j} \mathbb{E}\left[\mathbf{u}_{i,\ell}^2\right].$$

*For all $k$,*

$$\mathbb{E}\left[\langle \mathbf{u}_{i,:}^\top, \overline{W}_{:,j} \rangle \frac{\partial Z_{i,j}}{\partial K_{k,j}}\right] = 0.$$

*For all $\ell$,*

$$\mathbb{E}\left[\langle \mathbf{u}_{i,:}^\top, \overline{W}_{:,j} \rangle \frac{\partial Z_{i,j}}{\partial W_{\ell,j}}\right] = B_{i,j}^{(2)} \overline{W}_{\ell,j} \mathbb{E}\left[\mathbf{u}_{i,\ell}^2\right].$$

*Proof.* Given Lemma D.54 and Lemma D.56 (and the fact that $B^{(1)} = 0$) we have

$$\mathbb{E}\left[\langle \mathbf{u}_{i,:}^\top, \overline{W}_{:,j} \rangle \frac{\partial Z_{i,j}}{\partial B_{i,j}^{(2)}}\right] = \mathbb{E}\left[\langle \mathbf{u}_{i,:}^\top, W_{:,j} \rangle \langle \mathbf{u}_{i,:}^\top, \overline{W}_{:,j} \rangle\}\right]$$

$$= \sum_{\ell=0}^{d-1} \sum_{\ell'=0}^{d-1} W_{\ell,j} \overline{W}_{\ell',j} \mathbb{E}\left[\mathbf{u}_{i,\ell} \mathbf{u}_{i,\ell'}\right]$$

From Assumption D.45 we get that the summation will always be non-zero if and only if $\ell = \ell'$ so that the $\mathbf{u}$ variable has an even exponent. Therefore we get,

$$\mathbb{E}\left[\langle \mathbf{u}_{i,:}^\top, \overline{W}_{:,j} \rangle \frac{\partial Z_{i,j}}{\partial B_{i,j}^{(2)}}\right] = \sum_{\ell=0}^{d-1} W_{\ell,j} \overline{W}_{\ell,j} \mathbb{E}\left[\mathbf{u}_{i,\ell}^2\right]$$

as desired.

Next, for all $k$, by Lemma D.54 and Lemma D.56 (and the fact that $B^{(1)} = 0$)

$$\mathbb{E}\left[\langle \mathbf{u}_{i,:}^\top, \overline{W}_{:,j} \rangle \frac{\partial Z_{i,j}}{\partial K_{k,j}}\right] = \mathbb{E}\left[\langle \mathbf{u}_{i,:}^\top, W_{:,j} \rangle \mathbf{u}_{i-k,j} \langle \mathbf{u}_{i,:}^\top, \overline{W}_{:,j} \rangle\right]$$

$$= \sum_{\ell=0}^{d-1} \sum_{\ell'=0}^{d-1} W_{\ell,j} \overline{W}_{\ell',j} \mathbb{E}\left[\mathbf{u}_{i,\ell} \mathbf{u}_{i,\ell'} \mathbf{u}_{i-k,j}\right].$$

We simplify the above using Assumption D.45. The summation goes to 0 since there are an odd number of $\mathbf{u}$ terms, there will always be an odd exponent. Note that this is true for $k \leq i$. Recall from Equation (53) that for $k > i$,

$$\frac{\partial Z_{i,j}}{\partial K_{k,j}} = 0.$$

Therefore, for all $k$,

$$\mathbb{E}\left[\langle \mathbf{u}_{i,:}^\top, \overline{W}_{:,j} \rangle \frac{\partial Z_{i,j}}{\partial K_{k,j}}\right] = 0$$

as desired.

Next, for all $\ell$, by Lemma D.54 and Lemma D.56 (and the fact that $B^{(1)} = 0$)

$$\mathbb{E}\left[\langle \mathbf{u}_{i,:}^\top, \overline{W}_{:,j} \rangle \frac{\partial Z_{i,j}}{\partial W_{\ell,j}}\right] = \mathbb{E}\left[\left((K_{:,j} * \mathbf{u}_{:,j})[i] + B_{i,j}^{(2)}\right) \mathbf{u}_{i,\ell} \langle \mathbf{u}_{i,:}^\top, \overline{W}_{:,j} \rangle\right]$$

$$= \left(\sum_{k'=0}^{i} \sum_{\ell'=0}^{d-1} K_{k,j} \overline{W}_{\ell',j} \mathbb{E}\left[\mathbf{u}_{i-k,j} \mathbf{u}_{i,\ell'} \mathbf{u}_{i,\ell}\right]\right)$$

$$+ B_{i,j}^{(2)} \sum_{\ell'=0}^{d-1} \overline{W}_{\ell',j} \mathbb{E}\left[\mathbf{u}_{i,\ell} \mathbf{u}_{i,\ell'}\right]$$

We simplify the above using Assumption D.45. The first summation will always be 0 due to an odd exponent on the $\mathbf{u}$'s. The second summation piece will always be non-zero if and only if $\ell' = \ell$, giving us the even exponent on the $\mathbf{u}$ variable. Therefore we get,

$$\mathbb{E}\left[\langle \mathbf{u}_{i,:}^{\top}, \overline{\boldsymbol{W}}_{:,j}\rangle \frac{\partial \boldsymbol{Z}_{i,j}}{\partial \boldsymbol{W}_{\ell,j}}\right] = \boldsymbol{B}_{i,j}^{(2)}\overline{\boldsymbol{W}}_{\ell,j}\mathbb{E}\left[\mathbf{u}_{i,\ell}^2\right]$$

as desired. $\qquad\square$

Next, we restate Theorem D.51 and prove it:

**Theorem D.60** (Theorem D.51, restated). *Given Assumptions D.45, D.46, D.48, D.49, and a function*

$$f(\mathbf{u}) = \mathbf{u}\overline{\boldsymbol{W}}.$$

*Let $\boldsymbol{\theta}_0$ be such that, $\mathbb{E}\nabla_{\boldsymbol{\theta}}\overline{L}|_{\theta\leftarrow\theta_0} = \mathbf{0}$ with $c = 0$ and $d_{out} = d$. Then $\mathrm{BASECONV}(\mathbf{u}, \boldsymbol{\theta}_0, c, d_{out}) = f(\mathbf{u})$.*

*Proof.* From Lemma D.53 we get that

$$\mathbb{E}\left[\frac{\partial \overline{L}}{\partial \boldsymbol{B}_{i,j}^{(2)}}\right] = \sum_{i'=0}^{N-1}\sum_{j'=0}^{d_{out}-1}\mathbb{E}\left[\frac{\partial \overline{L}_{i',j'}}{\partial \boldsymbol{B}_{i,j}^{(2)}}\right]$$

$$= \mathbb{E}\left[\frac{\partial \overline{L}_{i,j}}{\partial \boldsymbol{B}_{i,j}^{(2)}}\right]$$

Recall our loss function from Equation (49). Then given Proposition D.47, Lemma D.58, and Lemma D.59 we know that

$$\mathbb{E}\left[\frac{\partial \overline{L}_{i,j}}{\partial \boldsymbol{B}_{i,j}^{(2)}}\right] = 2\mathbb{E}\left[\frac{\partial \boldsymbol{Z}_{i,j}}{\partial \boldsymbol{B}_{i,j}^{(2)}}\left(\boldsymbol{Z}_{i,j} - \left(\mathbf{u}\overline{\boldsymbol{W}}\right)_{i,j}\right)\right]$$

$$= 2\left(\boldsymbol{B}_{i,j}^{(2)}\sum_{\ell=0}^{d-1}\boldsymbol{W}_{\ell,j}^2\mathbb{E}\left[\mathbf{u}_{i,\ell}^2\right] - \sum_{\ell=0}^{d-1}\boldsymbol{W}_{\ell,j}\overline{\boldsymbol{W}}_{\ell,j}\mathbb{E}\left[\mathbf{u}_{i,\ell}^2\right]\right).$$

Setting this to 0 and solving for $\boldsymbol{B}_{i,j}^{(2)}$ gives us,

$$\boldsymbol{B}_{i,j}^{(2)} = \frac{\sum_{\ell=0}^{d-1}\boldsymbol{W}_{\ell,j}\overline{\boldsymbol{W}}_{\ell,j}}{\sum_{\ell=0}^{d-1}\boldsymbol{W}_{\ell,j}^2}.$$

Given Assumption D.49 we know that the numerator will be non-zero and given assumption Assumption D.48 we know the denominator will always be non-zero as well. Therefore we get that for all $i, j$

$$\boldsymbol{B}_{i,j} = \frac{\langle \boldsymbol{W}_{:,j}, \overline{\boldsymbol{W}}_{:,j}\rangle}{\langle \boldsymbol{W}_{:,j}, \boldsymbol{W}_{:,j}\rangle} \stackrel{\text{def}}{=} b_j. \tag{54}$$

Next, via Lemma D.53 we have for all $k \geq 0$:

$$\mathbb{E}\left[\frac{\partial \overline{L}}{\partial \boldsymbol{K}_{k,j}}\right] = \sum_{i'=0}^{N-1}\sum_{j'=0}^{d_{out}-1}\mathbb{E}\left[\frac{\partial \overline{L}_{i',j'}}{\partial \boldsymbol{K}_{k,j}}\right]$$

$$= \sum_{i'=0}^{N-1}\mathbb{E}\left[\frac{\partial \overline{L}_{i',j}}{\partial \boldsymbol{K}_{k,j}}\right]$$

From Equation (49) and Proposition D.47 we get

$$\mathbb{E}\left[\frac{\partial \overline{L}}{\partial \boldsymbol{K}_{k,j}}\right] = \sum_{i'=0}^{N-1}2\mathbb{E}\left[\frac{\partial \boldsymbol{Z}_{i',j}}{\partial \boldsymbol{K}_{k,j}}\left(\boldsymbol{Z}_{i',j} - \left(\mathbf{u}\overline{\boldsymbol{W}}\right)_{i',j}\right)\right]$$

Then by Lemma D.58 and Lemma D.59 we get

$$\sum_{i'=0}^{N-1} 2\mathbb{E}\left[\frac{\partial \boldsymbol{Z}_{i',j}}{\partial \boldsymbol{K}_{k,j}}\boldsymbol{Z}_{i',j} - \frac{\partial \boldsymbol{Z}_{i',j}}{\partial \boldsymbol{K}_{k,j}}\left(\mathbf{u}\overline{\boldsymbol{W}}\right)_{i',j}\right] = 2\left(\sum_{i'=0}^{N-1} \boldsymbol{K}_{k,j}\sum_{\ell'=0}^{d-1} \boldsymbol{W}_{\ell',j}^2 \mathbb{E}\left[\mathbf{u}_{i',\ell'}^2 \cdot \mathbf{u}_{i'-k,j}^2\right]\right)$$

$$= 2\boldsymbol{K}_{k,j}\sum_{\ell'=0}^{d-1} \boldsymbol{W}_{\ell',j}\sum_{i'=0}^{N-1} \mathbb{E}\left[\mathbf{u}_{i',\ell'}^2 \cdot \mathbf{u}_{i'-k,j}^2\right].$$

Since we know that the summation piece over $\ell'$ is always non-zero due to the even exponents on the $\mathbf{u}$ terms and at least one of $\boldsymbol{W}_{\ell',j} \neq 0$. Therefore, when setting this to 0 and solve for $\boldsymbol{K}_{k,j}$ gives us,

$$\mathbb{E}\left[\frac{\partial \overline{L}}{\partial \boldsymbol{K}_{k,j}}\right] = 0 \implies 2\left(\boldsymbol{K}_{k,j}\boldsymbol{W}_{\ell',j}^2\right) = 0$$

implying for all $k$, $\boldsymbol{K}_{k,j} = 0$. In other words,

$$\boldsymbol{K} = \mathbf{0}^{N\times d}.$$

Finally, via Lemma D.53 we have for all $\ell$:

$$\mathbb{E}\left[\frac{\partial \overline{L}}{\partial \boldsymbol{W}_{\ell,j}}\right] = \sum_{i'=0}^{N-1}\sum_{j'=0}^{d_{out}-1} \mathbb{E}\left[\frac{\partial \overline{L}_{i',j'}}{\partial \boldsymbol{W}_{\ell,j}}\right]$$

$$= \sum_{i'=0}^{N-1} \mathbb{E}\left[\frac{\partial \overline{L}_{i',j}}{\partial \boldsymbol{W}_{\ell,j}}\right]$$

From Equation (49) and Proposition D.47 we get

$$\mathbb{E}\left[\frac{\partial \overline{L}}{\partial \boldsymbol{W}_{\ell,j}}\right] = \sum_{i'=0}^{N-1} 2\mathbb{E}\left[\frac{\partial \boldsymbol{Z}_{i',j}}{\partial \boldsymbol{W}_{\ell,j}}\left(\boldsymbol{Z}_{i',j} - \left(\mathbf{u}\overline{\boldsymbol{W}}\right)_{i',j}\right)\right]$$

Then from Lemma D.58 and Lemma D.59 we get

$$\sum_{i'=0}^{N-1} 2\mathbb{E}\left[\frac{\partial \boldsymbol{Z}_{i',j}}{\partial \boldsymbol{W}_{\ell,j}}\boldsymbol{Z}_{i',j} - \frac{\partial \boldsymbol{Z}_{i',j}}{\partial \boldsymbol{W}_{\ell,j}}\left(\mathbf{u}\overline{\boldsymbol{W}}\right)_{i',j}\right] = 2\left(\sum_{i'=0}^{N-1} \boldsymbol{W}_{\ell,j}\sum_{k'=0}^{i'} \boldsymbol{K}_{k',j}^2 \mathbb{E}\left[\mathbf{u}_{i'-k',j}^2 \mathbf{u}_{i',\ell}^2\right]\right)$$

$$+ 2\left(\left(\boldsymbol{B}_{i',j}^{(2)}\right)^2 \boldsymbol{W}_{\ell,j}\mathbb{E}\left[\mathbf{u}_{i',\ell}^2\right] - \boldsymbol{B}_{i',j}^{(2)}\overline{\boldsymbol{W}}_{\ell,j}\mathbb{E}\left[\mathbf{u}_{i',\ell}^2\right]\right)$$

Which simplifies to

$$2\left(\boldsymbol{W}_{\ell,j}\sum_{i'=0}^{N-1}\left(\boldsymbol{B}_{i',j}^{(2)}\right)^2 \mathbb{E}\left[\mathbf{u}_{i',\ell}^2\right] - \overline{\boldsymbol{W}}_{\ell,j}\sum_{i'=0}^{N-1} \boldsymbol{B}_{i',j}^{(2)}\mathbb{E}\left[\mathbf{u}_{i',\ell}^2\right]\right).$$

We can drop the summation with $\boldsymbol{K}$ in it as we know $\boldsymbol{K} = \mathbf{0}$. Recall that from (54), $\boldsymbol{B}_{i,j}^{(2)} = b_j$. Then we can rewrite the above as

$$\mathbb{E}\left[\frac{\partial \overline{L}}{\partial \boldsymbol{W}_{\ell,j}}\right] = 2b_j\left(\sum_{i'=0}^{N-1} \mathbb{E}\left[\mathbf{u}_{i',\ell}^2\right]\right)\left(\boldsymbol{W}_{\ell,j}b_j - \overline{\boldsymbol{W}}_{\ell,j}\right)$$

we know from Assumption D.45 that the first summation will always be non-zero since there's an even exponent on the $\mathbf{u}$ variable and we know that $b_j$ is non-zero. Therefore setting

$$\mathbb{E}\left[\frac{\partial \overline{L}}{\partial \boldsymbol{W}_{\ell,j}}\right] = 0$$

tells us that $\boldsymbol{W}_{\ell,j}b_j - \overline{\boldsymbol{W}}_{\ell,j} = 0$ or,

$$\boldsymbol{W}_{\ell,j} = \frac{\overline{\boldsymbol{W}}_{\ell,j}}{b_j}. \tag{55}$$

Given the above value for $\boldsymbol{W}_{\ell,j}$ and recall we have $\boldsymbol{K} = \boldsymbol{0}$ and $\boldsymbol{B}^{(2)} = (\overline{\mathbf{b}_0}\overline{\mathbf{b}_1}...\overline{\mathbf{b}_{d-1}})$ where each $\overline{\mathbf{b}}_j$ is a column vector comprised of all $b_j$ values. Therefore, when we take $\text{BASECONV}(\mathbf{u})$ we get

$$\text{BASECONV}(\mathbf{u}) = (\mathbf{u}\boldsymbol{W}) \odot \left(\boldsymbol{0}^{N \times d} * \mathbf{u} + \boldsymbol{B}^{(2)}\right)$$

$$= (\mathbf{u}\boldsymbol{W}) \odot \left(\boldsymbol{B}^{(2)}\right)$$

We can rewrite $\boldsymbol{B}^{(2)}$ as

$$\boldsymbol{B}^{(2)} = \left(\mathbf{1}^{N \times d}\right) \begin{pmatrix} b_0 & 0 & \dots & 0 \\ 0 & b_1 & \dots & 0 \\ & & \ddots & \\ 0 & 0 & \dots & b_{d-1} \end{pmatrix}.$$

let us call this diagonal matrix on the right, $\boldsymbol{D}$. Then note that by [Equation (55)](#)

$$\boldsymbol{W} = \overline{\boldsymbol{W}}\boldsymbol{D}^{-1}.$$

Therefore, we have

$$\text{BASECONV}(\mathbf{u}) = \mathbf{u}\overline{\boldsymbol{W}}\boldsymbol{D}^{-1} \odot \mathbf{1}^{N \times d}\boldsymbol{D}$$

$$= \mathbf{u}\overline{\boldsymbol{W}} \odot \mathbf{1}^{N \times d}$$

$$= \mathbf{u}\overline{\boldsymbol{W}},$$

as desired. In the above the second inequality follows since $\boldsymbol{D}$ is a diagonal matrix. $\qquad \square$

### D.5.8   MULTIPLY

Note that for $i \in [N]$, $j \in [d_{out}]$, the $i, j$-th entry of $\text{MULTIPLY}(a, b, d_{\text{out}})$ is

$$\text{MULTIPLY}(a, b, d_{\text{out}})_{i,j} = \mathbf{u}_{i,j+a} \cdot \mathbf{u}_{i,j+b}.$$

**Lemma D.61.** *Fix $i, j, a, b, c$ where $i \in [N]$ and $a, b, c \in [d - d_{out}]$ and $j \in [d_{out}]$. Then we have*

$$\mathbb{E}\left[(\mathbf{u}_{i,j+a} \cdot \mathbf{u}_{i,j+b}) \frac{\partial \boldsymbol{Z}_{i,j}}{\partial \boldsymbol{B}^{(2)}_{i,j+c}}\right] = 0. \tag{56}$$

*When $k > 0$:*

$$\mathbb{E}\left[(\mathbf{u}_{i,j+a} \cdot \mathbf{u}_{i,j+b}) \frac{\partial \boldsymbol{Z}_{i,j}}{\partial \boldsymbol{K}_{k,j+c}}\right] = 0. \tag{57}$$

*When $a = b$ then for all $c$:*

$$\mathbb{E}\left[(\mathbf{u}_{i,j+a} \cdot \mathbf{u}_{i,j+b}) \frac{\partial \boldsymbol{Z}_{i,j}}{\partial \boldsymbol{K}_{0,j+c}}\right] = \mathbb{E}\left[\mathbf{u}_{i,j+a}^2 \cdot \mathbf{u}_{i,j+c}^2\right] \boldsymbol{W}_{j+c,j+c}.$$

*When $a = c$ then for all $b$:*

$$\mathbb{E}\left[(\mathbf{u}_{i,j+a} \cdot \mathbf{u}_{i,j+b}) \frac{\partial \boldsymbol{Z}_{i,j}}{\partial \boldsymbol{K}_{0,j+c}}\right] = \mathbb{E}\left[\mathbf{u}_{i,j+c}^2 \cdot \mathbf{u}_{i,j+b}^2\right] \boldsymbol{W}_{j+b,j+c}. \tag{58}$$

*When $b = c$ then for all $a$:*

$$\mathbb{E}\left[(\mathbf{u}_{i,j+a} \cdot \mathbf{u}_{i,j+b}) \frac{\partial \boldsymbol{Z}_{i,j}}{\partial \boldsymbol{K}_{0,j+c}}\right] = \mathbb{E}\left[\mathbf{u}_{i,j+c}^2 \cdot \mathbf{u}_{i,j+a}^2\right] \boldsymbol{W}_{j+a,j+c}.$$

*For all other values of $a, b, c$ (i.e. $a, b, c$ are all distinct),:*

$$\mathbb{E}\left[(\mathbf{u}_{i,j+a} \cdot \mathbf{u}_{i,j+b}) \frac{\partial \boldsymbol{Z}_{i,j}}{\partial \boldsymbol{K}_{0,j+c}}\right] = 0.$$

*Next, when $\ell = j + a$ and $b = c$*

$$\mathbb{E}\left[(\mathbf{u}_{i,j+a} \cdot \mathbf{u}_{i,j+b}) \frac{\partial \boldsymbol{Z}_{i,j}}{\partial \boldsymbol{W}_{\ell,j+c}}\right] = \mathbb{E}\left[\mathbf{u}^2_{i,j+c} \cdot \mathbf{u}^2_{i,j+a}\right] \boldsymbol{K}_{0,j+c}.$$

*When $\ell = j + b$ and $a = c$,*

$$\mathbb{E}\left[(\mathbf{u}_{i,j+a} \cdot \mathbf{u}_{i,j+b}) \frac{\partial \boldsymbol{Z}_{i,j}}{\partial \boldsymbol{W}_{\ell,j+c}}\right] = \mathbb{E}\left[\mathbf{u}^2{}_{i,j+c} \cdot \mathbf{u}^2{}_{i,j+b}\right] \boldsymbol{K}_{0,j+c}. \tag{59}$$

*When $\ell = j + c$ and $a = b$,*

$$\mathbb{E}\left[(\mathbf{u}_{i,j+a} \cdot \mathbf{u}_{i,j+b}) \frac{\partial \boldsymbol{Z}_{i,j}}{\partial \boldsymbol{W}_{\ell,j+c}}\right] = \mathbb{E}\left[\mathbf{u}^2{}_{i,j+a} \cdot \mathbf{u}^2{}_{i,j+c}\right] \boldsymbol{K}_{0,j+c}.$$

*For all other values of $\ell, a, b, c$,*

$$\mathbb{E}\left[(\mathbf{u}_{i,j+a} \cdot \mathbf{u}_{i,j+b}) \frac{\partial \boldsymbol{Z}_{i,j}}{\partial \boldsymbol{W}_{\ell,j+c}}\right] = 0.$$

*Proof.* Let us begin with

$$\mathbb{E}\left[\mathbf{u}_{i,j+a} \cdot \mathbf{u}_{i,j+b} \frac{\partial \boldsymbol{Z}_{i,j}}{\partial \boldsymbol{B}^{(2)}_{i,j+c}}\right].$$

From Lemma D.54 and Lemma D.56 we can simplify this to the following (recall that $\boldsymbol{B}^{(1)} = \boldsymbol{0}$):

$$\mathbb{E}\left[\mathbf{u}_{i,j+a} \cdot \mathbf{u}_{i,j+b} \left(\langle \mathbf{u}_{i,:}^\top, \boldsymbol{W}_{:,j+c}\rangle\right) \frac{\partial}{\partial \boldsymbol{B}^{(2)}_{i,j+c}} \left(\boldsymbol{K}_{:,j+c} * \mathbf{u}_{:,j+c}[i] + \boldsymbol{B}^{(2)}_{i,j+c}\right)\right]$$
$$= \mathbb{E}\left[\mathbf{u}_{i,j+a} \cdot \mathbf{u}_{i,j+b} \left(\langle \mathbf{u}_{i,:}^\top, \boldsymbol{W}_{:,j+c}\rangle\right)\right].$$

This can be rewritten as the following

$$\sum_{\ell'=0}^{d-1} \mathbb{E}\left[\mathbf{u}_{i,j+a} \mathbf{u}_{i,j+b} \mathbf{u}_{i,\ell'}\right] \boldsymbol{W}_{\ell',j+c}.$$

From Assumption D.45 we know this is always 0 as there's an odd number of $\mathbf{u}$'s being multiplied together each iteration of the summation. Therefore we get

$$\mathbb{E}\left[\mathbf{u}_{i,j+a} \cdot \mathbf{u}_{i,j+b} \frac{\partial \boldsymbol{Z}_{i,j}}{\partial \boldsymbol{B}^{(2)}_{i,j+c}}\right] = 0.$$

Next, let us consider for $k \leq i$,

$$\mathbb{E}\left[(\mathbf{u}_{i,j+a} \cdot \mathbf{u}_{i,j+b}) \frac{\partial \boldsymbol{Z}_{i,j}}{\partial \boldsymbol{K}_{k,j+c}}\right].$$

From Lemma D.54 and Lemma D.56 we can simplify this to the following (recall that $\boldsymbol{B}^{(1)} = \boldsymbol{0}$):

$$\mathbb{E}\left[\mathbf{u}_{i,j+a} \cdot \mathbf{u}_{i,j+b} \left(\langle \mathbf{u}_{i,:}^\top, \boldsymbol{W}_{:,j+c}\rangle\right) \frac{\partial}{\partial \boldsymbol{K}_{k,j+c}} \left(\boldsymbol{K}_{:,j+c} * \mathbf{u}_{:,j+c}[i] + \boldsymbol{B}^{(2)}_{i,j+c}\right)\right]$$
$$= \mathbb{E}\left[\mathbf{u}_{i,j+a} \cdot \mathbf{u}_{i,j+b} \left(\langle \mathbf{u}_{i,:}^\top, \boldsymbol{W}_{:,j+c}\rangle\right) \mathbf{u}_{i-k,j+c}\right].$$

This can be rewritten as the following

$$\sum_{\ell'=0}^{d-1} \mathbb{E}\left[\mathbf{u}_{i,j+a} \mathbf{u}_{i,j+b} \mathbf{u}_{i,\ell'} \mathbf{u}_{i-k,j+c}\right] \boldsymbol{W}_{\ell',j+c}.$$

We have the following cases about the expected value of the above:

1. When $k > 0$ for any $\ell', a, b,$ or $c$ we get the expected value is 0.

2. When $a = b$, we get the expected value is $\mathbb{E}\left[\mathbf{u}^2_{i,j+a} \cdot \mathbf{u}^2_{i,j+c}\right] \boldsymbol{W}_{j+c,j+c}$ if and only if $k = 0$ and $\ell' = j + c$.

3. When $a = c$, we get the expected value is $\mathbb{E}\left[\mathbf{u}_{i,j+c}^2 \cdot \mathbf{u}_{i,j+b}^2\right] \boldsymbol{W}_{j+b,j+c}$ if and only if $k = 0$ and $\ell' = j + b$.

4. When $b = c$, we get the expected value is $\mathbb{E}\left[\mathbf{u}_{i,j+c}^2 \cdot \mathbf{u}_{i,j+a}^2\right] \boldsymbol{W}_{j+a,j+c}$ if and only if $k = 0$ and $\ell' = j + a$.

5. For all other values of $\ell', a, b$, and $c$, we get the expected value is $0$.

Via Assumption D.45, the reasoning for 1 and 5 is that there will always be an odd exponent on the $\mathbf{u}$'s. Then the reasoning for Items 2 to 4 is that there will always be an even exponent on the $\mathbf{u}$'s.

Next, let us consider,

$$
\mathbb{E}\left[\left(\mathbf{u}_{i,j+a} \cdot \mathbf{u}_{i,j+b}\right) \frac{\partial \boldsymbol{Z}_{i,j+c}}{\partial \boldsymbol{W}_{\ell,j+c}}\right].
$$

From Lemma D.54 and Lemma D.56 we can simplify this to the following:

$$
\mathbb{E}\left[\mathbf{u}_{i,j+a} \cdot \mathbf{u}_{i,j+b}\left(\boldsymbol{K}_{:,j+c} * \mathbf{u}_{:,j+c}[i] + \boldsymbol{B}_{i,j+c}^{(2)}\right) \frac{\partial}{\partial \boldsymbol{W}_{\ell,j+c}}\left(\langle \mathbf{u}_{i,:}^\top, \boldsymbol{W}_{:,j+c}\rangle\right)\right]
$$
$$
=\mathbb{E}\left[\mathbf{u}_{i,j+a} \cdot \mathbf{u}_{i,j+b}\left(\boldsymbol{K}_{:,j+c} * \mathbf{u}_{:,j+c}[i] + \boldsymbol{B}_{i,j+c}^{(2)}\right) \mathbf{u}_{i,\ell}\right]
$$

This can be rewritten as

$$
\sum_{k'=0}^{i}\left(\mathbb{E}\left[\mathbf{u}_{i,j+a}\mathbf{u}_{i,j+b}\mathbf{u}_{i-k',j+c}\mathbf{u}_{i,\ell}\right] \boldsymbol{K}_{k',j+c}\right) + \mathbb{E}\left[\mathbf{u}_{i,j+a}\mathbf{u}_{i,j+b}\mathbf{u}_{i,\ell}\right] \boldsymbol{B}_{i,j+c}^{(2)}
$$

This second term goes to zero via Assumption D.45 as there will always be an odd exponent on the term we take the expected value of. The first summation will be zero or non-zero given specific cases, just as we previously saw. Here they are:

1. When $k' > 0$, for any $\ell, a$, or $b$ we get that the expected value is $0$.

2. When $a = b$, we get the expected value is $\mathbb{E}\left[\mathbf{u}_{i,j+a}^2 \cdot \mathbf{u}_{i,j+c}^2\right] \boldsymbol{K}_{0,j+c}$ if and only if $k' = 0$ and $\ell = j + c$.

3. When $a = c$, we get the expected value is $\mathbb{E}\left[\mathbf{u}_{i,j+c}^2 \cdot \mathbf{u}_{i,j+b}^2\right] \boldsymbol{K}_{0,j+c}$ if and only if $k' = 0$ and $\ell = j + b$.

4. When $b = c$, we get the expected value is $\mathbb{E}\left[\mathbf{u}_{i,j+c}^2 \cdot \mathbf{u}_{i,j+a}^2\right] \boldsymbol{K}_{0,j+c}$ if and only if $k' = 0$ and $\ell = j + a$.

5. For all other values of $\ell, a, b$, and $c$, we get that the expected value is $0$.

Via Assumption D.45, the reasoning for 1 and 5 is that there will always be an odd exponent on the $\mathbf{u}$'s. Then the reasoning for Items 2 to 4 is that there will always be an even exponent on the $\mathbf{u}$'s. Each of these scenarios, covers the pieces in the lemma statement.

$\square$

Next, we restate Theorem D.50 and prove it:

**Theorem D.62** (Theorem D.50, restated). *Given Assumptions D.45, D.46, D.48, and a function*

$$
f(\mathbf{u}, a, b, d_{out}) = \mathbf{u}_{:,a:a+d_{out}-1} \odot \mathbf{u}_{:,b:b+d_{out}-1},
$$

*where $a, b \in [d - d_{out}]$ and without loss of generality assume $a \leq b$ then take $c = a$. Let $\boldsymbol{\theta}_0$ be such that, $\mathbb{E}\nabla_{\boldsymbol{\theta}}\overline{L}|_{\theta \leftarrow \theta_0} = \mathbf{0}$ then $\text{BASECONV}(\mathbf{u}, \boldsymbol{\theta}_0)[:, c : c + d_{out} - 1] = f(\mathbf{u})$.*

*Proof.* For $i, i' \in [N]$ and $j, j' \in [d_{out}]$ and $a, b, c$ defined in theorem statement, let us consider

$$\mathbb{E}\left[\frac{\partial \overline{L}}{\partial \boldsymbol{B}_{i,j+c}^{(2)}}\right] = \sum_{i'=0}^{N-1} \sum_{j'=0}^{d_{out}-1} \mathbb{E}\left[\frac{\partial \overline{L}_{i',j'}}{\partial \boldsymbol{B}_{i,j+c}^{(2)}}\right]$$

$$= \mathbb{E}\left[\frac{\partial \overline{L}_{i,j}}{\partial \boldsymbol{B}_{i,j+c}^{(2)}}\right],$$

where the second equality follows from Lemma D.53.

Recall our loss function from Equation (49). Then given Proposition D.47 we have

$$\mathbb{E}\left[\frac{\partial \overline{L}}{\partial \boldsymbol{B}_{i,j+c}^{(2)}}\right] = 2\,\mathbb{E}\left[\frac{\partial \boldsymbol{Z}_{i,j}}{\partial \boldsymbol{B}_{i,j+c}^{(2)}}\left(\boldsymbol{Z}_{i,j} - (\mathbf{u}_{i,j+a} \cdot \mathbf{u}_{i,j+b})\right)\right].$$

Simplifying and plugging in values from Lemma D.58 and Equation (56) from Lemma D.61 we get

$$2\,\mathbb{E}\left[\frac{\partial \boldsymbol{Z}_{i,j}}{\partial \boldsymbol{B}_{i,j+c}^{(2)}}\left(\boldsymbol{Z}_{i,j} - (\mathbf{u}_{i,j+a} \cdot \mathbf{u}_{i,j+b})\right)\right] = 2\,\mathbb{E}\left[\boldsymbol{Z}_{i,j}\frac{\partial \boldsymbol{Z}_{i,j}}{\partial \boldsymbol{B}_{i,j+c}^{(2)}} - (\mathbf{u}_{i,j+a} \cdot \mathbf{u}_{i,j+b})\frac{\partial \boldsymbol{Z}_{i,j}}{\partial \boldsymbol{B}_{i,j+c}^{(2)}}\right]$$

$$= 2\,\boldsymbol{B}_{i,j+c}^{(2)} \sum_{\ell'=0}^{d-1} \mathbb{E}\left[\mathbf{u}_{i,\ell'}^2\right] \boldsymbol{W}_{\ell',j+c}^2.$$

From Assumption D.46 we know the expected values of the squared input terms will be positive. Then from Assumption D.48 we know that at least one entry in $\boldsymbol{W}_{:,j+c}$ is non-zero. Therefore the above summation will be non-zero. Further implying, when we set

$$\mathbb{E}\left[\frac{\partial \overline{L}}{\partial \boldsymbol{B}_{i,j+c}^{(2)}}\right] = 0,$$

we can conclude that $\boldsymbol{B}_{i,j+c}^{(2)} = 0$ for all $i, j$. Explicitly,

$$\boldsymbol{B}_{:,c:c+d_{out}-1}^{(2)} = \boldsymbol{0}^{N \times d-1}. \tag{60}$$

Recall from Lemma D.52 that we only consider values of the parameters in column range $\{c, \dots, c + d_{out} - 1\}$. Therefore, moving forward, the $\boldsymbol{B}^{(2)}$ terms will be dropped from equations to simplify them. Next let us consider for all $k > 0$,

$$\mathbb{E}\left[\frac{\partial \overline{L}}{\partial \boldsymbol{K}_{k,j+c}}\right] = \sum_{i'=0}^{N-1} \sum_{j'=0}^{d_{out}-1} \mathbb{E}\left[\frac{\partial \overline{L}_{i',j'}}{\partial \boldsymbol{K}_{k,j+c}}\right]$$

$$= \sum_{i'=0}^{N-1} \mathbb{E}\left[\frac{\partial \overline{L}_{i',j}}{\partial \boldsymbol{K}_{k,j+c}}\right]$$

the second equality follows from Lemma D.53. Then from Equation (49) we get

$$\mathbb{E}\left[\frac{\partial \overline{L}}{\partial \boldsymbol{K}_{k,j+c}}\right] = \sum_{i'=0}^{N-1} 2\,\mathbb{E}\left[\frac{\partial \boldsymbol{Z}_{i',j}}{\partial \boldsymbol{K}_{k,j+c}}\left(\boldsymbol{Z}_{i',j} - (\mathbf{u}_{i',j+a} \cdot \mathbf{u}_{i',j+b})\right)\right].$$

Simplifying and plugging in values from Lemma D.58 and Equation (57) from Lemma D.61 we get

$$\sum_{i'=0}^{N-1} 2\,\mathbb{E}\left[\frac{\partial \boldsymbol{Z}_{i',j}}{\partial \boldsymbol{K}_{k,j+c}}\left(\boldsymbol{Z}_{i',j} - (\mathbf{u}_{i',j+a} \cdot \mathbf{u}_{i',j+b})\right)\right] = 2\sum_{i'=0}^{N-1} \boldsymbol{K}_{k,j+c} \sum_{\ell'=0}^{d-1} \boldsymbol{W}_{\ell',j+c}^2 \mathbb{E}\left[\mathbf{u}_{i',\ell'}^2 \cdot \mathbf{u}_{i'-k,j+c}^2\right]$$

$$= 2\,\boldsymbol{K}_{k,j+c}\left(\sum_{\ell'=0}^{d-1} \boldsymbol{W}_{\ell',j+c}^2 \sum_{i'=0}^{N-1} \mathbb{E}\left[\mathbf{u}_{i',\ell'}^2 \cdot \mathbf{u}_{i'-k,j+c}^2\right]\right).$$

From Assumption D.46 we know the expected values of the squared input terms will be positive. Then from Assumption D.48 we know that at least one entry in $\boldsymbol{W}_{:,j+c}$ is non-zero. Therefore the summation piece will always be non-zero. Therefore, when setting

$$\mathbb{E}\left[\frac{\partial \overline{L}}{\partial \boldsymbol{K}_{k,j+c}}\right] = 0,$$

we can conclude that $\boldsymbol{K}_{k,j+c} = 0$ for all $j$, $k > 0$. This implies that we have for all $j \in [d_{out}]$:

$$\boldsymbol{K}_{:,j+c} \neq \boldsymbol{0} \Leftrightarrow \boldsymbol{K}_{0,j+c} \neq 0. \tag{61}$$

Next let us consider,

$$\mathbb{E}\left[\frac{\partial \overline{L}}{\partial \boldsymbol{W}_{\ell,j+c}}\right] = \sum_{i'=0}^{N-1}\sum_{j'=0}^{d_{out}-1}\mathbb{E}\left[\frac{\partial \overline{L}_{i',j'}}{\partial \boldsymbol{W}_{\ell,j+c}}\right]$$

$$= \sum_{i'=0}^{N-1}\mathbb{E}\left[\frac{\partial \overline{L}_{i',j}}{\partial \boldsymbol{W}_{\ell,j+c}}\right]$$

The above follows from Lemma D.53. Then from Equation (49) we have

$$\mathbb{E}\left[\frac{\partial \overline{L}}{\partial \boldsymbol{W}_{\ell,j+c}}\right] = \sum_{i'=0}^{N-1} 2\,\mathbb{E}\left[\frac{\partial \boldsymbol{Z}_{i',j}}{\partial \boldsymbol{W}_{\ell,j+c}}\left(\boldsymbol{Z}_{i',j} - (\mathbf{u}_{i',j+a}\cdot\mathbf{u}_{i',j+b})\right)\right]$$

After simplifying and plugging in values from Lemma D.58 and Lemma D.61 (and recall that $\boldsymbol{B}^{(2)}[:,c:c+d_{out}-1] = \boldsymbol{0}$ and $a = c$) we get the following for $\ell \neq j + b$:

$$\sum_{i'=0}^{N-1} 2\,\mathbb{E}\left[\frac{\partial \boldsymbol{Z}_{i',j}}{\partial \boldsymbol{W}_{\ell,j+c}}\left(\boldsymbol{Z}_{i',j} - (\mathbf{u}_{i',j+a}\cdot\mathbf{u}_{i',j+b})\right)^2\right] = 2\sum_{i'=0}^{N-1}\boldsymbol{W}_{\ell,j+c}\sum_{k'=0}^{i'}\boldsymbol{K}_{k',j+c}^2\mathbb{E}\left[\mathbf{u}_{i'-k',j+c}^2\cdot\mathbf{u}_{i',\ell}^2\right]$$

$$= 2\,\boldsymbol{W}_{\ell,j+c}\sum_{i'=0}^{N-1}\sum_{k'=0}^{i'}\boldsymbol{K}_{k',j+c}^2\mathbb{E}\left[\mathbf{u}_{i'-k',j+c}^2\cdot\mathbf{u}_{i',\ell}^2\right].$$

From Assumption D.45 we know the expected values of squared input terms will be non-zero. Then from Equation (61) and Assumption D.48 we know that the 0-th entry of each column of $\boldsymbol{K}$ is non-zero. Therefore, this summation is always non-zero. Further we can say, when setting

$$\mathbb{E}\left[\frac{\partial \overline{L}}{\partial \boldsymbol{W}_{\ell,j+c}}\right] = 0,$$

we can conclude that $\boldsymbol{W}_{\ell,j} = 0$ for $\ell \neq j + b$. Explicitly,

$$\boldsymbol{W}_{:,j+c} \neq \boldsymbol{0} \Leftrightarrow \boldsymbol{W}_{j+b,j+c} \neq 0. \tag{62}$$

Now let us consider the following for $\ell = j + b$, by Lemma D.58 and Lemma D.61 (and recall that $\boldsymbol{B}^{(2)}[:,c:c+d_{out}-1] = \boldsymbol{0}$ and $a = c$)

$$2\sum_{i'=0}^{N-1}\mathbb{E}\left[\frac{\partial}{\partial \boldsymbol{W}_{j+b,j+c}}\left(\boldsymbol{Z}_{i',j} - (\mathbf{u}_{i,j+a}\cdot\mathbf{u}_{i',j+b})\right)\right]$$

$$= 2\left(\sum_{i'=0}^{N-1}\boldsymbol{W}_{j+b,j+c}\sum_{k'=0}^{i'}\boldsymbol{K}_{k',j+c}^2\mathbb{E}\left[\mathbf{u}_{i'-k',j+c}^2\cdot\mathbf{u}_{i',j+b}^2\right] - \mathbb{E}\left[\mathbf{u}_{i',j+c}^2\cdot\mathbf{u}_{i',j+b}^2\right]\boldsymbol{K}_{0,j+c}\right).$$

From Assumption D.48 and Equation (61) let us simplify the above to the following:

$$2\left(\sum_{i'=0}^{N-1}\boldsymbol{W}_{j+b,j+c}\boldsymbol{K}_{0,j+c}^2\mathbb{E}\left[\mathbf{u}_{i',j+c}^2\cdot\mathbf{u}_{i',j+b}^2\right] - \mathbb{E}\left[\mathbf{u}_{i',j+c}^2\cdot\mathbf{u}_{i',j+b}^2\right]\boldsymbol{K}_{0,j+c}\right)$$

$$= 2\left(\boldsymbol{W}_{j+b,j+c}\boldsymbol{K}_{0,j+c}^2\sum_{i'=0}^{N-1}\mathbb{E}\left[\mathbf{u}_{i',j+c}^2\cdot\mathbf{u}_{i',j+b}^2\right] - \boldsymbol{K}_{0,j+c}\sum_{i'=0}^{N-1}\mathbb{E}\left[\mathbf{u}_{i',j+c}^2\cdot\mathbf{u}_{i',j+b}^2\right]\right)$$

$$= 2\left(\boldsymbol{W}_{j+b,j+c}\boldsymbol{K}_{0,j+c}^2 - \boldsymbol{K}_{0,j+c}\right)\sum_{i'=0}^{N-1}\mathbb{E}\left[\mathbf{u}_{i',j+c}^2\cdot\mathbf{u}_{i',j+b}^2\right].$$

From Assumption D.45 we know the summation pieces are both always non-zero. Therefore when setting

$$
\mathbb{E}\left[\frac{\partial \overline{L}}{\partial \boldsymbol{W}_{j+b,j+c}}\right] = 0 \implies 2\left(\boldsymbol{W}_{j+b,j+c}\boldsymbol{K}_{0,j+c}^2 - \boldsymbol{K}_{0,j+c}\right) = 0
$$
$$
\implies \boldsymbol{W}_{j+b,j+c}\boldsymbol{K}_{0,j+c}^2 - \boldsymbol{K}_{0,j+c} = 0
$$
$$
\implies \boldsymbol{W}_{j+b,j+c}\boldsymbol{K}_{0,j+c}^2 = \boldsymbol{K}_{0,j+c}
$$
$$
\implies \boldsymbol{W}_{j+b,j+c}\boldsymbol{K}_{0,j+c} = 1.
$$

In the above the last equality follows form the fact that $\boldsymbol{K}_{0,j+c} \neq 0$. Thus, the above gives us

$$
\boldsymbol{K}_{0,j+c} = \frac{1}{\boldsymbol{W}_{j+b,j+c}}. \tag{63}
$$

Note that this is a valid assignment since Equation (62) and Assumption D.48 implies $\boldsymbol{W}_{j+b,j+c} \neq 0$. Therefore, given Assumption D.48 and the above values; $\boldsymbol{W}_{j+b,j+c} \neq 0$ for all $j \in [d_{out}]$ and $\boldsymbol{W}_{i',j'} = 0$ for all other $(i',j')$. Note that a multiplication on the right of $\mathbf{u}$ with this lower left shift matrix will shift the input to the left by $b - c$. And we have $\boldsymbol{K}_{:,c:c+d_{out}-1} = \left(\begin{smallmatrix}\frac{1}{\boldsymbol{W}_{j+b,j+c}} \cdots \frac{1}{\boldsymbol{W}_{j+b+d_{out}-1,j+c+d_{out}-1}} \\ \mathbf{0}^{N-1 \times d_{out}}\end{smallmatrix}\right)$, $\boldsymbol{B}_{:,c:c+d_{out}-1}^{(2)} = \mathbf{0}^{N \times d_{out}}$. Let us use these pieces to show that $\text{BASECONV}(\mathbf{u})[:,c:c+d_{out}-1] = \mathbf{u}_{:,a:a+d_{out}-1} \odot \mathbf{u}_{:,b:b+d_{out}-1}$ (without loss of generality, where $c = a$).

Recall that $\boldsymbol{B}^{(1)} = \mathbf{0}$. Then indeed,

$$
\text{BASECONV}(\mathbf{u})[:,c:c+d_{out}-1] = (\mathbf{u} \cdot \boldsymbol{W})_{:,c:c+d_{out}-1} \odot \left(\boldsymbol{K}_{:,c:c+d_{out}-1} * \mathbf{u}_{:,c:c+d_{out}-1}\right).
$$

Plugging in our values we get

$$
\left(\mathbf{u} \cdot \begin{pmatrix} 0 & 0 & 0 & 0 & 0 & 0 & 0 & 0 & 0 \\ 0 & 0 & 0 & \vdots & \vdots & \vdots & 0 & 0 & 0 \\ \vdots & & 0 & 0 & 0 & & \vdots \\ & \boldsymbol{W}_{(b,c)} & 0 & & 0 \\ & 0 & \ddots & 0 \\ & 0 & 0 & \boldsymbol{W}_{(b+d_{out}-1,c+d_{out}-1)} \\ \vdots & 0 & 0 & 0 & \vdots \\ 0 & 0 & 0 & \vdots & \vdots & \vdots & 0 & 0 & 0 \\ 0 & 0 & 0 & 0 & 0 & 0 & 0 & 0 & 0 \end{pmatrix}\right)[:,c:c+d_{out}-1]
$$
$$
\odot \left(\left(\left(\begin{pmatrix}\frac{1}{\boldsymbol{W}_{(b,c)}} \cdots \frac{1}{\boldsymbol{W}_{(b+d_{out}-1,c+d_{out}-1)}} \\ \mathbf{0}^{N-1 \times d_{out}}\end{pmatrix}\right) * \mathbf{u}\right)[:,c:c+d_{out}-1] + \mathbf{0}^{N \times d_{out}}\right).
$$

Let's define

$$
\boldsymbol{D} \stackrel{\text{def}}{=} \begin{pmatrix} \boldsymbol{W}_{(b,c)} & 0 & 0 \\ 0 & \ddots & 0 \\ 0 & 0 & \boldsymbol{W}_{(b+d_{out}-1,c+d_{out}-1)} \end{pmatrix}.
$$

Then we can say

$$\mathbf{u}\cdot\begin{pmatrix} 0 & 0 & 0 & 0 & 0 & & 0 & & 0 & 0 & 0 \\ 0 & 0 & 0 & \vdots & \vdots & & \vdots & & 0 & 0 & 0 \\ & \vdots & & 0 & 0 & & 0 & & & \vdots & \\ & & \boldsymbol{W}_{(b,c)} & 0 & & 0 & & & & & \\ & & & 0 & \ddots & & 0 & & & & \\ & & & 0 & 0 & \boldsymbol{W}_{(b+d_{out}-1,c+d_{out}-1)} & & & & & \\ & \vdots & & 0 & 0 & & 0 & & & \vdots & \\ 0 & 0 & 0 & \vdots & \vdots & & \vdots & & 0 & 0 & 0 \\ 0 & 0 & 0 & 0 & 0 & & 0 & & 0 & 0 & 0 \end{pmatrix}[:,c:c+d_{out}-1] = \mathbf{u}_{:,b:b+d_{out}-1}\cdot\boldsymbol{D}$$

Also note that

$$\boldsymbol{K}_{:,c:c+d_{out}-1} * \mathbf{u}_{:,c:c+d_{out}-1} = \mathbf{u}_{:,c:c+d_{out}-1}\begin{pmatrix} \frac{1}{\boldsymbol{W}_{(b,c)}} & 0 & 0 \\ 0 & \ddots & 0 \\ 0 & 0 & \frac{1}{\boldsymbol{W}_{(b+d_{out}-1,c+d_{out}-1)}} \end{pmatrix}$$

$$= \mathbf{u}_{:,c:c+d_{out}-1} \cdot \boldsymbol{D}^{-1}$$

which gives us

$$(\mathbf{u}\boldsymbol{W} \odot \boldsymbol{K} * \mathbf{u})[:,c:c+d_{out}-1] = (\mathbf{u}_{:,b:b+d_{out}-1}\cdot\boldsymbol{D}) \odot (\mathbf{u}_{:,c:c+d_{out}-1}\cdot\boldsymbol{D}^{-1})$$

$$= \mathbf{u}_{:,b:b+d_{out}-1} \odot \mathbf{u}_{:,c:c+d_{out}-1}$$

$$= \mathbf{u}_{:,b:b+d_{out}-1} \odot \mathbf{u}_{:,a:a+d_{out}-1}.$$

Where in the last equality we used the fact that $a = c$. Therefore, we have shown that the gradients of the expected loss function is 0,

$$\text{BASECONV}(\mathbf{u})[:,c:c+d_{out}] = \mathbf{u}_{:,a:a+d_{out}-1} \odot \mathbf{u}_{:,b:b+d_{out}-1}$$

as desired. For the case of $c = b$, the proof remains the same, just values for $a$ and $b$ are swapped where necessary. □

A corollary of the above is that MULTIPLY implements the SQUARE function

**Corollary D.63.** *Given Assumptions D.45, D.46, D.48, and a function*

$$f(\mathbf{u}) = \mathbf{u} \odot \mathbf{u}.$$

*Let $\boldsymbol{\theta}_0$ be such that, $\mathbb{E}\nabla_{\boldsymbol{\theta}}\overline{L}|_{\theta\leftarrow\theta_0} = \mathbf{0}$ with $c = 0$ and $d_{out} = d$. Then $\text{BASECONV}(\mathbf{u},\boldsymbol{\theta}_0,0,d) = f(\mathbf{u})$.*

*Proof.* The proof follows when we have values $c = 0$ and $d_{out} = d$ for the Theorem D.62. □

We now revisit the importance of Assumption D.48. Specifically, the following definition is a stronger version of the complement of Assumption D.48. The following essentially states that there are many ways to get the expected gradients of the loss function to be 0, though this doesn't imply that we have learned the exact solution, as we recover in Corollary D.63.

**Definition D.64.** Define $(\text{assumption})^{\complement}$ to be

- $\boldsymbol{B}^{(2)} = \mathbf{0}$

- For all $j \in [d]$, either

    (i) $\boldsymbol{W}_{:,j} = \mathbf{0}$ and $\boldsymbol{K}_{0,j} = 0$

(ii) $\boldsymbol{K}_{:,j} = \boldsymbol{0}$ and $\boldsymbol{W}_{j,j} = 0$

The following theorem is to emphasize, there are many ways to get expected value of the gradients of the loss function to be 0.

**Theorem D.65.** *Let $\theta^*$ satisfy Definition D.64. Then, $\mathbb{E}\nabla_\theta \overline{L}\big|_{\theta \leftarrow \theta^*} = \boldsymbol{0}$ when $f(\mathbf{u}) = \mathbf{u} \odot \mathbf{u}$ (where $c = 0$ and $d_{out} = d$).*

*Proof.* This proof considers values of $j \in [d]$. Via Lemma D.58 and Lemma D.61 when $k > 0$ we have

$$\mathbb{E}\left[\frac{\partial \overline{L}}{\partial \boldsymbol{K}_{k,j}}\right] = \boldsymbol{K}_{k,j} \sum_{i'=0}^{N-1} \sum_{\ell'=0}^{d-1} \boldsymbol{W}_{\ell',j}^2 \mathbb{E}\left[\mathbf{u}_{i',\ell'}^2 \cdot \mathbf{u}_{i'-k,j}^2\right].$$

This expected value goes to zero since every column $j$ either $\boldsymbol{K}_{:,j}$ or $\boldsymbol{W}_{:,j}$ is $\boldsymbol{0}$.

When $k = 0$ we have

$$\mathbb{E}\left[\frac{\partial \overline{L}}{\partial \boldsymbol{K}_{0,j}}\right] = \boldsymbol{W}_{j,j}\left(\boldsymbol{K}_{0,j}\boldsymbol{W}_{j,j} - 1\right)\left(\sum_{i'=0}^{N-1} \mathbb{E}\left[\mathbf{u}_{i',j}^4\right]\right).$$

This expected value goes to zero since in (i) and (ii) from Definition D.64, $\boldsymbol{K}_{0,j} = \boldsymbol{W}_{j,j} = 0$.

Next we have for all $\ell, j$ where $\ell \neq j$,

$$\mathbb{E}\left[\frac{\partial \overline{L}}{\partial \boldsymbol{W}_{\ell,j}}\right] = \boldsymbol{W}_{\ell,j} \sum_{i'=0}^{N-1} \mathbb{E}\left[\mathbf{u}_{i',\ell}^2\right]\left(\boldsymbol{K}_{0,j}^2 \mathbb{E}\left[\mathbf{u}_{i',j}^2\right] + \boldsymbol{B}_{i',j}^{(2)}\right)$$

This expected value goes to zero since column $j$ either $\boldsymbol{K}_{:,j}$ or $\boldsymbol{W}_{:,j}$ is $\boldsymbol{0}$.

Then when $\ell = j$ we have

$$\mathbb{E}\left[\frac{\partial \overline{L}}{\partial \boldsymbol{W}_{j,j}}\right] = \sum_{i'=0}^{N-1} \boldsymbol{W}_{j,j}\boldsymbol{K}_{0,j}^2 \mathbb{E}\left[\mathbf{u}_{i',j}^4\right] - \boldsymbol{K}_{0,j}\mathbb{E}\left[\mathbf{u}_{i',j}^4\right]$$

This expected value goes to zero since in (i) and (ii) from Definition D.64, $\boldsymbol{K}_{0,j} = \boldsymbol{W}_{j,j} = 0$

And we know that since $\boldsymbol{B}^{(2)}$ is all zeros, we don't need to consider the gradient of the loss function to it. $\qquad\square$

What the above proves is that there are infinite instantiations of parameters such that the expected gradient loss is 0. However note that in Definition D.64, for all $j$ either $\boldsymbol{K}_{k,j}$ for $k \neq 0$ or $\boldsymbol{W}_{\ell,j}$ for $\ell \neq j$ are unconstrained. In other words, we can set these values arbitrarily, which means we get $\overline{L} \to \inf$ but we still have the expected gradient loss to be $\boldsymbol{0}$. This shows that some form of Assumption D.48 is necessary to prove Corollary D.63.

