# OpenReview forum: "Towards Learning High-Precision Least Squares Algorithms with Sequence Models"
_ICLR.cc/2025/Conference — ICLR 2025 Poster_

### Official Review · Reviewer_XPf9 · 2024-10-16

**Soundness:** 3
**Presentation:** 3
**Contribution:** 3
**Rating:** 6
**Confidence:** 2

**Summary:**

This paper investigates whether sequence models can learn to perform numerical algorithms. This paper finds that Transformer fails to meet criteria, such as machine precision and numerical generality, and instead propose alternative architectures that overcomes these limitations.

**Strengths:**

The strength of this paper is not only showing the limitation of Transformer, but also proposing alternative architecture that experimentally makes progress toward learning to solve numerical algorithms. Furthermore, there are several theoretical analyses to support experiments. For instance, it is interesting to show that, in Theorem D.34, single-head causal softmax attention cannot represent a square function.

**Weaknesses:**

As mentioned in Discussion and Limitations, proposed methods seem to be challenging in maintaining stable and precise training with deep networks.

**Questions:**

Out of interest, could you explore some theoretical analysis of proposed methods such as universal approximation or convergence of training? For instance, similarly in [Yun et al 2020] that shows the universality of Transformer, can we (expect to) show the universality of proposed architecture?

---

> ### Author Response · Authors · 2024-11-24
> **Response to Reviewer XPf9**
>
> Thank you for your thoughtful review! We appreciated that you found our experimental results with polynomial-based architectures promising and our theoretical results interesting.
>
> We’d like to highlight our common response, where we contextualize the contributions of our work within the literatures of the in-context learning and scientific ML communities. Briefly:
> - Our observations that Transformers fail to learn precise and general solutions **directly challenge the prevailing hypothesis about in-context learning in the literature.** This suggests that Transformers must be learning qualitatively different algorithms than gradient descent, with important implications for how we understand and potentially improve in-context learning.
> - As far as we are aware, **we are the first to investigate and isolate effects of the model architecture and optimizer on precision in a controlled setting.** Our analysis of least squares serves as an essential testbed for understanding fundamental precision limitations that impact more complex numerical tasks, e.g. solving differential equations.
>
> Please refer to the common response for a more detailed discussion. Below, we address your questions and comments.
>
> ## W1: Stable training with deep networks remains a bottleneck
> Thanks for this point! We agree that scaling to deeper networks is the key bottleneck to learning more complex numerical algorithms.
>
> We want to highlight that disentangling the effects of model depth, architecture, and optimizer is non-trivial and comprises an important contribution of our work. Although precision has been a known challenge in the SciML literature (e.g. refer to [1], a survey of 76 works in this space), our work is, as far as we can tell, the first that investigates the separate effects of architecture and optimizer on precision in a simple controlled setting: least squares. We think the observation that scaling can detrimentally impact precision, even with a theoretically expressive architecture, is a surprising and important takeaway from our work, which the wider ML community may find interesting.
>
> ## Q1: Universal approximation and training dynamics for BaseConv
> Thanks for this thoughtful question! We address these two points below:
>
> _Universal approximation:_
>
> Universal approximation of our architecture in fact follows from Proposition D.43 (now D.44 in the revision), which proves that the BaseConv architecture can efficiently approximate arbitrary smooth multivariate functions using the universal approximation property of polynomials. Previously, this result was relegated to Appendix D.4. We have revised the manuscript to include a clearer discussion of this result in Section 4.1.
>
> _Training dynamics:_
>
> Thank you for this interesting question! We are also very interested in obtaining theoretical results about the training dynamics of BaseConv. However, we point out that the equivalent result for Transformers is rare except for simplified models -- the only such result we are aware of examines 1-layer non-causal linear attention under gradient flow [2] -- and represents an entire effort on its own. As such, we leave this question to future work.
>
> As an aside, we want to emphasize that although results of the form “exact solution implies zero population gradient” exist in the literature [3,4], the _converse_ (“zero population gradient implies recovery of exact solution”) is rare. Our existing results roughly say that BaseConv perfectly recovers the Square and Linear primitives if its population gradient is zero. **To the best of our knowledge, ours is the first such result** for sequence model architectures (such as Transformers) that have been used to create large language models. We emphasize this point in the revision of Section 4.1.
>
> We thank you again for your insightful questions, which we believe have improved the clarity of our work!
>
> **References:**
> 1. Nick McGreivy, Ammar Hakim. Weak baselines and reporting biases lead to overoptimism in machine learning for fluid-related partial differential equations. Nature Machine Intelligence, 2024.
> 2. Ruiqi Zhang, Spencer Frei, and Peter L Bartlett. Trained transformers learn linear models in-context. JMLR 2023.
> 3. Kwangjun Ahn, Xiang Cheng, Hadi Daneshmand, Suvrit Sra. Transformers learn to implement preconditioned gradient descent for in-context learning. NeurIPS 2023.
> 4. Arvind Mahankali, Tatsunori B. Hashimoto, Tengyu Ma. One Step of Gradient Descent is Provably the Optimal In-Context Learner with One Layer of Linear Self-Attention. ICLR 2024.

---

> > ### Comment · Reviewer_XPf9 · 2024-11-25
> >
> > Thank you very much for your reply. I will maintain my score.

---

> > > ### Author Response · Authors · 2024-11-25
> > >
> > > Thank you for reviewing our response! We were excited to address your thoughtful questions about universal approximation and training dynamics, including highlighting our universal approximation result (Proposition D.43) and discussing the challenges of training dynamics analysis. We would appreciate hearing if there are specific aspects of these topics that you feel need further clarification or development. Your feedback would be valuable for improving the final version of our paper.

---

> > > > ### Author Response · Authors · 2024-12-01
> > > > **Follow-up**
> > > >
> > > > Dear Reviewer XPf9,
> > > >
> > > > Thank you again for reviewing our work! As the discussion period ends shortly, we wanted to check if you have any further questions or found our responses helpful?
> > > >
> > > > Please let us know and thank you for your time!

---

### Official Review · Reviewer_TtTT · 2024-10-26

**Soundness:** 3
**Presentation:** 3
**Contribution:** 3
**Rating:** 6
**Confidence:** 3

**Summary:**

This paper observes empirically that standard (softmax-attention-based) transformers cannot achieve machine precision or numerical generality for solving linear regression and proposes an alternative polynomial architecture based on the BaseConv module that achieves nearly machine precision and numerical generality close to that of gradient descent (GD).

**Strengths:**

- The observations that standard transformers cannot solve linear regression to machine precision and are brittle to changes in distribution are interesting as they discredits a hypothesis supported by a plethora of recent literature that simplified and standard transformers learn to perform GD on linear regression problems. This is especially the case for the machine precision observation, as it is novel as far as I know (brittleness to changes in distribution has been observed elsewhere, e.g. [1] below).

- Theoretical and experimental claims are well-substantiated. The experiments are thorough and well-executed within the scope of the problem.

- The paper is very well-written.

**Weaknesses:**

While I think that observation that standard transformers are not learning to perform GD or Newton's method on linear regression is an important and relevant contribution, I'm not sure that coming up with a new architecture and developing a training procedure for solving linear regression to machine precision is of interest to the ICLR community, for the following reasons:

- Outside of scientific ML, solving linear regression to machine precision is not of interest. The paper does not discuss whether the proposed model and training procedure can shed insights on other problems besides solving linear regression to MP.

- Within scientific ML, it is not clear why practitioners would want to solve linear regression with a large and costly model that requires 50-80 layers to solve even small-scale problems to machine precision (as suggested by the experiments) rather than simply running GD.

-  I can understand why it is interesting from a theoretical perspective, however to cater to a theoretical audience, I think the paper can use more theoretical development. In particular, I think the paper should move their proof of the inability of softmax attention to express multiplication from the appendix to the main body, and further develop this theory to show that standard transformers cannot express GD. Of the two theoretical results currently in the paper, one is borrowed from another paper, and the other is an informal statement that does not fully show the claimed expressivity of their proposed BaseConv model, since it does not show that BaseConv can achieve zero population gradient.

(Minor) Missing related works:

[1] Zhang et al., Trained Transformers Learn Linear Models In-Context, JMLR 2024 -- Like Ahn et al. 2024, shows that linear transformers learn GD.

[2] Collins et al., In-Context Learning with Transformers: Softmax Attention Adapts to Function Lipschitzness, arxiv 2024 -- (and Related Works therein) -- also argues that standard transformers are not solving regression with GD.

**Questions:**

n/a

---

> ### Author Response · Authors · 2024-11-24
> **Response to Reviewer TtTT (Part 1)**
>
> Thank you for your thoughtful and detailed review! We appreciated that you found our observations insightful and our experimental and theoretical results thorough and well-substantiated. Below, we address your questions and comments.
>
> ## W1: Transferring our insights beyond linear regression
> This is an important point -- thank you for highlighting it! We focus on least squares as a first step for two key reasons:
> - **High precision is crucial for scientific ML**, where small errors can compound catastrophically (e.g., in climate modeling and turbulence simulations). Least squares provides a controlled setting to examine fundamental precision limitations. (We further discuss this point in our response to **W2** below.)
> - **Least squares serves as a useful primitive for differential equations.** As a first step towards exploring these connections, we include new experimental results on solving ODEs in-context in Appendix C.4. Our insights about limited precision and generality transfer directly to the ODE setting, whereas our proposed techniques achieve high precision.
> Please refer to the common response for more details.
>
> ## W2: Practicality of our least squares techniques
> Thanks for this comment! We agree that learning to perform gradient descent with ML on least squares may be of limited practicality for its own sake. Rather, we think our results may be interesting to the broader scientific ML community as a _fundamental analysis of precision bottlenecks_ within typical training setups. In our common response, we discuss the importance of high precision for SciML and contextualize our contributions within the existing SciML literature. Briefly, as far as we are aware, **we are the first to investigate and isolate effects of the model architecture and optimizer on precision in a controlled setting.** Please refer to the common response for a more detailed discussion.
>
> ## W3: Theoretical results are not comprehensive (e.g. proving BaseConv will train to zero population gradient)
> Thank you for this thoughtful and detailed comment! We first make one clarifying comment, then address the points you raise:
>
> _Key theoretical result (Theorem 4.1) is borrowed from another paper (Theorem H.21 of [1])_:
>
> This is an important point that deserves a clearer discussion in the manuscript -- thank you for bringing it up! It’s true that we use the statement of Theorem H.21 from [1], showing the equivalence of arithmetic circuits and the BaseConv architecture, as key theoretical grounding for our method. However, we want to highlight that for the class of numerical algorithms we consider in this work, **our theoretical results are stronger**. Specifically, while Theorem H.21 implies a poly-log-factor increase in parameters (specifically the number of layers) translating from general arithmetic circuits to BaseConv, in our work we show by construction that the primitives Read, Linear, and Multiply _incur only a **constant** factor loss_ -- this improvement is especially important in the context of number of layers that we need. Since the numerical algorithms we consider are compositions of these primitives, this poly-log savings also holds for GD and Newton’s method. We emphasize this point in the revision of Section 4.1, and move the result we borrow from [1] to Appendix D.
>
>
> _Highlighting that softmax attention cannot express multiply:_
>
> Due to space limitations, we are unable to move the proof of this key result to the main text. However, in the revision, we now include a proper theorem statement for this result in Section 3.3, with a pointer to the full proof in Appendix D.2.4. We hope this improves the clarity of the main text!

---

> ### Author Response · Authors · 2024-11-24
> **Response to Reviewer TtTT (Part 2)**
>
> _Proving softmax attention cannot express gradient descent on least squares:_
>
> Thanks for this interesting question! We believe the inexpressibility of Square and Multiply tasks is the “right” level of abstraction to anchor on, in that we believe it is the *most fundamental* observation that accounts for why softmax attention Transformers struggle to precisely learn numerical algorithms like GD in practice:
> - **Our proofs for Square and Multiply capture the main theoretical observation: softmax cannot express polynomials.** At a high level, our proofs about the expressibility of Square and Multiply with softmax attention reduce to the simple but important observation that softmax can’t exactly express polynomials (e.g. $u_1^2$ or $u_1 u_2$ for real-valued inputs $u_1$, $u_2$). We believe that a proof for the inexpressibility of e.g. GD for least squares would reduce to the same basic observation. In our revision, we now include a brief discussion of this intuition in Section 3.3.
> - **Our primitives are fundamental.** We show in Appendix D.2 that basic numerical algorithms, GD and Newton’s method, can be expressed as compositions of three linear algebra primitives: Read, Linear, and Multiply. The fact that Transformers struggle to learn the simple Multiply primitive provides strong evidence that Transformers would also struggle to learn numerical algorithms that require _compositions_ of primitives, including Multiply.
> - **Our theory results are supported by empirical results.** We show experimentally that Transformers struggle to precisely learn the Multiply synthetic, even when scaling in depth, width, and training iterations (Figures 7 and 8 in Appendix C.1).
>
> We agree that it would be interesting to work towards a result that “softmax Transformers cannot implement GD for least squares”. However, we believe such a proof would only hold for a specific formatting/prompting of the least squares task, and the proof idea would be to show that GD for least squares is equivalent to computing a polynomial of inputs, which softmax cannot express exactly. One result we believe should be possible to prove with our techniques is that “1-layer causal softmax attention cannot implement _our specific task formulation_ of GD for least squares”. Due to the limited discussion period, we chose to spend our time obtaining additional experimental results, but we would be happy to commit to producing the detailed proof of such a result if the reviewer believes it would be interesting to the community.
>
> _Proving BaseConv can train to zero population gradient:_
>
> This is also a great point -- thank you for highlighting this important detail! Our existing results roughly say that BaseConv perfectly recovers the Square and Linear primitives if its population gradient is zero. We want to emphasize that although results of the form “exact solution implies zero population gradient” exist in the literature [2,3], the _converse_ (“zero population gradient implies recovery of exact solution”) is rare. **To the best of our knowledge, ours is the first such result** for sequence model architectures (such as Transformers) that have been used to create large language models. We emphasize this point in the revision of Section 4.1.
>
> We are also very interested in obtaining theoretical results about the training dynamics of BaseConv. However, we point out that the equivalent result for Transformers is rare except for simplified models -- the only such result we are aware of examines 1-layer non-causal linear attention under gradient flow [4] -- and represents an entire effort on its own. As such, we leave this question to future work.
>
> In response to your detailed and insightful comments, we believe the presentation and clarity of Section 4 has been greatly strengthened -- thanks again!
>
> ## W4: Related work
> Thank you for pointing out these missing related works! We now include them in the extended related work section of Appendix A.2.
>
> **References:**
> 1. Simran Arora, Sabri Eyuboglu, Aman Timalsina, Isys Johnson, Michael Poli, James Zou, Atri Rudra, Christopher Ré. Zoology: Measuring and Improving Recall in Efficient Language Models. ICLR 2024.
> 2. Kwangjun Ahn, Xiang Cheng, Hadi Daneshmand, Suvrit Sra. Transformers learn to implement preconditioned gradient descent for in-context learning. NeurIPS 2023.
> 3. Arvind Mahankali, Tatsunori B. Hashimoto, Tengyu Ma. One Step of Gradient Descent is Provably the Optimal In-Context Learner with One Layer of Linear Self-Attention. ICLR 2024.
> 4. Ruiqi Zhang, Spencer Frei, and Peter L Bartlett. Trained transformers learn linear models in-context. JMLR 2023.

---

> > ### Author Response · Authors · 2024-12-01
> > **Follow-up**
> >
> > Dear Reviewer TtTT,
> >
> > Thank you again for reviewing our work! As the discussion period ends shortly, we wanted to check if you have any further questions or found our responses helpful?
> >
> > Please let us know and thank you for your time!

---

### Official Review · Reviewer_frTT · 2024-10-27

**Soundness:** 3
**Presentation:** 3
**Contribution:** 2
**Rating:** 6
**Confidence:** 3

**Summary:**

The paper shows that the in-context learning setup for transformers cannot learn linear regression solutions with machine precision or numerical generality, and identify architectural and optimization factors that contribute to this phenomenon. It then proposes a new recipe for training to machine precision, including using BaseConv architecture, explicitly learning gradients, and reducing stochastic gradient noise.

**Strengths:**

The paper addresses a novel idea, namely learning high-precision algorithms with transformers. Their proposed training setup is well motivated by empirical observations.

**Weaknesses:**

1. Not all the notation needed to understand the main text is defined in the main text itself, e.g. What are the arguments to MULTIPLY? This makes some parts hard to understand without jumping to the appendix.

2. As part of their training setup, the authors propose to learn GD iterates explicitly, which requires knowing the value of the gradient. It is not clear to me why this setup is comparable to the standard in-context setup, which never explicitly computes gradients of the regression instance, or how this scenario could arise in practice.

3. It is unclear whether the proposed setup can generalize to practical scenarios beyond simple linear regression.

**Questions:**

See weaknesses.

---

> ### Author Response · Authors · 2024-11-24
> **Response to Reviewer frTT (Part 1)**
>
> Thank you for your thoughtful review! We appreciate that you found the high-precision algorithm learning setting novel and found our proposed techniques well-motivated.
>
> We’d like to highlight our common response, where we contextualize the contributions of our work within the literatures of the in-context learning and scientific ML communities. Briefly:
> - Our observations that Transformers fail to learn precise and general solutions **directly challenge the prevailing hypothesis about in-context learning in the literature.** This suggests that Transformers must be learning qualitatively different algorithms than gradient descent, with important implications for how we understand and potentially improve in-context learning.
> - As far as we are aware, **we are the first to investigate and isolate effects of the model architecture and optimizer on precision in a controlled setting.** Our analysis of least squares serves as an essential testbed for understanding fundamental precision limitations that impact more complex numerical tasks, e.g. solving differential equations.
>
> Please refer to the common response for a more detailed discussion. Below, we address your questions and comments.
>
>
> ## W1: Clarity of notation in the main text (e.g. definition of the Multiply primitive)
>
> Thank you for this comment! Towards making the main text more easily understandable on its own, we make the following changes to the manuscript:
> - We move the definitions of the Read, Linear, and Multiply primitives to Section 3.3.
> - We now introduce and discuss the connections between the simplified training setups we propose, explicit gradient descent (GD) and $k$-step GD, at the beginning of Section 5.
> We hope these changes improve the clarity of the work!
>
> ## W2.1: Relation between the explicit gradient task and standard in-context least squares
>
> This is an important point that deserves a clearer discussion in the manuscript – thank you for bringing it up!
>
> The explicit gradient task and standard in-context least squares are connected via the multistep GD task. Specifically, given the standard GD iterates
>
> $$\mathbf{x}\_{i+1} := \mathbf{x}\_{i} - \eta \nabla \mathcal{L}(\mathbf{x}\_i),$$
>
> the goal of the $k$-step GD task is to predict $\mathbf{x}_k$ given inputs $(\mathbf{A}, \mathbf{b}, \mathbf{x}_0)$. Note that (up to a residual connection), the explicit gradient task is equivalent to $1$-step GD, and standard in-context least squares is equivalent to taking $k \to \infty$ (though in practice, Figure 4 suggests $k \approx 80$ is equivalent, up to machine precision). Thus, we view the explicit gradient task as a natural simplification of standard in-context least squares, where the $k$ parameter allows us to smoothly interpolate between the two extremes of difficulty. Note that at the end of Section 5, we show that our proposed techniques allow us to scale to $k=4$, and using our controlled setting, we identify _model depth_ as the remaining bottleneck to learning more complex algorithms.
>
> Previously, we briefly mentioned the $k$-step GD task at the end of Section 5, relegating a detailed definition to Appendix B.3.4 and discussing experimental results in Appendix C.3. We have added a brief discussion of these connections to the start of Section 5. Thanks again for raising this question -- we hope this discussion more clearly motivates the simplified training setups we investigate!

---

> > ### Author Response · Authors · 2024-11-24
> > **Response to Reviewer frTT (Part 2)**
> >
> > ## W2.2: Simplicity of the explicit GD training setup
> > Thank you for your comments about the explicit GD task! We agree that the explicit GD task we investigate is simpler than the standard in-context learning regression setup. We argue our results may be interesting to the ML community because it is _precisely the simplicity of this task_ that highlights fundamental limitations of the Transformer architecture.
> >
> > Our initial observation that Transformers struggle to precisely solve the standard in-context least squares task motivated us to explore a simplified task, towards disentangling the effects of the training setup, the architecture, and the optimizer. Surprisingly though, we find that **Transformers still fail to achieve high precision on the much simpler explicit gradient task.** We summarize our existing results (Figures 1 and 9 in the manuscript) in Table 1, where we compare the highest-precision models we are able to train on in-context least squares vs. explicit gradient using Transformers vs. BaseConv, our polynomial architecture.
> >
> > | Method      | In-context LS | Explicit GD |
> > |:------------|:-------------|:------------|
> > | Transformer | $5.7 \times 10^{-9}$ | $1.9 \times 10^{-2}$ |
> > | BaseConv    | $2.0 \times 10^{-7}$ | $4.2 \times 10^{-13}$ |
> >
> > As you highlight, in the explicit gradient task, the model is provided _much more supervision_ (equivalent to a step-by-step algorithmic rollout) than can be expected in practical settings, e.g. fluids modeling. As such, the fact that Transformers struggle to solve even this simple problem suggests they may face fundamental difficulties scaling to more realistic and challenging settings.
> >
> > ## W3: Transferring our insights beyond linear regression
> > This is another important point -- thank you for highlighting it! We focus on least squares as a first step for two key reasons:
> > - **High precision is crucial for scientific ML**, where small errors can compound catastrophically (e.g., in climate modeling and turbulence simulations). Least squares provides a controlled setting to examine fundamental precision limitations.
> > - **Least squares serves as a useful primitive for differential equations.** As a first step towards exploring these connections, we include new experimental results on solving ODEs in-context in Appendix C.4. Our insights about limited precision and generality transfer directly to the ODE setting, whereas our proposed techniques achieve high precision.
> >
> > Please refer to the common response for more details.

---

> > > ### Comment · Reviewer_frTT · 2024-11-25
> > >
> > > Thanks for the clarifications. I find that my concerns have been adequately addressed in the revised manuscript and am happy to raise my score.

---

> > > > ### Author Response · Authors · 2024-11-25
> > > >
> > > > Thank you for your quick response and for taking the time to review our revisions! Your feedback helped us significantly improve the clarity and presentation of our work. We appreciate your engagement during this discussion period!

---

### Author Response · Authors · 2024-11-24
**General Response**

We thank the reviewers for their time and effort reviewing our work and for their constructive comments! Reviewers appreciated the novelty of our insights about the precision limitations of Transformers [frTT, TtTT, XPf9]; in particular, Reviewer TtTT noted our findings are particularly surprising as they challenge “a hypothesis supported by a plethora of recent literature that simplified and standard transformers learn to perform [gradient descent] on linear regression”. Reviewers found our experiments “thorough and well-executed” [frTT, TtTT] and our theoretical results “interesting and well-substantiated” [TtTT, XPf9].

In this common response, we:
- contextualize the relevance of our contributions within the in-context learning and SciML literatures;
- demonstrate the generality of our insights by providing new experiments on the more practical setting of _in-context ODE solving_; and
- summarize changes to our manuscript.

Please find our comments for individual reviewers in their respective threads.
# Contextualizing our contributions
## In-context learning
As noted by Reviewer TtTT, **our observations about the precision limitations of Transformers directly challenge the prevailing hypothesis that Transformers implement in-context learning via gradient descent.** This hypothesis is argued by a long line of recent work [1,2,3,4,5], which is focused on the setting of in-context linear regression. In our work, we provide three pieces of evidence that suggest this story is _incomplete_:
1. **Transformer precision saturates with depth.** For the simplest least squares problems (small, noiseless, fully determined systems), we scale Transformer depth and show that MSE saturates $1,000,000 \times$ worse than we expect based on float32’s machine epsilon. In contrast, gradient descent provably converges to machine precision on least squares.
2. **Transformers are brittle to out-of-distribution inputs.** In contrast to theoretical convergence guarantees for gradient descent on least squares $Ax=b$, we observe that Transformers are not robust to changes to the distribution of $b$. We show that a simple rescaling of inputs by a factor of $10 \times$ worsens MSE by $100,000,000 \times$.
3. **Transformers struggle to implement basic linear algebra primitives precisely.** Specifically, we show Transformers struggle to precisely implement simple floating-point multiplications both empirically (scaling Transformers in depth, width, and training iterations) and theoretically (proving that 1-layer causal softmax Transformers cannot exactly implement multiplication). Since implementing gradient descent on least squares requires precise multiplications, our findings suggest that it is unlikely the “algorithm” Transformers learn to perform is gradient descent.

Our findings highlight that Transformers learn qualitatively different algorithms from gradient descent that are limited in both precision and generality. Our proposed techniques represent a first step towards developing model training recipes that can learn proper generalizable algorithms, with potential implications for both theoretical understanding and practical capabilities of sequence models.
## Scientific ML
Reviewers frTT and TtTT raised important questions about the relevance of learning gradient descent (GD) iterates with ML models in practice and about the generalizability of our insights to more realistic settings, e.g. for scientific tasks. We address these points here.

We first highlight that our work builds upon a broad effort in the scientific ML literature towards tackling problems (e.g. PDEs) traditionally solved with numerical methods [6,7,8]. We agree that learning to perform gradient descent with ML on least squares may be of limited practicality for its own sake. Rather, we think our results may be interesting to the broader scientific ML community as a _fundamental analysis of precision bottlenecks_ within typical training setups.

We point out that the importance of high-precision ML is well-established. Specifically in scientific settings, **high-precision ML presents a unique challenge** that is crucial for:
- Accurately combining equational and data-driven modeling. This includes equation discovery [9], e.g. recovering equational knowledge from data, and physics-informed learning [10], e.g. enforcing equational knowledge within a model.
- Maintaining stable temporal rollouts in simulations of nonlinear phenomena, e.g. climate or turbulence modeling, where small errors can accumulate exponentially when making long time-horizon forecasts [11].

Although the SciML community has made exciting progress in recent years, numerical methods are still known to outperform existing ML methods in precision even on simple PDE benchmarks (e.g. refer to [12], a survey of 76 works in this space).

---

> ### Author Response · Authors · 2024-11-24
> **General Response (continued)**
>
> As far as we are aware, **we are the first to investigate and isolate effects of the model architecture and optimizer on precision in a controlled setting:** least squares. We make the following contributions:
> - We find that typical training recipes for sequence models (e.g. Transformers, Adam, and StepLR schedulers) encounter surprising precision barriers when applied to numerical tasks. Our results about the limitations of Transformers are timely. Recent works [13,14] have investigated the use of Transformers for foundation models for differential equations, so we believe it is important to highlight precision bottlenecks that are fundamental to the Transformer architecture.
> - In our controlled setting of least squares, we show that properly learning a simple numerical algorithm (GD) requires a major reworking of standard architecture and optimizer design choices. In doing so, we identify an important gap between **expressivity** and **learnability**. We believe this observation is particularly relevant to SciML, where theoretical backing for standard methods (e.g. PINNs [6], neural operators [7]) focuses on existence results like universal approximation theorems (e.g. [6,15,16]). Our work shows that expressibility is not enough -- better training procedures are also required to learn high-precision models, and our proposed techniques represent a first step towards developing them.
> # Additional experimental results on in-context ODEs
> Beyond connections to prior in-context learning work, we focus on least squares because it serves as a useful primitive for differential equations. To demonstrate this, we include new experimental results on _in-context ODE solving_, addressing comments from Reviewers frTT and TtTT. We note that solving differential equations in-context with Transformers is a framework that has been explored in recent papers [13,14,17]. Here, we follow the setup from [17] and include a comprehensive discussion of experimental details in Appendix C.4.
>
> We find that our observations from least squares transfer to the setting of in-context ODEs:
> - **Transformers struggle to learn precise solutions.** We find that a 12-layer, 9M parameter Transformer model only achieves $O(10^{-4})$ MSE, almost $10^{10} \times$ worse than float32 machine precision would imply. Furthermore, as in least squares, we observe precision saturation with model size -- we find scaling the depth of the model by $2\times$ does not improve precision.
> - **Transformers exhibit brittle generalization.** We observe that Transformers are not robust to changes to the distributions of ODE parameters, forcing functions, and boundary terms. For example, we show that for initial conditions, a simple rescaling by a factor of $10\times$ worsens MSE by $100,000\times$. Please refer to Appendix C.4 for more detailed results.
> - **Our proposed techniques obtain precise and general solutions.** Prior work [17] shows that in-context ODEs can be solved by reducing to an equivalent least squares problem. We train a BaseConv architecture on explicit gradients for the equivalent least squares problems, and obtain $O(10^{-10})$ MSE ($1,000,000\times$ better than our best end-to-end Transformers) when applied iteratively.
>
> We hope these preliminary results on differential equations demonstrate that our insights generalize to broader tasks of interest to the SciML community.
> # Summary of changes
> - Move definitions of primitives to Section 3.3 (Reviewer frTT)
> - Move theorem about inability of softmax attention to exactly express Square and Multiply to Section 3.3 (Reviewer TtTT)
> - Restructure Section 4.1 to highlight our theoretical contributions: Theorem 4.1 (zero population loss implies perfect recovery of primitives) and Theorem 4.2 (universal approximation) (Reviewer frTT, TtTT, XPf9)
> - Discuss connections between $k$-step GD, explicit GD, and standard in-context least squares tasks in Section 5 (Reviewer frTT)
> - Incorporate additional related work (Reviewer TtTT), including relevant SciML papers, in Appendix A.2
> - Restructure Appendix C:
> 	- Move discussion of LayerNorms/MLPs ablation to Appendix C.2
> 	- Include preliminary results on in-context ODEs in Appendix C.4 and Section 5 (Reviewers frTT, TtTT)
> - Add a formal proof of the inability of softmax attention to exactly express Multiply in Appendix D (Corollary D.32) (Reviewer TtTT)

---

> > ### Author Response · Authors · 2024-11-24
> > **General Response (references)**
> >
> > ## References
> > 1. Ekin Akyürek, Dale Schuurmans, Jacob Andreas, Tengyu Ma, and Denny Zhou. What learning algorithm is in-context learning? investigations with linear models. ICLR 2023.
> > 2. Johannes Von Oswald, Eyvind Niklasson, Ettore Randazzo, João Sacramento, Alexander Mordvintsev, Andrey Zhmoginov, and Max Vladymyrov. Transformers learn in-context by gradient descent. ICML 2023.
> > 3. Deqing Fu, Tian-Qi Chen, Robin Jia, and Vatsal Sharan. Transformers Learn to Achieve Second-Order Convergence Rates for In-Context Linear Regression. NeurIPS 2024.
> > 4. Yu Bai, Fan Chen, Huan Wang, Caiming Xiong, and Song Mei. Transformers as statisticians: Provable in-context learning with in-context algorithm selection. NeurIPS 2023.
> > 5. Ruiqi Zhang, Spencer Frei, and Peter L Bartlett. Trained transformers learn linear models in-context. JMLR 2023.
> > 6. M. Raissi, P. Perdikaris, G.E. Karniadakis. Physics-informed neural networks: A deep learning framework for solving forward and inverse problems involving nonlinear partial differential equations. Journal of Computational Physics 2019.
> > 7. Zongyi Li, Nikola Kovachki, Kamyar Azizzadenesheli, Burigede Liu, Kaushik Bhattacharya, Andrew Stuart, Anima Anandkumar. Fourier Neural Operator for Parametric Partial Differential Equations. ICLR 2021.
> > 8. Tung Nguyen, Johannes Brandstetter, Ashish Kapoor, Jayesh K. Gupta, Aditya Grover. ClimaX: A foundation model for weather and climate. ICML 2023.
> > 9. Eric J. Michaud, Ziming Liu, Max Tegmark . Precision Machine Learning. Entropy 2023.
> > 10. Multi-stage Neural Networks: Function Approximator of Machine Precision. Journal of Computational Physics 2024.
> > 11. Uriel Frisch. Turbulence: The Legacy of A. N. Kolmogorov.
> > 12. Nick McGreivy, Ammar Hakim. Weak baselines and reporting biases lead to overoptimism in machine learning for fluid-related partial differential equations. Nature Machine Intelligence 2024.
> > 13. Liu Yang, Siting Liu, Tingwei Meng, and Stanley J. Osher. In-context operator learning with data prompts for differential equation problems. PNAS 2023.
> > 14. Maximilian Herde, Bogdan Raonić, Tobias Rohner, Roger Käppeli, Roberto Molinaro, Emmanuel de Bézenac, Siddhartha Mishra. Poseidon: Efficient Foundation Models for PDEs.
> > 15. Lu Lu, Pengzhan Jin, George Em Karniadakis. DeepONet: Learning nonlinear operators for identifying differential equations based on the universal approximation theorem of operators. NeurIPS 2024.
> > 16. Nikola Kovachki, Samuel Lanthaler, Siddhartha Mishra. On Universal Approximation and Error Bounds for Fourier Neural Operators. JMLR 2021.
> > 17. Jerry W. Liu, N. Benjamin Erichson, Kush Bhatia, Michael W. Mahoney, Christopher Ré. Does In-Context Operator Learning Generalize to Domain-Shifted Settings? The Symbiosis of Deep Learning and Differential Equations III, NeurIPS Workshop 2023.

---

### Meta-Review · Area_Chair_M17R · 2024-12-24

**Metareview:**

The paper examines the in-context learning ability of transformers and shows that it cannot learn in-context linear regression solutions with machine precision or numerical generality. Additionally, the paper proposes a new approach for training to machine precision. The paper also provides empirical evidence to verify these claims.

The reviews for the paper were borderline to leaning positive. Most reviewers acknowledge that the paper solves an interesting problem and highlight the novelty. While the practical applicability of this approach is somewhat arguable, I think it is worth to let the community know about this issues and broadly understand/address them. I recommend acceptance.

**Additional Comments On Reviewer Discussion:**

The reviewers acknowledge the novelty of the approach. There were concerns regarding practical applicability of the exact setting discussed in the paper. The authors addressed these weaknesses reasonably during the rebuttal.

---

### Decision · Program_Chairs · 2025-01-22

Accept (Poster)